pattern recognition/artificial intelligence/theory of computing

natural language processing, narrative theory

**Author for correspondence:**
Pavan Holur
e-mail: pholur@ucla.edu

†These authors contributed equally to this study.

# Modelling social readers: novel tools for addressing reception from online book reviews

Pavan Holur[1,†], Shadi Shahsavari[1,†], Ehsan Ebrahimzadeh[1], Timothy R. Tangherlini[2,†] and Vwani Roychowdhury[1,†]

[1]Department of Electrical and Computer Engineering, University of California Los Angeles, Los Angeles, CA, USA
[2]Department of Scandinavian, University of California Berkeley, Berkeley, CA, USA

 PH, 0000-0002-8495-9416

Social reading sites offer an opportunity to capture a segment of readers' responses to literature, while data-driven analysis of these responses can provide new critical insight into how people 'read'. Posts discussing an individual book on the social reading site, *Goodreads*, are referred to as 'reviews', and consist of summaries, opinions, quotes or some mixture of these. Computationally modelling these reviews allows one to discover the non-professional discussion space about a work, including an aggregated summary of the work's plot, an implicit sequencing of various subplots and readers' impressions of main characters. We develop a pipeline of interlocking computational tools to extract a representation of this reader-generated shared narrative model. Using a corpus of reviews of five popular novels, we discover readers' distillation of the novels' main storylines and their sequencing, as well as the readers' varying impressions of characters in the novel. In so doing, we make three important contributions to the study of infinite-vocabulary networks: (i) an automatically derived narrative network that includes meta-actants; (ii) a sequencing algorithm, REV2SEQ, that generates a consensus sequence of events based on partial trajectories aggregated from reviews, and (iii) an 'impressions' algorithm, SENT2IMP, that provides multi-modal insight into readers' opinions of characters.

## 1. Introduction

Online reader comments about works of literary fiction offer an intriguing window onto how people read. Although previously ignored in the realm of computational literary studies, recent

studies have shown the value of these resources in understanding reader reception [1–3]. These comments can provide useful insight into how readers imagine the main storylines of a novel, how they understand the fictional struggles of characters and how they develop varying impressions of the work. Taken together, the reviews of a single novel provide a view onto the collective imagining of what is important in the novel, including aspects of plot, the interactions between various characters, and even the metadiscursive space of authors, genre, critics, film adaptations and movie stars. These reviews thus provide impetus for a data-driven analysis of readers' responses to a work of literary fiction [4]. They also help us understand how readers create an 'imagined community' of readers, an extension of Fish's notion of 'communities of interpretation', engaged in the collective enterprise of literary analysis [5,6].

'Reader response theory' experienced a brief and productive heyday in literary theory during the 1960s. Despite the considerable attention this theoretical premise received, the focus of much of this work centred on the hypothetical and highly theorized 'individual reader'. This theoretical orientation was expanded to include groups of readers, and led in part to Stanley Fish's important contributions concerning 'communities of interpretation'—groups of readers who, through their shared experiences, converged on similar readings of texts [6,7]. The consideration of broad-scale responses of readers to works of fiction, however, remains understudied, not because of a lack of interest on the part of literary historians and theorists, but because of a lack of access to those readers' responses, and a lack of methods to address this at times noisy data. While there is considerable investigation into how groups of individuals are likely to read (or to have read), investigations of how large groups of people outside of an experimental setting respond to the same work of fiction have only recently become possible [8].

The advent of social reading sites on the Internet that allow individual readers to join in wide-ranging discussions of individual works of fiction through reader-generated reviews and comment threads on those reviews has enabled a revisiting of fundamental questions of how people respond to literary fiction [1,4,7,8]. Perhaps best known among these sites in the USA is *Goodreads* that, along with sites like it, represents an online attempt to reproduce the face-to-face space of book clubs and library groups, where there is no 'right' answer to reading the work (as there might be, at least implicitly, in a classroom), nor any hierarchy of critical insight (as there might be in a forum where professional reviewers or literary critics might dominate the conversation) [7,9]. Because these sites archive the reader reviews and the ensuing comment threads on those reviews, they offer an opportunity to explore computationally how people respond to individual works of fiction, and how they explore such a work as communities of interpretation emerge [4]. Since these reviews are unguided explorations of fiction, and since many of the readers read and review purely for entertainment, it is unlikely that the readings encode the types of literary-theoretical engagement found in academic work. Instead, the reviews encode a popular engagement with literature, focusing on aspects of plot, character and the struggles of the characters in their fictional worlds. The goal of our work is to model the collective expressions of thousands of readers as they review the same work of fiction. Can we discover what they see as important? Can we discover divergences in their readings? Do their reviews provide us with information about reading, remembering and retelling? And do they tell us anything about the process of writing a review itself?

As is common on most social media platforms, some reviewers on *Goodreads* engage more than others in back-and-forth conversations in the comment threads in order to clarify or justify a point. While reviewer interactions occur and might bias a particular post one way or another, we view the review data collectively as the result of a 'natural experiment' on a collaborative online forum for free dialogue. By virtue of reviewing a book, and in engaging with the reviewing platform, reviewers have joined an imagined community of book readers whose interactions, whether in the comment threads or in the asynchronous nature of reading others' reviews, have created a community (or communities) of interpretation [5,6]. When a reader reads, they necessarily read in the context of what they have read before; and when reviewers write a review, it is in this spirit of contributing to an ongoing and growing interpretive body of knowledge about the work in question. In the methods presented here, we do not mine the network of the interconnected users on the platform, but instead restrict the analysis to the network of the narratives they recall and modify. Any deliberations, debate and uncertainty among users flow through the tools we present below, which remain invariant to this collective bargaining among reviewers. An explicit example can be observed in our representation of character complexity wherein dissonance among users about their impressions of a character, whether expressed in conversation with one another or to the *Goodreads* platform itself, blind to others' opinions, is captured in our complexity histogram (figure 7).

In our work, reader reviews, in the aggregate, converge on an underlying, yet broad, narrative framework [7]. This framework, based on a relaxation of the narrative theories of Algirdas Greimas [10–12], is represented as a narrative network where the nodes are actants and the edges are interactant relationships. Since reader reviews often spill into discussions of other works by the same author, works in the same genre or film adaptations of the novel, the actants in the network are expanded beyond the canonical census of a novel's characters to include that metadiscursive space.

Importantly, book reviews also encode the varying impressions that readers form of the novel's characters. By reading reviews at 'Internet scale', one can consider the extent to which partial reviews contribute to a representation of the character-interaction network of the target novel, how readers' partial comments on the sequence of events in a novel can be aggregated to produce a series of sequential directed acyclic paths through that novel (i.e. different interpretations of the storyline or storylines present in the work) and how readers build a collective understanding of complex characters, even if their individual views of the characters may not capture that same complexity. These problems can be seen as part of a formal, computational assessment of readers' response to even quite long and complex novels, such as *The Hobbit* or *To Kill a Mockingbird*, to name but two of our target works.

To address these challenges, we develop a pipeline of interlocking computational tools to extract and aggregate a meaningful representation of the reader-generated shared narrative framework modelled as a network. This framework is based on a structured open-world infinite-vocabulary network of interconnected actants and their relationships. In this work, we add two new algorithms to the pipeline, which was originally developed to determine the underlying generative narrative framework for social media conversations [13–15]. The first algorithm, REV2SEQ, determines the readers' collective reconstruction of the plot of the target work by sequencing the interactant relationships and representing these as paths through the overall narrative framework graph. The second algorithm, SENT2IMP, presents a representation of the collective, at times differing, opinions of characters in a novel. In addition, we expand the narrative framework graph to include the important metadiscursive and extra-diegetic nodes noted above, thereby providing a fuller picture of how readers contextualize their engagement with a work of fiction.

## 2. Related work

Work on reader response theory has, in the past, focused on the responses of individual readers to the act of reading [16,17]. Theoretical work in this area supported the development of the idea that the process of reading is constitutive of the text itself [18]. In this view, it is the reader and the individual reading of the text, more so than the author, that takes a position of primacy. The study of reader response has had a considerable impact on a broad range of fields outside of literary studies, including cognitive psychology [19] and phenomenology [20,21]. The phenomenology of reading extends to explorations of how different readers can exhibit a wide range of responses to the same piece of fiction [22]. Yet, because of the difficulty of capturing records of non-specialists' experiences of a particular work of fiction, a great deal of early work was done in controlled classroom or laboratory environments, with limited numbers of readers and relatively short texts [23].

One of the foremost developments in reader response theory was the introduction of the concept of interpretive communities, anchored in the work of Fish [6]. He suggests, in part, that the interpretation of a text is culturally anchored, non-deterministic and mutable, yet not infinite in its scope. Rather, each reader reads a work from this position as a member of a community of readers, proffering limits on how a work can be read [9]. Consequently, social reading sites may capture the collective readings of members of various interpretive communities and the boundaries of those possible readings, a suggestion confirmed by recent work [2,3,8].

The computational analysis of narrative and literary fiction has a long history at the intersection of literary studies and artificial intelligence [24]. The important distinction between *fabula*, the chronological order of events for a narrative, and *syuzhet*, the ordering of those events in the work of fiction, became fundamental concepts in the study of narrative [25,26]. Subsequently, the computational possibilities of morphological approaches to narrative such as Propp's study of the Russian fairy tale [27] and other structural theorists were widely recognized as opportunities for devising computational approaches to the study of narrative [28,29]. The intersection of reader response theory, narrative theory and artificial intelligence has also been explored in the context of understanding character complexity and narrative unpredictability [30], echoing ideas in early artificial intelligence research [31].

Social reading platforms offer an opportunity to explore these aspects of reader response and narrative structure outside of the laboratory or classroom setting [1,2,4,32–34]. Work on these platforms has focused on an ethnography of online reading [35], considerations of reader review sentiment [36], how gender and intimacy are intertwined in reviews [37] and broader linguistic characteristics of the reviews [38]. Other studies have focused on *Goodreads* in the context of socially networked reading [39], the moderation of reader responses [40], recommender systems and participants' motivations for reviewing [41], grassroots genre classification [3] and prediction of the success of a book given certain features in the reviews [42]. There has also been recognition of the disconnect between easy-to-gather metrics such as positive ratings and an understanding of how people actually read or remember a book [43].

A considerable and important body of work has emerged on studies of the social reading platform, WattPad, that has as a main feature the facility for readers to comment directly on a paragraph, as well as see other readers' comments on that same paragraph [2,8]. Studying these comments across a broad range of writing allows for a macro-level analysis of reader engagement, the behaviours of different classes of readers, as well as similarities in reading practice across language groups [2]. Other work on this corpus has shown the relationship between the sentiment arcs in the work of fiction compared with the sentiment arcs of the readers in response to the work based on an aggregation of the readers' marginal comments [8].

Taken as a whole, these lines of inquiry allow one to explore how non-specialist readers experience a piece of the literature. Indeed, exploring the responses of thousands of readers to the same work of fiction may come close to a natural experiment addressing the reader response problem, where the readers themselves choose what they share about their response to the work [1]. In so doing, the readers may also be providing insight into the formation of interpretive communities [3], while simultaneously offering clues as to what they value in works of fiction, how they understand the complexities of characters, and how they remember and retell the events and the sequence of those events that they feel are most important to include in their review.

The methods presented here address two main aspects of this reader response problem: (i) the aggregation of the readers' conceptions of the narrative framework, and the possible paths through that framework, and (ii) the derivation of an overview of the readers' range of impressions of characters and events in the novel, as well as their impressions of meta-narrative features.

Existing computational approaches to modelling stories and event sequences use labelled original story data via numerical embeddings to train a supervised event sequence generator [44,45]. The generated embeddings are then used as input to recurrent neural networks (RNN) for modelling *sequences* of events and predicting the probability that one event succeeds another. Our research differs in several key ways: (i) While these models are largely supervised, our pipeline estimates sequences in an unsupervised way: reviews are not inherently sequenced relative to the events in a story. (ii) While these prior works use stories directly for training, we use *reader reviews* of a literary work rather than the work itself. This secondary data lends our analysis to the study of reader reception in addition to standard knowledge extraction. (iii) While we do use pretrained embeddings—in this case BERT—we present an *explainable* algorithm to extract story sequences comparable to the ground truth, in addition to summarizing other facets of a story's reception.

Evaluation of the sentiments expressed in reviews often relies on high-level sentiment analysis and/or fixed-vocabulary opinion mining and classification tasks [8,46,47]. In these works, the classes of sentiments or opinions are fixed, and the object of classification is the readers' responses to the work of fiction as a whole or, in the case of recent work on WattPad, to individual paragraphs in the work [2]. Our approach, by way of contrast, considers sentiment and genre at the phrase-level of reviews and maintains a robust transparency into the raw data throughout the pipeline. In addition, our analysis of reader response focuses on readers' discussions of actants associated with the novel, providing a complementary approach to the paragraph-level, often event-based approach, made possible by the analysis of readers' marginal comments such as those found in WattPad [2,8]. The aggregation of these types of sentiments, or *impressions*, is an open-ended task: since we do not know the range of impressions a literary work might create for a reader, our impression-mining is fully unsupervised. The open-ended nature of this task is directly related to the burgeoning interest in developing infinite vocabulary knowledge graph (KG) generation pipelines [48]. In our KG context, *infinite vocabulary* means that the node and edge labels are not defined according to a schema or rule *prior* to data aggregation. Instead the goal is to estimate that schema from the data. A majority of existing approaches aggregate relationships via OpenIE [49] (from the Stanford Natural Language Processing (NLP) Toolkit) or a comparable relationship extractor. Once the infinite vocabulary network is estimated, a post-processing block attempts to compress the KG via existing ontologies such as WikiMedia [50] and

YAGO [51]. Our pipeline approaches KG creation as a *narrative modelling* problem which is, by nature, infinite-vocabulary: these narratives feature rich characters, complicated storylines and numerous events and situations. The model we develop boosts the off-the-shelf OpenIE relationship extractor with interconnected blocks of co-reference resolution and other filtering techniques that increase the recall of the relationship extraction pipeline and the resulting KG [14].

## 3. Data

We created a corpus of reader reviews for five novels from *Goodreads*: *Frankenstein* [52], *Of Mice and Men* [53], *The Hobbit* [54], *Animal Farm* [55] and *To Kill a Mockingbird* [56]. We chose these books from the list of most commonly rated fictional works on the site, focusing on works with ratings greater than 500 000. We then considered the number of reviews for these highly rated novels, recognizing that many highly rated novels have considerably fewer reviews than ratings. At the time of data collection, for example, *The Hobbit* had more than 2.5 million ratings, but only 44 831 reviews.

Once we had a candidate list of works to consider, we reduced that list to our five target novels, with the goal of selecting works that had significantly different types of characters (humans, supernatural creatures, animals), different narrative structures, and that employed a variety of narrative framing devices (frame narratives, flashback, etc.). *The Hobbit*, for instance, is a complex multi-episodic narrative that unfolds in a largely linear fashion, following in broad strokes the well-known 'hero's journey' plot [57]. It uses a mixture of character types, including wizards, hobbits, dragons and dwarves, while the setting is a fictional fantasy world modelled loosely on the Nordic mythological one. By way of contrast, *Of Mice and Men* is a realistic novella set in depression-era California with a small cast of human characters, and instantiates Vonnegut's 'From Bad to Worse' plot [58]. *To Kill a Mockingbird* is a complex intertwined narrative told from the first-person point-of-view of a child, and engages with nuanced ideas about justice and race in America. *Frankenstein*, written as an epistolary novel and thus told largely in flashback, has multiple, complex characters, both human and non-human. Its plot is largely linear with a clearly recognizable hero and villain. *Animal Farm*, in contrast, has a very large cast of characters, many of them anthropomorphized animals, with a plot that cycles through episodes that bear striking resemblance to earlier ones (such as the rebuilding of the windmill).

The reviews of a target novel were extracted using a custom pipeline since existing corpora either did not contain the correct books, enough book reviews, or enough book reviews of appropriate length. For example, the UCSD *Goodreads* review dataset [33,34], despite having an enormous number of book reviews, did not have enough book reviews of appropriate length per book that our methods require.

After choosing our target novels, we designed a scraper to operate in accordance to the specifications of the *Goodreads* site. After downloading the reviews for each book, we filtered and cleaned them to avoid problems associated with spam, incorrectly posted reviews (i.e. for a different novel), or non-English reviews. We also filtered for garbled content such as junk characters and for posts that were so short as to be unusable. Following the process of filtering, we had the following number of eligible reviews per book: *Frankenstein* (2947), *The Hobbit* (2897), *Of Mice and Men* (2956), *Animal Farm* (2482) and *To Kill a Mockingbird* (2893). These reviews can be found in our data repository [59].

We then passed the scraped reviews to the first data-preparation modules of our pipeline. During this largely NLP-based process, we found two main types of phrases: (i) Plot phrases containing both the actants and their relationships. These phrases describe what a reviewer believes happened to a subset of the actants, and how they interacted with each other and, consequently, these phrases are of primary interest to us. (ii) Opinion phrases that reveal a reader's opinions about the book, the author, or the characters and events in the book. The relationships we extract from these phrases are the predominant ones when aggregated across all readers' posts.

## 4. Methodology

A network of characters (and other actants) connected by inter-character relationships—typically verb phrases—can serve as a useful representation of an aggregated model of readers' understanding of the narrative scope of a novel. This model has the advantage that it can show multiple, at times competing, claims to the underlying storyline (or storylines) of the target work. In previous work, we introduced a pipeline addressing two important tasks that are instrumental in constructing this network representation: entity mention grouping (EMG) and inter-actant relationship clustering (IARC) [60], which we summarize below.

The EMG task is a labelling process that aggregates multiple entity mentions from the extracted relationship phrases (subject $\hat{s}$ or object $\hat{o}$) into a single character. This aggregation is accomplished through an evaluation of the similarity between a pair of entity mentions by observing their interactions with other entity mentions. For example, in *The Hobbit*, the two entity mentions, 'Bilbo' and 'Baggins', frequently interact with the entity mention 'Gandalf', and with semantically equivalent relationships; as a result, these two mentions, 'Bilbo' and 'Baggins', probably refer to the same character, here the hobbit, 'Bilbo Baggins'.

To formulate this task, let the set of entity mentions empirically observed in reviews be $\hat{E}$. A smaller character set $E$ refers to a finite vocabulary of distinct characters in a literary work. The EMG step is then defined as a surjective function $f : \hat{E} \rightarrow E$ that maps entity mentions to characters. The resultant mapping of entity mentions to characters in the EMG task provides a semantically informed aggregation tool for the original corpus of relationship tuples. Tuples sharing entity mentions mapped to the same character can now be aggregated to form larger relationship sets between a pair of characters (as opposed to a pair of entity mentions). For example, in *Of Mice and Men*, the entity mentions 'George' and 'Milton' are successfully mapped to the character 'George' with the EMG task.

The IARC task, on the other hand, is designed to aggregate relationship phrases generated as output from a relationship extraction module. A relationship tuple consists of a subject mention, relationship phrase and object mention $(\hat{s}, \hat{r}, \hat{o})$ that is directly extracted from a review and thus contains partial information about the structure of the underlying narrative model [60]. For every ordered pair of characters $\{e_i, e_j\}$, we obtain an aggregated set of relationship phrases from the reviews, $\hat{R}_{ij}$, that connects $e_i$ to $e_j$.

The EMG task implicitly aids in the aggregation of larger sets of relationship phrases since it aggregates entity mentions for each character. $\hat{R}_{ij}$ is the union of all the relationship phrases between entity mentions that compose $e_i$ and $e_j$. We seek to cluster these relationship phrases in $\hat{R}_{ij}$ and assign each resulting cluster to a label in a set $R_{ij}$. This process of clustering, inter-actant relationship clustering (IARC), can accordingly be defined by another surjective function $g_{ij} : \hat{R}_{ij} \rightarrow R_{ij}$. Specific details about assembling the set of labels $R_{ij}$ for the IARC task are provided in the relevant subsections below. For example, a few of the relationship phrases between 'Atticus' Finch and 'Tom' Robinson in reviews include: defends, is defending, protects, represents, supports. These phrases are semantically similar and appear in the same cluster; this cluster is subsequently labelled 'defends'.

In earlier work, relationship extraction along with the two-step process (EMG, IARC) to condense the relationship tuple space was applied to the corpus of reader reviews for four works of literary fiction: *Of Mice and Men*, *To Kill a Mockingbird*, *The Hobbit* and *Frankenstein* [60]. The resulting aggregated networks, which we label 'narrative frameworks', represent the broad consensus across all the reviews of the story network, with each node in the network representing a character and each directed edge representing a relationship between a pair of characters. Importantly, the narrative framework graph is derived entirely from the reader reviews.

This preliminary work lacked important KG tools helpful for addressing problems in reader response and narrative modelling. For example, in addition to estimating the narrative framework of a novel, which may reflect the readers' understanding of important characters and inter-character relationships, it is useful to estimate a larger graph that also includes key, often metadiscursive or extra-diegetic, actants related to the novel's *reception* such as the 'author', a 'reviewer' or even references to cross-media adaptions such as the 'film director' and 'actors' who played particular characters in the film. Similarly, while earlier work generated story network frameworks that were static, summarizing the entire story, estimating the temporal dynamics of the story underlying the reviews can greatly expand the usefulness of these graphs. This dynamic view of the novel can provide insight into how reviewers remember the unfolding of the narrative. Last, while the previous narrative graphs highlighted the interactions *between* actors, they did not model the reviewers' *impressions* of the actors, important information in the context of an analysis of reader response.

The following sections describe the rest of our methodology that: (i) extends the network generation pipeline to create expanded narrative framework network graphs; (ii) introduces a new application of this pipeline for story *sequencing,* and (iii) profiles frequently mentioned characters in the selected novels based on context-specific reviewers' impressions.

## 4.1. Expanded story network graph

While the nodes in a *regular* narrative framework graph for a novel are derived from an associated external resource such as a canonical character list from *SparkNotes* or the work itself, the discovered relationships from our pipeline (edges) are extracted from readers' reviews. These extractions reveal

both readers' thoughts and their impressions of characters in the works under discussion. Not surprisingly, then, the reader-derived networks expand upon the regular narrative framework graphs, with additional nodes for film directors, authors and other interlocutors, and additional edges for these additional actants' activities with the core story.

To facilitate this expansion, we augment the regular story network for a novel with nodes for frequent entities *not* among the character mentions. First, we rank mentions based on their frequency and then we pick the top ranked entity mentions to add them to the story graph. For example, in *The Hobbit*, some of the extra candidate mentions are: ['Tolkien', 'book', 'story', 'adventure', 'movie', 'Jackson'].

Next, we find the edges that exist between these candidate nodes and the regular story network nodes. If a candidate node has significant edges with the existing characters in the story network, we augment the graph with this node. For example, the candidate node 'Tolkien' is connected to the character 'Bilbo' with the verb phrases ['to masterfully develop', 'provides', 'knocked', 'introduced']. There are other interesting but distant mentions of candidate nodes that do not have *direct* connections to nodes in the original story graph: for example, 'Jackson' (a reference to the director of *The Hobbit* film, Peter Jackson) only appears in the triplet (Jackson's changes, distort, Tolkien's original story). While this node represents the reviewers' acknowledgement of the novel's movie adaptation, we do not represent 'Jackson' as a node in our expanded graph due to its sparse connection with the rest of the story network.

As a result, our algorithm discovers the available relationships between main novel characters and each metadiscursive candidate node mention. It draws an edge only if there are more than five relationships between the candidate and a story graph character. If that candidate node has a degree of 3 or greater, it is added to the expanded story network graph.

## 4.2. REV2SEQ: a new event sequencing algorithm

As noted, each *Goodreads* review provides partial information for creating a collective summary of the reviewed work. Specifically, reviewers share their interpretation of the story's sequence of events, motivated in part by a universal understanding of time and in part to convey temporal information about the objective storyline(s) to a reader of the review [61]. Within this shared timeline, reviewers intersperse their opinions, argue over aspects of theme and plot in their comments, and offer other thoughts about style, imagery or comparisons with other works of fiction. Individual reviews sometimes highlight pivotal or critical moments in the story and their interdependence, and at other times elaborate on minutiae that have, at least for the reviewer, great importance.

We observe for example, that some reviewers of *The Hobbit* not only write (i) that 'Bilbo meets Gandalf' and (ii) that 'Bilbo faces Smaug', but also that Bilbo faces Smaug *after* meeting Gandalf. We would like to capture this additional consensus information from the collection of reviews as a means for understanding the *dynamics* of how readers imagine the sequence of events in a novel. This observation motivates the REV2SEQ tool.

Within an individual review, one can use multiple syntactic tools—including tense, punctuation, retrospective language and conjunctions—to encode temporal precedence relationships among events. Unfortunately, the NLP tools required to decode a linear timing sequence from such complex syntax are not very well developed [62]. Given that the reader reviews are short and meant for social media sharing, we hypothesize that most reviews are written in a linear manner, and that temporal precedence is usually accurately encoded by the order in which events are mentioned in the text. Any errors introduced by this assumption in the inferred temporal precedence order—for example, due to some readers using tense to indicate temporal dependence when the event mention order is actually reversed—would create cyclical temporal dependencies among events when aggregated over all the reviews. Our accompanying hypothesis is that, because of the large number of reviews in which the majority follows a linear time encoding scheme, such cyclical dependencies can be computationally resolved by incorporating a majority voting scheme in the associated algorithms. Our results, evaluated by story-aware testers, validate our hypotheses, with the output of the computational steps yielding aggregated sequencing results close to the ground truth.

In order to verify this hypothesis, we first define an event as an ordered tuple of characters connected by a relationship cluster. In other words, an event $p \in P$ is an ordered tuple of the subject character ($s$), relationship label ($r$) and object character ($o$), where $s \in E$, $o \in E$ and $r \in R_{so}$. Recall that $E$ is the set of characters as returned by the EMG task and $R_{so}$ is the set of labels that tag the relationship clusters in the IARC task. Each review, $p_k$, then consists of a sequence of events: $[(s_1, r_1, o_1), \ldots, (s_n, r_n, o_n)]$. For example, a reviewer of *The Hobbit* might mention «Bilbo Baggins»'s upbringing in «Hobbiton» before

describing «Bard»'s slaying of «Smaug». Similarly, in reviews for *Of Mice and Men*, reviewers more frequently mention that «George» finds the «ranch» before noting that «Lennie» kills «Curley's wife». If one considers all of the reviews in the aggregate, they should present a summary of the novel's storylines from the perspective of the reviewers, including a series of alternative or abbreviated paths from the beginning of the novel to its end. Consequently, the challenge is to jointly estimate *the sequence of events* that describes the story of a particular work of fiction based solely on the reviews.

## 4.2.1. Implementation details

To construct an event, we extract entity groups and relationships by running the entity mention grouping (EMG) and the inter-actant relationship clustering (IARC) functions. A new implementation of the IARC task encodes relationship phrases with entailment-based pretrained BERT embeddings [63] that are subsequently clustered with HDBSCAN, a density-based algorithm suitable for clustering BERT embeddings [64,65]. The resulting clusters are labelled with the most frequent sub-phrase in each cluster along with contextual words that best connect the subject character to the object character based on the review text from which the relationship tuples were extracted. Noisy (diffuse) clusters are filtered by HDBSCAN and further manually filtered so that we only consider high-quality relationship clusters. For example, clusters that involve relationships that are not sequenceable—such as the relationship cluster in *Of Mice and Men*: «Lennie» ['wish' {for}, 'hope' {for}] «ranch»—are avoided via a keyword search for hypothetical words 'wish' and 'hope' and also by manually parsing the resultant HDBSCAN clusters.

Occasionally, some of the automatically generated clusters are semantically similar. For example, in *The Hobbit*, we find two semantically similar clusters: «Gandalf» ['recruits', 'recruit'] «Bilbo» and «Gandalf» ['chose', 'chooses'] «Bilbo». In such cases, the clusters are merged before labelling and the final event unambiguously describes the action of «Gandalf» *recruiting* «Bilbo» for the task of accompanying the Dwarves. These labels constitute the range of the mapping for the IARC task $\mathcal{R}(g_{ij})$ where $i$ refers here to «Bilbo» and $j$ to «Gandalf».

Events are not aggregated from relationship tuples where: (i) the parent sentences are hypothetical (e.g. ones that contain 'would', 'could', 'should'); (ii) subject and object phrases have headwords that are identical; are in a list of stopwords that contain mentions such as ['I', 'We', 'Author']; or do not have associated characters produced by the EMG task; or (iii) parent relationship phrases start with verbs in their infinitive form. If the relationship part of a relationship tuple features a single word (generally a verb), it is reduced to its lemmatized form. These filters are applied while retaining the relative ordering of the events in each review.

The event sequence described by every review is aggregated into a shared precedence matrix $\hat{M} \in \mathbb{R}^{(|P|+2) \times (|P|+2)}$ where $|P|$ is the number of distinct events. In this matrix, one row and one column is reserved for each event with two additional rows and columns for the objective START and objective TERMINATE of a story. $\hat{M}[i, j]$ aggregates the number of reviews in which event $p_i$ precedes $p_j$.

Matrix $\hat{M}$ is preprocessed as follows: (i) the row representing START is set to a constant to model a reviewer's starting event in their own partial sequence of events as a uniform distribution; (ii) similarly, the column representing TERMINATE is set to a very small constant to represent the likelihood that a reviewer might terminate their review after any event; (iii) finally, the rows of $\hat{M}$ are normalized to model the *likelihood* of a subsequent event mapped to a particular column based on a current event mapped to a particular row. We call this processed matrix $M$.

Matrix $M$ can now be re-imagined as modelling the empirical likelihood of directed edges in a graph $\mathcal{G}$ whose nodes are the events mapped to the rows and columns of $M$. In an ideal scenario, $\mathcal{G}$ would be a directed acyclic graph (DAG) with every event $p_i$ unambiguously preceding or succeeding another event $p_j$. This, however, is not the case in our estimation—our approach uses event precedence as a *proxy* for event sequences, and this proxy is noisy. This noise manifests itself in the form of cycles, where $p_i \to p_j \dots, p_k \to p_i$ is an inconclusive and cyclical ordering of events. Therefore, we seek to find a way to resolve these cycles.

Estimating a DAG from $\mathcal{G}$ by removing the minimum number of edges is a candidate solution but it is also an NP-complete problem. We can, however, perform two trivial steps toward accomplishing this goal: (i) remove isolates (single event disjoint paths from START to END) and self-loops; and (ii) remove two-node cycles by comparing $M[i, j]$ and $M[j, i]$ for all $(i,j)$ pairs and keeping only the larger value—a majority rule. To remove longer cycles and to create the final event sequence network, we introduce the sequential breadth first search (SBFS) algorithm which employs a greedy heuristic to remove edges that form loops.

## 4.2.2. Sequential breadth first search algorithm

The SBFS algorithm (see algorithm 1) is motivated by Topological Sorting, a classical algorithm used to query whether a directed graph is indeed acyclic. Starting from a node with 0 in-degree, the naive algorithm parses the directed edges of the graph to generate a flattened and linear representation. If the algorithm is unable to create this representation, the graph is not a DAG. Our algorithm is reminiscent of the Topological Sorting algorithm in that it allows an event (node) discovered in an earlier iteration of the sorting to be pushed further along the shared linear representation if another event discovered in a later iteration in fact precedes it. This linear representation is interpreted as the estimated objective timeline of the story.

On the other hand, our implementation differs from a simple topological sorting in that, at each step, multiple vertices can co-occur *in parallel* to one another. This feature is critical in our use case because stories often have parallel plot lines which a simple and linear topological sorting algorithm would not explore. Furthermore, a greedy heuristic removes cycles of length greater than 2 (note that self-loops and frequent two-node cycles have already been filtered). If an encountered edge creates a cycle during the construction of a DAG, that edge is broken. This sequencing algorithm satisfies precedence constraints greedily and models closely the iterative steps that a human might take in order to resolve the event sequence for a novel: having observed an event, a human would explore all potential succeeding events and resolve precedence conflicts based on their existing worldview of the event sequence.

---

**Algorithm 1.** Sequential breadth first search (SBFS).

**Input:** $M$, the filtered precedence matrix
**Output:** $\mathcal{G}$, the consensus event sequence, $T$, the resulting time-steps

CURRENT TIME $= 0$
Current Stack, $s_{curr} \leftarrow [0]$
Next Stack, $s_{next} \leftarrow []$
Precedence matrix for the final event sequence, $M' \leftarrow$ zeros like $M$

**while** $s_{curr}$ is not empty **do**
 **for** pop element $i$ from $s_{curr}$ **do**
 Succeeding events to $i$: children = indices where $(M[i][:] > 0)$
 **for** child in children **do**
 **if** there already exists a path in $M'$ from child to $i$ **then**
 Ignore path $i \rightarrow$ child causing cycle.
 **else**
 $M'[i][\text{child}] = 1$
 $T[\text{child}] = \text{CURRENT TIME} + 1$
 **if** child not in $s_{next}$ **then**
 push child into $s_{next}$
 **end if**
 **end if**
 **end for**
 **end for**

 $s_{curr} \leftarrow s_{next}$
 $s_{next} \leftarrow []$
 CURRENT TIME $=$ CURRENT TIME $+ 1$
**end while**
$\mathcal{G} \leftarrow$ graph of $M'$

---

The output of the algorithm is a directed edge cover $\mathcal{G}'$ of $\mathcal{G}$ and associated time steps per event $T(v) \ \forall v \in V(G)$. The resulting event sequence network now describes a generalized topological ordering of an estimated DAG, ordered by time steps $T$. Extraneous edges are automatically removed from $\mathcal{G}'$ such that if there is a path from $e_k$ to $e_j$ through $e_i$, the direct edge from $e_k$ to $e_j$ is deleted.

During visualization, directed edges always face downwards and, as a trivial verification, the node representing START appears at the top of the event sequence and the node representing TERMINATE, appears at the bottom.

### 4.2.3. Evaluating the sequencing networks

To quantify the accuracy of these sequencing results, we asked $N_{\text{rev}} = 5$ independent reviewers to evaluate every edge in the graphs for each novel by labelling them as 1: *correct*, 0: *incorrect* or X: *unsure*. Those marked *X* are globally labelled as either all 0 or all 1 to obtain the lower bound and upper bound (along with the error bound) for the algorithm's performance. Reviewers were required to have read the stories and they used *SparkNotes* as a reference to disambiguate otherwise obscure precedence relationships. Each reviewer reviewed all five networks to negate any variability due to implicit human biases across the stories. Edges that comprised a source labelled START or a target labelled TERMINATE were ignored in the computation of accuracy: these specific source and target labels are not intrinsic to a story's plot but are instead artefacts from our algorithm—not removing these edges would result in inflated performance measures. Accuracy is measured in two ways— where $\hat{E}$ is the subset of edges in each sequence network that does not originate from START and does not end at TERMINATE:

— Weighted Accuracy, $\text{score}_{\text{weighted}} = (1/N_{\text{rev}} \times |\hat{E}|) \sum_{i=1}^{N_{\text{rev}}} \sum_{j=1}^{|\hat{E}|} \text{label}_{i,j}$,
— Simple Majority, $\text{score}_{\text{simple majority}} = (1/|\hat{E}|) \sum_{j=1}^{|\hat{E}|} \text{Majority}\{\text{label}_{1,j}, \ldots, \text{label}_{N_{\text{rev}},j}\}$.

## 4.3. SENT2IMP: character impression extraction

While reviews contain story synopses, they also include readers' impressions about various characters. The expanded narrative network does not capture this information as the relationships represented on the graph are always between a pair of actants (characters or meta-characters). To extract this additional information, which has considerable importance in understanding how readers read, we developed SENT2IMP, an unsupervised algorithm that aggregates user opinions in descriptive phrases of review text into distinct groups of semantically similar impressions.

An *impression* is formally defined as a cluster of semantically similar phrases from reviews that imply a single aspect of a character in the eyes of the readers. We note that the relationship extraction pipeline used in this work captures not only the relationships *between* characters but also a set of relationships in which readers have expressed their impressions of characters. For example, while we obtain the classical relationships such as «Gandalf» 'chooses' «Bilbo» for our expanded story networks, we also obtain relationships such as «Bilbo» 'is' «unbelievably lucky». These latter relationships, when aggregated, filtered and clustered, bring to light the different reader sentiments associated with each character. Our pipeline thus provides an unsupervised approach to model a character as a *mixture* of such impressions.

To extract these impressions, we start by selecting the relationship tuples labelled as 'SVCop' from the full set of extracted relationships. 'SVCop' are those with the structure, Subject Phrase → Verb Phrase → Copula. These relationship tuples typically consist of a noun phrase and an associated adjective phrase that provide descriptive information about the noun phrase. We observe that a majority of these 'copular' phrase relationships contain information about character impressions. For example, in the reviews of *Animal Farm*, the sentence 'Snowball was humble and a good leader' yields the two SVCop relationships: (Snowball, was, humble), (Snowball, was, a good leader), where the phrases 'humble' and 'a good leader' describe Snowball. In the aggregate, these phrases contribute to *impressions* of Snowball being both humble and a good leader. In addition, we note that these relationships are frequent and comprise a significant portion of the extracted relationships per literary work: reviews contain a wide range of reader impressions when compared with the original work. Filtering and clustering these frequent relationships supports the creation of a *robust* pipeline for impression extraction.

Once we have extracted the SVCop relationships from the entire corpus, we group the relationships with respect to the character $e_i \in E$ present in the subject phrases of the extractions. Next, these phrases are transformed into vectors using a fine-tuned BERT [63] embedding. We embed these phrases in the BERT space to acquire a measure of semantic similarity. These vectors are then passed to the HDBSCAN clustering algorithm, which determines the optimal set of clusters, $C'_{e_i}$, per character $e_i$. This approach results in a qualitatively optimal clustering of phrases.

Owing to the noisiness of the reader review data, as well as the inherent non-Gaussian distribution of BERT embeddings, some of the resulting clusters are not homogeneous. To mitigate the noisy clusters, we employ a modified term frequency–inverse document frequency (TF-IDF) [66] scoring function to weight the words in each cluster. The score of a word in a cluster is equal to its frequency in the same cluster times the TF-IDF score in the review corpus for that word, where each review is a document and each

word is a term. This score for a distinct word $w_m$ with frequency $f_{m,\hat{C}_k}$ in a cluster of phrases $\hat{C}_k \in C'_{e_i}$, $|\hat{C}_k| = N$ is given as

$$x[m, \hat{C}_k] = \text{TF-IDF}(w_m) \times f_{m,\hat{C}_k}.$$

After removing the stop words, we observe that the meaningful clusters have a skewed-tailed distribution over these scores. In some extreme cases, where a cluster only contains a few words or phrases, the distribution is centred on a high score with low variance. For example, for the character 'Gandalf', we find numerous 'Gandalf' clusters, with one that contains only the word 'wizard'. We select a cluster to be meaningful based on the variety of its highly scored words. An ideal cluster consists of a handful of high scoring words. The fewer low scoring words a cluster contains, the higher its quality and, consequently, we expect an ideal cluster to have less noise.

To quantify this cluster quality measure, we calculate the skewness $g_1$ [67]:

$$g_1 = \frac{m_3}{m_2^{3/2}},$$

where

$$m_i = \frac{1}{N} \sum_{n=1}^{N} (x[n, \hat{C}_k] - \bar{x})^i$$

is the biased sample's $i$th central moment, and $\bar{x}$ is the sample mean [68].

For a cluster $\hat{C}_k$, the skewness of the distribution of $x[m, \hat{C}_k]$, determines the quality of the cluster. High positive skewness shows that the cluster consists mostly of high-scored words with a low number of infrequent words. Occasionally, when a cluster has a small number of low-scored words, such as the 'wizard' cluster for 'Gandalf', it loses its skewness. Instead, we observe a high word score $(x(\cdot, \cdot))$ average. We consider a cluster to be valid if it has skewness above a certain threshold or if its words' scores have a high average with low variance.

As a result of these selection, clustering and filtering tasks, we obtain a filtered mixture of clusters of reader impressions per character, $i$, which we label, $C_{e_i}$. Each cluster presents a unique description of a character within a novel. For example, 'George' in *Of Mice and Men* has various clusters associated with him, such as ['basically Lennie's protector', 'the guardian', 'in charge of Lennie', 'Lennie guardian'] and ['clumsy', 'unhappy', 'sad', 'nervous', 'very rude', 'selfish', 'painfully lonely'].

In order to quantify the geometry of the obtained mixture, we define a distance metric between every pair of clusters of numerical embeddings (see algorithm 2). The resultant measure is in the range [−2, 2] and is close to 2 if the pair is semantically similar and close to −2 if the pair of clusters are semantic opposites. We once again use BERT embeddings where semantic similarity is measured as the cosine distance between a pair of phrase embeddings. Before computing the cosine similarity, we reduce the dimension of the BERT embeddings to four principal components using principal component analysis (PCA), having found that the resultant scores generalize well with this choice of principal components. It follows that for a *mixture* of clusters, we can represent the obtained inter-cluster distance measure between $\hat{C}_i$ and $\hat{C}_j$ (within a mixture $C_{e_i}$) on a heatmap that is symmetric—because our distance measure is symmetric—(see algorithm 3) for a particular character. To ensure that the heatmaps we generate are rich, we limit our study to those characters with at least four clusters of descriptive phrases ($|C_{e_i}| \geq 4$).

### 4.3.1. Quantifying and visualizing the complexity of a character

The heatmap for a single character shows an empirical measure of *character complexity*. In this task, we run algorithm 3 such that both characters in the pairwise comparison are identical. A large and high-variance heatmap implies that readers consider the character to be *complex*. This complexity may (i) reflect disagreement in the reader discussions about that character, implying that impressions of the character are *controversial*, or (ii) capture contradictory and multi-modal character traits that authors develop—or that readers constitute through their reading—to portray remarkable characters in their novels. One finds instances of both in the impression clusters of Bilbo Baggins (table 2): The first two clusters that portray Bilbo as 'unpleasant/boring' and 'loveable/charismatic' are most likely instances of readers' contradictory perceptions; Clusters 2 and 3, however, portray Bilbo simultaneously as a hero and a thief, and are most likely diverse dimensions inherent in the novel and picked up on by the readers. In other cases, when the heatmap is smaller and of low-variance, readers' perceptions of

the profiled character are not as diverse, perhaps because of the character's secondary role in the novel and/or the character's narrow/singular purpose in the story. Such a character would be less *complex*.

This qualitative understanding of *complexity* can be quantitatively described by *entropy*. If one assumes that the heatmap entries of any character are samples from an underlying random variable, and the complexity of a character is its entropy, then the more spread out the underlying distribution is, the higher its entropy. Since we have only a few samples—coming only from the lower half of any given symmetric heatmap matrix—it is best to model the random variable as a discrete probability distribution in the range of scores in our heatmap, [−2, + 2].

First, we define the number of bins for the distribution, $b$. Then, from the impressions heatmap of a single character, we slot all the values below the major diagonal into these bins. Because most of our heatmaps do not have enough values to populate each bin, we adopt a standard numerical technique [69] of using a smoothing kernel—in our case uniform—(of width $w$ bins) across the bins. In general, $w$ and $b$ are hyperparameters that change the sensitivity of calculating the *entropy*.

More formally, for a heatmap $S \in \mathbb{R}^{N \times N}$ where $N$ is the number of impression clusters for the associated character, the entropy (complexity) $\mathcal{H}$ is defined as

$$\mathcal{H} = - \sum_{\{i \in [b]\}} P_i \log P_i,$$

where

$$S_{ij} \xrightarrow[\forall i > j]{\text{aggregate}} \text{histogram bins}_{[-2,2]}^b \xrightarrow{\text{smoothing kernel (w), norm}} \text{Prob. Distribution } P_i.$$

## 4.3.2. Distinct character comparison

The observation that some characters are, in the minds of the readers as reflected in their reviews, more *complex* than others, motivates our use of the distance measure employed in algorithm 2 for impression clusters derived from a pair of *distinct* characters. In this case, we project onto the heatmap the distance measures between clusters from two separate mixtures (the heatmap will not be symmetric and may not be square). Such a representation highlights the smaller number of *contexts* in which two characters are similar even across novels. For example, in *To Kill a Mockingbird*, one of Atticus's clusters consists of the phrases ['the father of kids', 'the father of protagonist', 'the father of Jem', 'lenient father', 'the father of character', 'a father figure'] and can be compared with the first example of 'George''s attribute of being a 'guardian'. Although these two sets of phrases are not exactly the same, one can still recognize that 'George' and Atticus' are perceived similarly in their shared roles within their respective works.

---

**Algorithm 2.** Computing the similarity score between a pair of clusters of numerical embeddings.

---

**Input:** $\hat{E}_m, \hat{E}_l$: Two Clusters of Numerical Embeddings
**Output:** $S_{l,m}$
 \*\*\*\*\*\*\*\*\*\* From $\hat{E}_l$ to $\hat{E}_m$ \*\*\*\*\*\*\*\*\*\*
 $s_1 = 0$
 **for** every embedding $u$ in $\hat{E}_l$ **do**
 $s_1 = s_1 + \max_{v \in \hat{E}_m} \cos(v, u)$
 **end for**
 $S_1 = \frac{s_1}{|\hat{E}_l|}$
 \*\*\*\*\*\*\*\*\*\* From $\hat{E}_m$ to $\hat{E}_l$ \*\*\*\*\*\*\*\*\*\*
 $s_2 = 0$
 **for** every embedding $v$ in $\hat{E}_m$ **do**
 $s_2 = s_2 + \max_{u \in \hat{E}_l} \cos(v, u)$
 **end for**
 $S_2 = \frac{s_2}{|\hat{E}_m|}$
 $S_{l,m} = S_1 + S_2$

**Algorithm 3.** Computing the heatmap between a pair of mixtures of phrase clusters.

**Input:** $C_{e_i}$, $C_{e_j}$ two mixtures of phrase clusters such that,

$\quad C_{e_i} = [\hat{C}_{1,e_i}, \hat{C}_{2,e_i}, \ldots, \hat{C}_{M,e_i}]$

$\quad C_{e_j} = [\hat{C}_{1,e_j}, \hat{C}_{2,e_j}, \ldots, \hat{C}_{N,e_j}]$

**Output:** $S \in \mathbb{R}^{M \times N}$: A heatmap $S$ with $S_{lm}$ equal to the similarity between the $l^{th}$ cluster in one mixture and the $m^{th}$ cluster in another mixture.

> **for** iter in $[1, \ldots, M]$ **do**
> $\quad E_{\text{iter},e_i} = \text{BERT}\,[\hat{C}_{\text{iter},e_i}]$
> $\quad \{\gg E_{\text{iter},e_i} \in \mathbb{R}^{768 \times |\hat{C}_{\text{iter},e_i}|}$: *our BERT embeddings are 768-dimensional vectors.*}
> **end for**
> **for** iter in $[1, \ldots, N]$ **do**
> $\quad E_{\text{iter},e_j} = \text{BERT}\,[\hat{C}_{\text{iter},e_j}]$
> **end for**
>
> $[\{\hat{E}_{1,e_i}, \hat{E}_{2,e_i}, \ldots, \hat{E}_{M,e_i}\}, \{\hat{E}_{1,e_j}, \hat{E}_{2,e_j}, \ldots, \hat{E}_{N,e_j}\}]$
> $\quad = \text{PCA}\,[E_{1,e_i}, E_{2,e_i}, \ldots, E_{M,e_i}, E_{1,e_j}, E_{2,e_j}, \ldots, E_{N,e_j}]$
> $\{\gg \hat{E}_{\text{iter},e_i} \in \mathbb{R}^{4 \times |\hat{C}_{\text{iter},e_i}|}$: *4 principal components.*}
>
> **for** iterM in $[1, \ldots, M]$ **do**
> $\quad$ **for** iterN in $[1, \ldots, N]$ **do**
> $\quad\quad$ Perform Algorithm 2 for the pair of clusters of numerical embeddings, $\{\hat{E}_{\text{iterM},e_i}, \hat{E}_{\text{iterN},e_j}\}$
> $\quad$ **end for**
> **end for**

# 5. Results and discussion

## 5.1. Story network creation and expansion

The resulting story network graphs for the five works are presented in figure 1, with a clear visual distinction between the diagetic nodes (i.e. the novel's characters) and the metadiscursive or extra-diegetic nodes (i.e. actants not in the novel *per se*).

These expanded graphs reveal interesting features not only about readers' perceptions of the stories, but also of how readers conceptualize authorship as well as other external features relevant to an understanding of the novel. For example, the authors of *Of Mice and Men* and *The Hobbit* are directly linked to main characters in those novels, whereas for the other novels, the authors are only connected to the main story graph through intermediary nodes. The close connection of both Tolkien and Steinbeck to the main story graphs possibly highlights the readers' perceptions of the author as equally important to any discussion of the novel for these two works. For *Frankenstein*, by way of contrast, the expanded graph captures the complex discussions of 'authoriality' that pervade both the narrative and meta-narrative space. For the other two novels in our corpus, the author appears to play a slightly more divorced role, at least in the reader discussions. While our data collection occurred prior to the release of Harper Lee's *Go Set a Watchman* [70], which may well have triggered greater awareness of Lee as an author, the reviews for *To Kill a Mockingbird* and *Animal Farm* may be capturing a reduced awareness of the authorships of Lee and Orwell, as opposed to the broadly recognized authorships of Tolkien and Steinbeck.

The extended networks include a proliferation of generic terms such as 'people' and 'story' possibly capturing readers' awareness of other readers—the generic 'people'—and the narrativity of the work itself. Other terms such as 'place' and 'home' may reflect readers' efforts to tie the novel into their own understanding of the world and, possibly, considerations of locality and the domestic. The 'ways' node appearing in most of the expanded story graphs accounts for the strategies that characters pursue in the target novel. For example, in *To Kill a Mockingbird*, 'Scout' is connected to the 'ways' node with relationships such as 'looked' and 'thought', reflecting readers' awareness of Scout's efforts to both comprehend and solve the fundamental challenges she encounters in the novel.

These metadiscursive nodes share an additional interesting feature: the edges connecting them to the character nodes in the main story graph are mostly in a single direction (either toward the story graph or

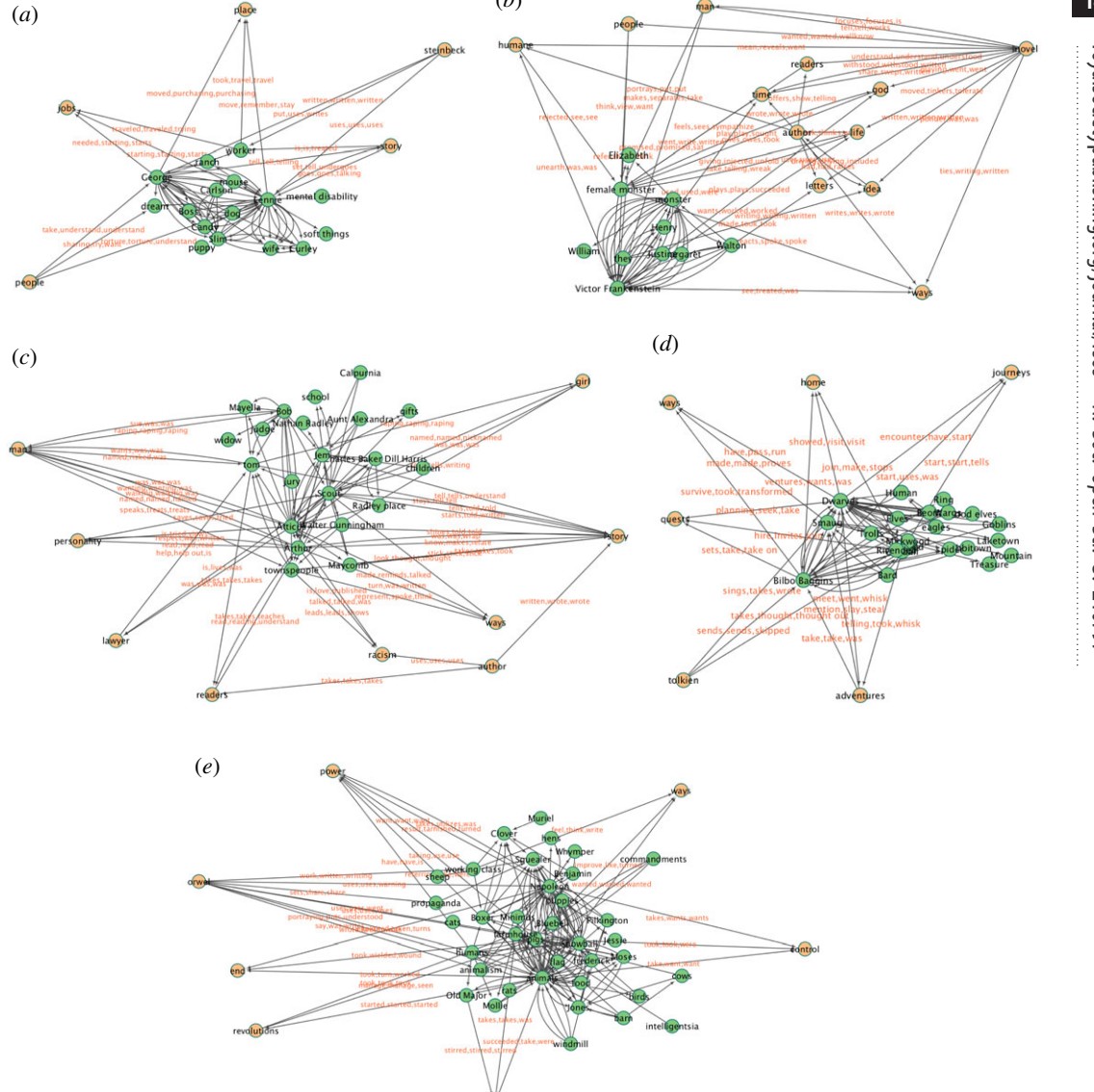

**Figure 1.** Expanded story network graphs. The expanded story networks for the five literary works—nodes that represent characters in the story are in green while the actants extending the original character story network are in orange. (*a*) *Of Mice and Men*: the node 'steinbeck' has an in-degree of 0 suggesting readers' understanding of the author's impact on creating complex story actors, while the actants have no meaningful return engagement. Similarly, the 'place' node cannot directly affect causal change in the story and as a result is very rarely found in the *subject* part of a relationship (the out-degree is 0). (*b*) *Frankenstein*: the subnetwork of 'letters', 'author' and 'novel' indicate that readers recognize the epistolary nature of *Frankenstein*. The common node 'people' (which is found in most of the graphs) represents the reviewers' perception of other reviewers. (*c*) *To Kill a Mockingbird*: important and intangible actants such as 'racism', 'lawyer', 'personality' compose the extended story network nodes in this graph. The 'personality' node reflects the novel's dedication to character development, be it of 'scout', 'atticus' or even 'arthur'. (*d*) *The Hobbit*: the readers' classification of the novel's genre is immediately apparent in the nodes 'adventures', 'quest', 'home' and 'journeys'. Inanimate actants such as 'home', 'journey', 'quest' and 'ways' typically have a very low out-degree (in this case 0) whereas 'tolkien' has a very low in-degree. The node 'ways' signals strategy: 'Dwarves' *have* 'ways' or 'Bilbo Baggins' *took* 'ways'. (*e*) *Animal Farm*: the nodes 'rebellion' and 'revolution', in conjunction with the nodes 'power' and 'control' highlight the sustained themes of power struggle, social dynamics and politics that lay at the ideological root of the novel. The author 'orwell' once again has a high out-degree and the node 'ways' once again signals strategy: 'hens' *think* of 'ways' and 'pigs' *wanted* 'ways'.

away from it) as in, for example, *Of Mice and Men* (figure 1*a*). By way of contrast, main story characters interact with each other generating a mixture of in- and out-edges. The *Animal Farm* graph presents an intriguing example of this directionality, with the 'Orwell' node having only outwardly directed edges, and the remaining additional metadiscursive nodes only having inwardly directed edges. There are two

exceptions in this graph to this general rule: the nodes 'revolution' and 'rebellion' share an outwardly directed edge connecting them to the diegetic node of the 'farmhouse'. Readers here collectively recognize not only the strategy of 'revolution' that animates the novel, but also the focus of the uprising on the locus of institutional authority, here the 'farmhouse'.

The expanded novel graph for *Frankenstein* reveals a large number of metadiscursive nodes that are not directly connected to the main story component, creating a secondary network of high-degree metadiscursive and extra-diegetic nodes, thereby capturing a broad reader conversation not centred on the novel itself. To highlight this aspect of the reader conversations, we extended the additional nodes by finding the interconnections between a pair of candidate mentions. This secondary network reveals a lively conversation not only about the story plot(s) but also about meta-narrative considerations such as the composition of the novel, its epistolary frame narrative, Mary Shelley's authorship, and philosophical speculation about 'God'.

Given the implications of the expanded narrative graphs, we would be remiss to dismiss *Goodreads* reviews as amateurish plot-focused summaries, since they capture more sophisticated speculations of a broad readership, reflecting a latent diversity of opinion that may well be an echo of Fish's communities of interpretation. Indeed, the methods presented here capture the voices of emerging literary critics and their engagement with the works of fictions and the other readers as they negotiate the boundaries of their interpretive communities. Consequently, we can see these graphs as capturing an emerging discussion of the complexity of a work of fiction, the relationship between authors and their works, the constitutive role that the acts of reading and reviewing play on the works in question, and the interpretive range of reader engagements that extends well beyond straightforward plot summary.

## 5.2. REV2SEQ: consensus event SEQuence networks from REViews

Despite the discovered complexities of these reviews, discussions of plot do loom large in these posts, even if these discussions are incomplete or jumbled. Despite the noise, which is probably the result of mis-remembering, forgetting or misreading, there are certain common features to these generated networks that, in the aggregate, reveal readers' complex understanding of the target novel's storylines. These features include: (i) The number of nodes is proportional to the number of distinct characters obtained from the EMG task considered jointly with the distinct relationship clusters between a pair of characters from the IARC task. Because of this correlation, *Animal Farm* and *The Hobbit* have more nodes than *Frankenstein*, *Of Mice and Men* or *To Kill a Mockingbird*, thereby revealing that reader reviews preserve this aspect of the narrative complexity of the target novel. (ii) A node that aggregates a key event in a narrative is found to have a higher degree. For example, in *The Hobbit*, «Bilbo» finds «ring» is a central event that has a high in-degree and out-degree. (iii) Contextual events, such as «Lennie» dreams of «ranch» in *Of Mice and Men*, that are true throughout the work but are inherently less subject to sequencing generally appear at the beginning of the sequence graphs, and help set the stage for the story. This effect is highlighted in *The Hobbit*, which is known for its detailed descriptions and intricate universe. Consequently, the event sequence estimated for *The Hobbit* has a large out-degree from the START node. (iv) Key finale events such as «George» killing «Lennie» (*Of Mice and Men*) or «Bard» killing «Smaug» (*The Hobbit*) are suitably positioned towards the end of the sequence within the estimated objective timeline from START to TERMINATE. (v) Stories with multiple parallel subplots (such as *The Hobbit*) are shown to have a larger branching factor than linear stories (such as *Frankenstein* and *To Kill a Mockingbird*). (vi) Disjoint paths from START to TERMINATE that do not interfere with the core network often describe supporting characters, the fringe events in which they place a role, and secondary storylines.

Each of these features provides us with specific information about how readers review (if not directly read) a novel, while also providing information on what individual readers see as important (or remember as important), and also the multiple perspectives on the novel generated by large numbers of readers, thus echoing the reader response literature on communities of interpretation. Indeed, the multiple paths through the graph echo the idea that there are many ways to read—and review—a novel.

For example, *The Hobbit* is characterized by a series of richly drawn characters and an intricate world in which the action unfolds. The plot of the novel is complex and multi-stranded, with multiple pivotal high-degree event nodes.

The emergent event sequence network for the novel has a high average branching factor, multiple high-degree nodes and parallel branches (figure 2). There are several relevant and verifiable event sub-sequences in this network including: (i) «gandalf» chooses «bilbo» →« bilbo» encounters «troll» →« bilbo» finds «ring» →« bard» kills «smaug»; and (ii) «bilbo» leaves «hobbiton» →« bilbo»

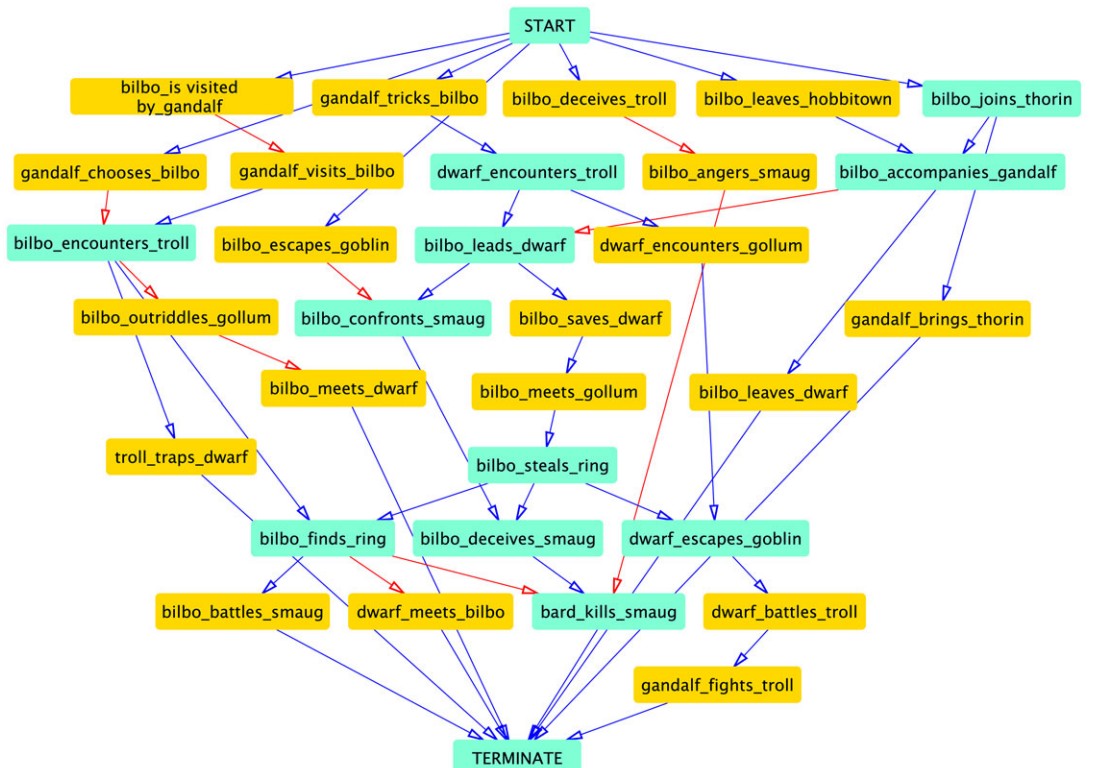

**Figure 2.** *The Hobbit*. The complete event sequence network—nodes with degree >2 are shaded turquoise and edges verifiable by at least two review samples are in red.

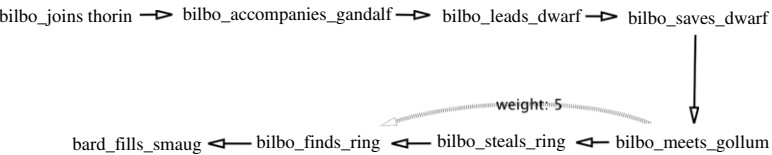

**Figure 3.** *The Hobbit*. Highest total node degree event sequence in the network; edges such as the shaded edge shown are removed during sequencing.

accompanies «gandalf» →« bilbo» leads «dwarf» →« bilbo» confronts «smaug» →« bilbo» deceives « smaug» →« bard» kills «smaug».

We empirically find that longer event sequences in this event sequence network (those that include many high-degree nodes) most often contain many critical details of the literary work's plot. For instance, the path of event nodes with the maximum total node degree from START to TERMINATE for *The Hobbit* is shown in figure 3. As such, there seems to be a broad reader consensus on which story points are 'important', while there are still many alternative pathways through the narrative thicket.

Events not explicitly connected in this chain (and in the network in general) may have precedence relationships between them. However, since our interest is to capture the most descriptive event sequence network, these shortcut edges are removed. A trivial example of this is the *absence* of edges from START to *every single* event. The resulting event sequence network can be verified against the precedence constraints determined by a majority of reviews. Some samples include:

— «gandalf» chooses «bilbo» →« bilbo» encounters «troll»: '[⋯] Gandalf chose this particular hobbit to round out the unlucky number of the dwarves [⋯] trolls encountered by Bilbo and co. [⋯]'
— «bilbo» deceives «smaug» →« bard» kills «smaug»: '[⋯] [Bilbo] deceives the dragon, Smaug. [⋯] the dragon was slain by a man of the lake [⋯]'

These examples support our initial hypothesis that reviewers implicitly encode the relative structure of time, the *fabula*, in their reviews; our algorithm extracts precedence information to yield insightful and true event sequences. The examples also highlight the complexity of using the reviews' tense and

other linguistic structures to sequence events: the first review is entirely written in the past tense (*chose, encountered*), while the second review mixes the past and present tenses (*deceives, was slain*).

The relationship extractor is intrinsically not programmed to extract relationships in a sequential manner. *The Hobbit* features 4 two-node cycles and 0 cycles of greater than 2 nodes. As mentioned in the methodology section, two-node cycles are resolved intuitively by considering the dominant (most frequent) direction of an edge between the two nodes. A closer look into the pair of event nodes comprising a few of these averted cycles provides additional insight into the complexity of the sequencing task and into the difficultly of ordering highly correlated and concurrent events: [«bilbo» *finds* « ring» AND « bilbo» *meets* « gollum»], [« dwarf» *battles* « troll» AND « gandalf» *fights* « troll»].

It might be useful to revisit the pipeline that is driving the aggregation of these event tuples. The *man of the lake* is a mention of «bard» as aggregated by the EMG task; similarly the *hobbit* is labelled «Bilbo». The HDBSCAN-driven IARC process clusters *slain* with «kills», which is the label of that particular directed relationship cluster between «bard» and «smaug». The relative positions of these events are also noteworthy: «bard» kills «smaug» appears toward the end of the story, «bilbo» accompanies «gandalf» is placed toward the beginning. A two-node detached event sequence in this network involves «Thorin», a supporting character, whose activities are largely ignored by reviewers in the aggregated reviews. As a result, the character's activities are not well incorporated into the core event sequence structure.

The sequence networks resulting from the reviews of the other four novels—*Of Mice and Men*, *Frankenstein*, *To Kill a Mockingbird* and *Animal Farm*—are provided in figure 4 and the interactive graphs are included in the data repository [59].

The accuracy of our sequencing networks with respect to human perception across the five novels is presented in table 1. The generally high accuracy (see Methodology) and tight error margins suggest not only that our networks consist of useful event sequences but also that our precedence edges are consistent across reviewers: the similarity of the two scores suggests that the event tuples are themselves interpretable, resulting in less confusion (there are few precedence relationships marked X). The higher error rate of *The Hobbit* and *Animal Farm* may be due to the style of each story; in *The Hobbit*, action scenes generally have many characters working together, in parallel or in the same setting, while in *Animal Farm*, there are many intentionally repetitive events that are difficult for the algorithm to resolve.

## 5.3. SENT2IMP: character impression

### 5.3.1. Single character impression

Our unsupervised method of character impression discovery provides insightful clusters about each character. A subset of these clusters for 'Bilbo' in *The Hobbit* is described in table 2. The first cluster provides a convincing argument that this character is *unpleasant*. The second one, in contrast, describes him as a hobbit of *impeccable personality*. These contradictory representations may capture a dichotomy of readers' impressions of Bilbo. The third and fourth clusters, however, are comparatively different, revealing disparate information about the character not related to sentiment at all, characterizing *Bilbo* as both a burglar and a hobbit, both of which are true. Such findings justify our assumption that SVCop relationships are worth consideration, as they not only capture the readers' broad range of perspectives on a character but also because these phrases form rich clusters of semantically aligned meanings that are not captured by standard supervised sentiment detection methods.

In addition to extracting rich clusters of actant-conditioned impressions, the HDBSCAN algorithm (with the default and constant distance threshold) clusters all *noisy* impressions into a separate cluster labelled '−1'. In this algorithm, we used a relatively high eps = 2 parameter to decrease the extreme sensitivity to noise. For example, for 'Bilbo', phrases such as ['the uncle of Frodo', 'unbelievably lucky', 'nostalgic'] are classified in the noise cluster. More examples can be found in the last row of table 2. Our results also show that there is a correlation between the perceived popularity of a character and the complexity of the impressions he or she elicits. These clusters of impressions can be informatively visualized with a dendrogram heatmap that sorts similar clusters with respect to correlation scores (see Methodology for how this score is computed). In order to find a label for a cluster, we pick the most frequent word in the cluster's phrase list excluding stop words.

A sample heatmap for 'Victor Frankenstein' is shown in figure 5. In this figure, there are three groups of impression clusters: (i) with labels such as, 'brilliant scientist', 'scientist of story', 'responsible' and

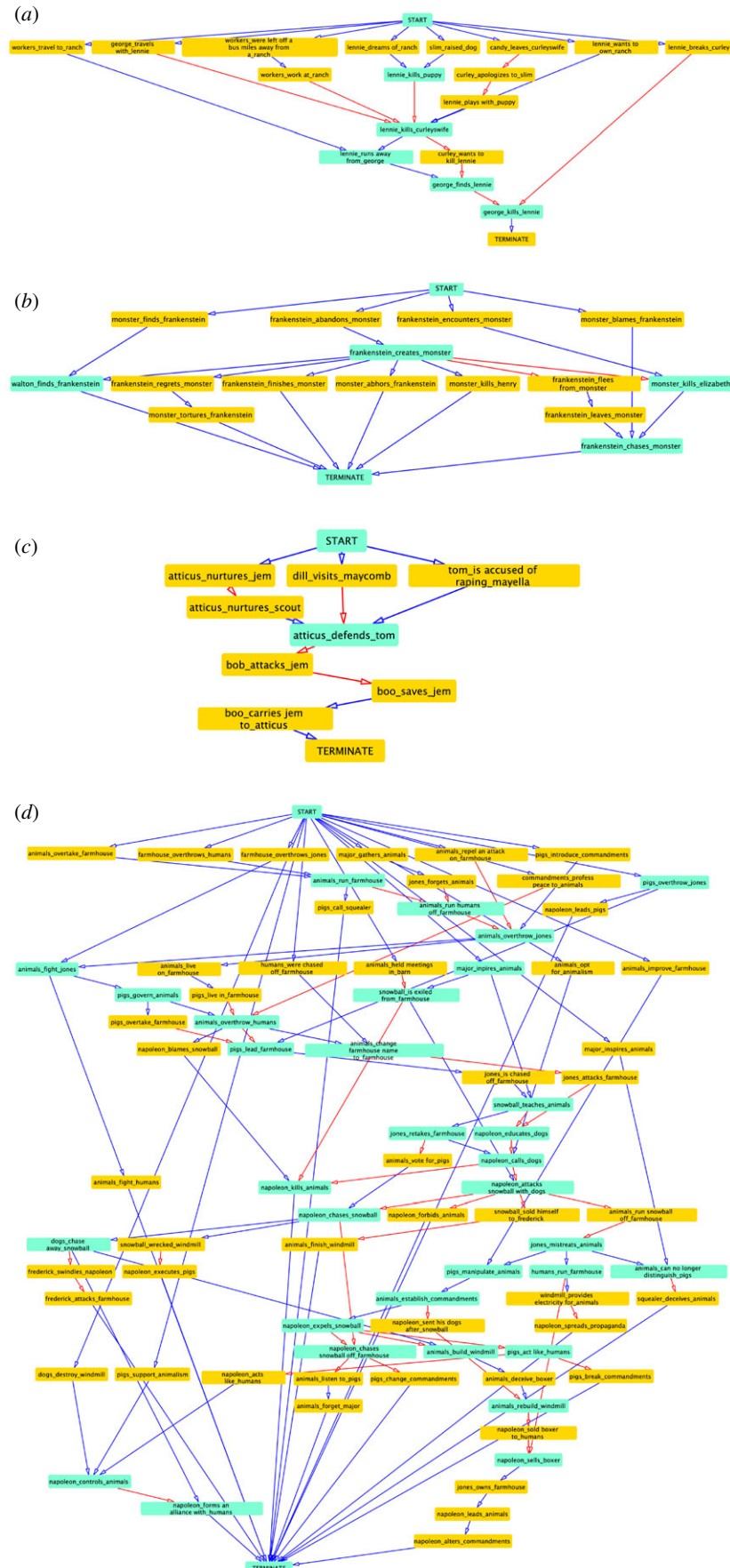

**Figure 4.** (*Caption overleaf.*)

**Figure 4.** (*Overleaf.*) The complete event sequence network of the *remaining* four novels. Nodes with degree greater than 2 are shaded turquoise and edges verifiable by at least two review samples are in red. The graphs can be further explored on our data repository [59]. (*a*) *Of Mice and Men*. Critical events such as «Lennie» killing «puppy», «Lennie» killing «Curley's wife» and «George» killing «Lennie» occur in succession in the story and this sequencing is reflected in the reviewers' posts. (*b*) *Frankenstein*. The network correctly captures the order of important events including: «Frankenstein» creates «monster» →« monster» kills «Elizabeth» →« Frankenstein» chases «monster». We also capture Frankenstein's regret upon his creating the monster. «Frankenstein» abandons «monster» occurs before his creating the monster, which is a false positive. (*c*) *To Kill a Mockingbird*. The event, «Atticus» nurtures «Jem» and «Scout», appears in the early part of the event sequence. The core sequencing structure is also verifiable: «Tom» is accused of raping «Mayella» → «Atticus» defends «Tom» →« Bob» attacks «Jem» →« Boo» saves «Jem» →« Boo» carries Jem to «Atticus». (*d*) *Animal Farm*. The story has several characters and routinely introduces events that have past precedent: the windmill has to be constructed twice; there are several uprisings on the farm; the commandments are changed multiple times. The character set promises to enlarge the event sequence network, while the similarities drawn between entities and their relationships in recurrent situations makes event sequence network generation a challenge. Nevertheless, we extract many useful sub-sequences including: «snowball» is exiled from «farmhouse» →« napoleon» attacks snowball with «pigs» →« napoleon» chases «snowball» →« napoleon» chases snowball off «farmhouse» →« pigs» change «commandments».

**Table 1.** Performance of the sequencing algorithm REV2SEQ on the five stories: the error margins were computed by estimating bounds for each score by replacing the labels marked *X* or *unsure* with all 0 s or all 1 s.

| story | score$_{weighted}$ (%) | score$_{simple\ majority}$ (%) |
|---|---|---|
| *Of Mice and Men* | 92.35 ± 5.29 | 94.12 ± 5.88 |
| *The Hobbit* | 75.75 ± 5.45 | 77.27 ± 1.51 |
| *Frankenstein* | 88.00 ± 4.00 | 90.00 ± 3.34 |
| *To Kill a Mockingbird* | 95.71 ± 4.28 | 100.00 ± 0.00 |
| *Animal Farm* | 78.57 ± 4.50 | 80.77 ± 2.74 |

**Table 2.** Example impression clusters for 'Bilbo' in *The Hobbit*: Clusters 1 and 2 describe impressions of 'Bilbo's character while clusters 3 and 4 describe his profession and community. Cluster marked −1 is noise. Labels for each cluster are aggregated based on the most frequent monograms per cluster.

| character | descriptors |
|---|---|
| Bilbo | The Hobbit |
| Cluster 1 | ['not the interesting character', 'timid not', 'not enthusiastic', 'reluctant', 'not the type of hero', 'less cute', 'not as cool', 'unsure of situation', 'a small unadventurous creature', 'Perhaps just not the kind of character', 'not as important', 'less cute'] |
| Cluster 2 | ['a true personality', 'an exemplary character', 'such a great character', 'resourceful', 'likable', 'still loveable', 'quite content', 'such a strong character', 'an amazing character', 'respectable', 'a great protagonist too', 'clever', 'such an amazing character', 'a peaceful', 'such an endearing character', 'a great choice', 'a fantastic lead character', 'quite engaging', 'cute', 'much charismatic character', 'such a fantastic Character', 'truly beautiful', 'enjoyable', 'just so charming', 'personable', 'able', 'the best character', 'quite skilled gets', 'awesome', 'smart'] |
| Cluster 3 | ['of course the burglar', 'a thief', 'a thief go', 'to a burglar', 'to a thief', 'to a thief', 'the burglar', 'their designated burglar', 'could a burglar', 'of course the burglar', 'a Burglar', 'a Burglar'] |
| Cluster 4 | ['a respectable hobbit', 'a respectable Hobbit', 'a sensible Hobbit', 'a clean well mannered hobbit', 'a respectable Hobbit', 'a sensible Hobbit', 'a proper hobbit'] |
| Cluster 5 | ['small', 'small', 'little', 'small', 'little'] |
| Cluster −1 | ['rich', 'the right man', 'a feisty character', 'the uncle of Frodo', 'unbelievably lucky', 'the perfect example of success', 'nostalgic', 'middle aged'] |

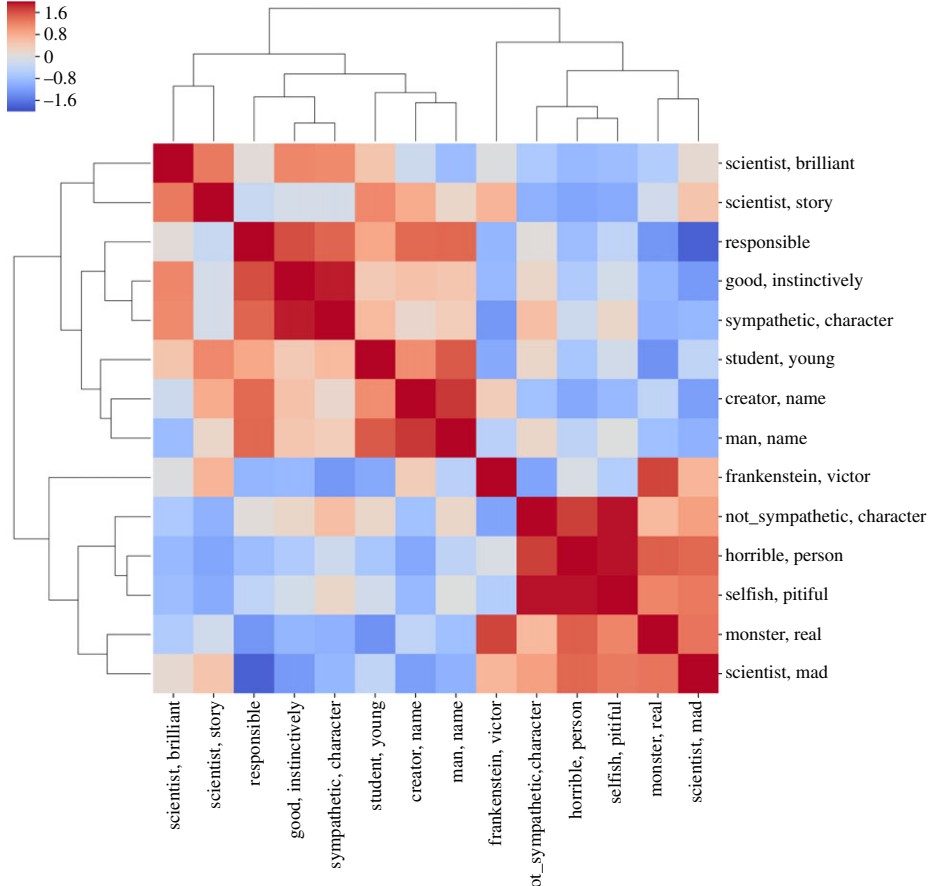

**Figure 5.** The (symmetric) heatmap for the character 'Victor Frankenstein'. The similarity scores between clusters of impressions labelled by the row/column headers are computed by algorithm 3. The sub-matrices that are deep red or blue imply a hierarchical structure to the mutual similarity or dissimilarity between groups of impression clusters. The diagonal entries are + 2 as a cluster of impressions is most similar to itself.

'instinctively good'; (ii) a group that includes 'young student', 'name of creator', 'name of man'; and finally, (iii) another group comprising, 'horrible person', 'selfish brat', 'real monster' and 'mad scientist'. On closer inspection, a cluster labelled 'Victor Frankenstein' stands out as a separate group with almost no correlation to other groups.

This representation also provides insight into the performance and limitations of BERT embeddings. Clusters that are similar should have a high similarity score (red) and clusters that are dissimilar should have a low similarity score (blue). For example, the clusters labelled 'selfish, pitiful' and 'horrible, person' have a high similarity score and the clusters labelled 'scientist, mad' and 'responsible' have a low similarity score. All the clusters are most similar to themselves so the major diagonal is deep red. There is a high similarity score between the clusters labelled 'sympathetic, character' and 'selfish, pitiful', due to the similarity between representative phrase {'a sympathetic character'} in the first cluster and the phrases {'miserable', 'sad', 'pitiful'} in the second cluster: after all, Frankenstein was a 'horrible person' for having created the monster but was also a person deserving of 'sympathy' and 'pity' for all the loss and grief that creation caused him.

### 5.3.2. Pairwise character impression comparison

Generally, in a literary work, each character plays various roles across a wide range of events in the *syuzhet* or storyline(s) of the novel. Characters most often also exhibit a diverse range of character traits. As such, each character is an individual, even if they share certain characteristics, or play similar roles, to other characters. For example, in *Animal Farm*, nearly all the characters are anthropomorphized animals, and live on a fictitious farm. Although there are multiple pigs in the story, each pig is distinctive from every other pig. In *To Kill a Mockingbird*, the characters are grounded in reality, sharing many recognizable characteristics (at least for American audiences) of small town

America, and the central crisis of the novel and the myriad reactions of the characters creates an empathetic potential for the reader. Yet each reader brings to their experience of the novel a set of external experiences and conditions. These experiences allow each reader—and each reviewer—an opportunity to augment the construction of storylines and characters in the novel. To avoid falling prey to the 'intentional fallacy' [71], where a critic tries to untangle the intentions of an author, the methods we devise here turn instead to an exploration of the constitutive nature of the reader reviews. Because each reviewer brings with them their own unique approach to reading, and given the wide range of characters and events in a novel, one might expect that these characters, especially mined from reviews, cannot be compared.

We find, however, that, while writing reviews, reviewers collate their character impressions into clusters of descriptors that are more semantically consistent *across characters* than the raw reviews would initially suggest. One possible reason for this finding could be a result of reviewers mapping their impressions into a shared consensus model of a character in an effort to write more convincing reviews and thereby receive more positive response from the broader community of reviewers. This could be based on reading other reviews of the same book, or reacting to comment threads on their own review or other reviews. Because of this semantic similarity in character descriptors, the impression clusters *enable inter-character comparison.*

The results of these inter-character comparisons may capture readers' broader understanding of fictional characters, and the process by which communities of interpretation emerge. The alignments of character impressions across multiple fictional works may in turn reflect the consistency of approaches to reading, so that the text is constituted in a complex manner across many readings.

To illustrate these intriguing areas of character overlap across different works of fiction, we produce heatmaps for pair-wise comparison of distinct characters, as in figure 6. Here we compare the impression clusters of 'Victor Frankenstein' from *Frankenstein* to those of 'Atticus Finch' from *To Kill a Mocking Bird*. The seemingly unlikely pair exhibit a surprising series of overlaps based on the readers' impressions of these characters. For example, the two have a high similarity score for clusters describing aspects of gender, responsibility and overall strength of character (as evidenced by the row/column labels in the figure).

One particularly interesting similarity is found in the clusters labelled 'father, kids' for 'Atticus' and 'creator, name' for 'Frankenstein'. This similarity reflects a twofold process: first, the recognition of the readers of the similarity in these roles and second, the worldview encoded into BERT embeddings. BERT embeddings, as seen in single character heatmaps, carry additional artefacts into the realm of cross-character evaluation. For example, the cluster labelled 'responsible' from 'Frankenstein' and the cluster labelled 'lawyer' from 'Atticus Finch' have a highly negative similarity score. This suggests either that the readers have a negative bias against lawyers, or that pretrained BERT embeddings are biased, or both: regardless of the source of this bias, the combined model integrating reviewer comments and the cosine-distance measure when applied to BERT embeddings seem to suggest that lawyers are *not* responsible.

Plotting the entropy of the single-character heatmaps can assist in the quantification of a character's perceived *complexity*. The resulting bar plot is presented in figure 7. Not surprisingly, the relative number of impression clusters empirically correlates to the entropy: reviewers describe a wider range of impressions for complex characters than for less complex ones. However, this feature alone cannot explain all the trends observed in the plot. For 'Jones', 'Napoleon' and 'Boxer' in *Animal Farm*, each character is associated with a roughly equal fraction of impression clusters, yet 'Napoleon' emerges as a more complex character in the readers' conceptualizations of *Animal Farm* characters; this is not surprising, as 'Napoleon' is the most enduring villain in the plot. It is also noteworthy that the three actors are ascribed by readers similar roles in the plot and this similarity extends partially to their complexity measure. In *To Kill a Mockingbird*, 'Atticus' is a central focus of the novel and it is his character that takes the spotlight as he defends 'Tom'. Indeed, 'Boo' and 'Scout' appear in the novel in many scenes to *support* 'Atticus'.

*Of Mice and Men* focuses on the dynamic between 'George' and 'Lennie', a pair of characters with notably different personalities, and the inherent complexity in their relationship. The resulting duality in character impressions, the limited number of additional characters in the novel, and a linear timeline results in a similar complexity profile for this pair of actants. The readers' impressions of characters that extend beyond the one-dimensional dismissal of inherently bad characters such as 'Curley' may motivate them to focus more intently on these two, plumbing the depths of their personalities and trying to understand their decisions in the context of the cruel economic environment of Depression-era America.

*The Hobbit* rigorously follows the genre conventions of fantasy action-adventure, with a fairly clear delineation of 'good guys' and 'bad guys'. As a result, 'Bilbo', the main protagonist, attracts the most

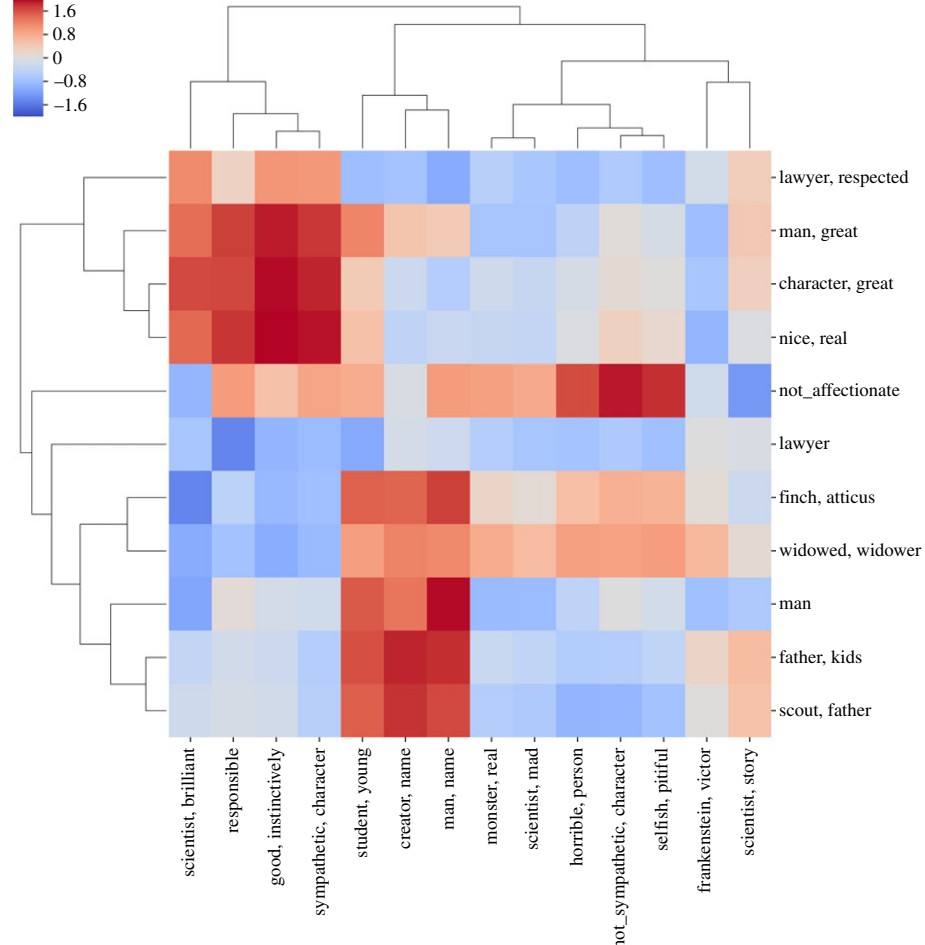

**Figure 6.** The (asymmetric) heatmap comparing the character 'Victor Frankenstein' from *Frankenstein* and 'Atticus Finch' from *To Kill a Mockingbird*. The similarity scores between clusters of impressions labelled by the row/column headers are computed by algorithm 3. The colour coding of impression clusters suggests valuable information stored in these representations about pairwise character similarity across novels, capturing the readers' process of aligning impressions from one novel to impressions created while reading another novel.

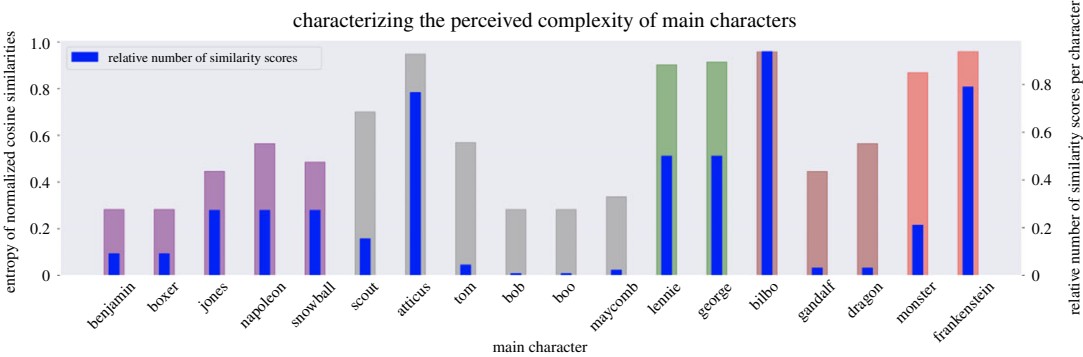

**Figure 7.** A measure of perceived *complexity* per character across novels. The colour blue corresponds to the relative number of empirical samples per character-specific heatmap used to compute entropy (prior to smoothing). Each translucent colour corresponds to a specific novel and plotted are the respective entropies of characters that have at least four impression clusters. We found $b = 50$, and $w = 3$ to be optimal hyperparameter choices to explore the differences in the complexity measure between characters.

attention in discussion forums, which, in turn, contributes to a greater perceived complexity. Indeed, a feature of these complexity measures is that complexity increases with the amount of attention an actant attracts from the reviewers. This aspect is not a failing, however, and captures instead a part of

the mental model that readers create as they read the novel and that they subsequently feel is important enough to share with other readers.

Last, in *Frankenstein*, while the 'Monster' wreaks havoc, it is in fact 'Frankenstein', the scientist, whose work raises ethical, moral and social concerns. Reader discussions about the character 'Frankenstein' and his complex positioning in the novel ultimately foster debate about the purpose of science and frequently consider whether the scientist 'Frankenstein' was perhaps the real monster. This effect is projected on both the relative number of impression clusters and the resulting complexity measure for the character in the reader reviews.

The consensus impression and narrative models created by our framework enable one to turn the spotlight back on individual readers. While the individual reviews collectively reflect and encode the whole, the whole in turn constructs a rubric to better understand the parts, i.e. how individuals both align with and differ from the collective. For example, one might ask, what makes a review informative and useful? We may not be able to identify the exact features that constitute a good review but, according to our model, expanded interpretations of the overall story space, rich event sequences that emphasize main characters, and unique impressions of these characters constitute important proxy targets. If a review makes use of at least some of these features, it will consist of enough information about the novel to be self-sustaining in the overall discussion space on the work without requiring access to that original work. Equally important is the presence of markers that indicate departures from the general consensus. Such departures not only set apart the individual review from the rest, but might be the seeds for the future emergence of new collective impressions in an evolving dialogue over novels and characters.

Consider, for example, an impression cluster for 'Atticus Finch' extracted by the SENT2IMP algorithm and labelled 'man, great' (figure 6). This cluster of impressions collected from all the reviews of the novel consists of the phrases: {'a good father', 'the loving father', 'the best dad', 'a man of integrity' ⋯ }. Similarly, there is another impression cluster labelled 'father, kids,' comprising the phrases {'the father of protagonist', 'the father of Jem', …}, and emphasizing his role as a father, while naming his children. Scout, the daughter and protagonist in the novel, has an impression cluster, labelled 'narrator, smart' comprising the phrases: {'really smart', 'very thoughtful, 'a smart girl', …}, bringing out a key attribute that has given the novel a lasting legacy. Tom Robinson, yet another pivotal character, has an impression cluster 'innocent, man' with the phrases: {'a mere poor victim of circumstances', 'innocent', 'a good black man', …}, and correctly portraying him as a victim of racial bias and violence.

In the light of the preceding collective impression clusters, let us consider the following review for *To Kill a Mockingbird*:

> **Review 1**: 'I think that To Kill A Mockingbird has such a prominent place in American culture because it is a naive, idealistic piece of writing in which naivete and idealism are ultimately rewarded. [...] Atticus is a good father, wise and patient; Tom Robinson is the innocent wronged; Boo is the kind eccentric; Jem is the little boy who grows up; Scout is the precocious, knowledgable child.'

The reviewer clearly aligns with and contributes to the majority views on the characters, Atticus Finch and Scout. The review also hints at an important event in the overall storyline, namely that 'Tom Robinson' is innocent (implicitly of a crime) but has been wronged, a situation captured clearly in our event sequencing graph (figure 4*c*). Reviews such as this one contribute significantly more information to the review ecosystem than a more cryptic review such as the following about *The Hobbit*:

> **Review 2**: 'Maybe one day soon I'll write a proper review of The Hobbit. In the meantime, I want to say this: If you are a child, you need to read this for Gollum's riddles. [...]'

This review is not only brief, but also skips references to a majority of the storylines, event sequences or character impressions. It does, however, emphasize the role of Gollum, a hugely popular character in the movie adaptation of *The Hobbit*, and the reader's evaluation of the suitability for children of the character's riddles. Our model thus provides an evidential measure, and one can objectively conclude—as admitted by the reviewer—that the review plays only a peripheral role in the overall review space, yet retains the potential to seed further discussions.

# 6. Limitations

Reviews in *Goodreads* are often short (within a post), noisy (with non-ASCII characters, emojis, incomplete sentences, slang expressions) and casual, marked by poor grammar and punctuation. In addition, not all of the reviews are reflective of a substantive engagement with the work in question. For example, at one point it became popular among high school teachers to ask students to post

reviews to *Goodreads* as homework and, because of the forced nature of this assignment, some of the reviews are less thoughtful than might otherwise be expected. Possibly because of this—and frequent teacher admonitions not to write 'plot summaries'—reviews that contain information about sequencing (which requires at least two eligible events occurring in succession to one another) are few. Furthermore, the poor punctuation in reviews, possibly reflective of the ways in which readers write their reviews, often results in poor NLP performance during different stages of the relationship extraction pipeline.

Other limitations are typical of most unsupervised pipelines. The story network and expansion method, for example, uses the EMG task to cluster entity mentions into character labels. This mapping may not be intuitive at times: for example, some reviewers use generic terms for characters such as 'the pig' or 'a guy'. These general terms may create confusion in some novels where they could describe multiple characters (e.g. in *Animal Farm* there are many characters that are pigs).

The story sequencing pipeline is sensitive to rare events, since these often do not have connections with more commonly aggregated events. Rare events, consequently, are susceptible to isolation from reliable trajectories in the sequence network. Our greedy algorithm places these events at the earliest possible timestamp and, without much interaction with the core network, these events quickly reach the TERMINATE state. For example, in *The Hobbit*, «Gandalf» brings «Thorin» is an event that could be integrated into the mainline event trajectories but, since reviewers are not inspired to talk about this event, the event largely disappears. At other times, reviewers retrospectively provide their opinions and highlight later events early in their reviews. As a result, these reviews contribute to the phenomenon of certain narrative-altering events appearing earlier than they should when compared with the ground truth. Cycles that are broken by the SBFS algorithm may also include edges that are important in sequencing. Breaking multi-event (greater than or equal to 3) cycles greedily does not impact our sequences significantly because this scenario is rare, as illustrated by the fraction of edges (weighted from matrix $M$) neglected due to multi-event cycle creations relative to the total weighted edges in the processed precedence matrix, $M_{ij}$: *Of Mice and Men*: 0.0%, *Frankenstein*: 0.0%, *The Hobbit*: 0.0%, *To Kill a Mockingbird*: 17.6%, *Animal Farm*: 6.80%.

There are also certain novel-specific limitations to this work. For example, the event sequence network for *Animal Farm* is subject as a whole to more wrongly ordered sub-event sequences than the other novels. *To Kill a Mockingbird* has a high fraction of edges that create cycles partially due to the very few eligible event sequence samples in reviews—the produced sequence network is sparse (figure 4). In addition, relationship extractors are poor at evaluating improperly punctuated, retrospective, qualified and/or otherwise syntactically sparse reviews resulting in noise in the aggregated relationships and derived trajectories. They are also poor at evaluating the order of relationships in sentences such as, 'Frankenstein creates a monster, abandons it'. In this example, the relationship extractor cannot, with a general rule, evaluate whether the abandonment occurs before or after the creation. Much, but not all, of this noise is removed through aggregation across reviews.

Finally, in the impression extraction algorithm, several characters do not have enough clusters representing the readers' different perceptions of that character. While this is, in part, a property of the reviews themselves—rare characters do not garner much attention—it limits our analysis of the *complexity* of characters to only popular ones: entropy can only be computed on those heatmaps that have representative samples across the range of similarity scores. Even with these scores, we employ a smoothing kernel, the characteristics of which, including width $w$ and bins $b$, change the absolute value of the entropy. Furthermore, the clustering of phrases into the 'noise' cluster is highly sensitive to the *core distance* parameter in density clustering. While the fine-tuned BERT embeddings employed in this work are trained to optimize similarity based on the cosine distance, we have seen that there are inherent biases—the cosine distance between 'lawyer' and 'responsible' is highly negative. The embeddings are also occasionally random—proper names such as 'Atticus' and 'Bilbo' do not have tuned representations in the BERT space—and this affects the quality of the impressions heatmaps.

# 7. Conclusion

Reader reviews, such as those from *Goodreads*, are not often considered in the context of literary analysis. We believe, however, that they provide an intriguing window into the broad cultural memories of 'what a book is about'. Sophisticated analyses of theme, or the deep anchoring of a literary work in a detailed intellectual, social and historical context, may at times elude the thousands of reviewers contributing individual reviews to these social reading sites. Yet, despite these failings, the reviews still capture the

meaningful thoughts of thousands of readers, each with their own diverse motivations for reading and reviewing, and are thus reflective of these readers' literary engagement [1,4,9]. Although they are usually unknown to each other, the readers of a particular work of fiction implicitly create an imagined community that shares, at least for some time, an interest in that work [5]. The evidence suggests that both individually and in the aggregate, these imagined communities of readers whittle down the novel to certain essential features: a stripped-down series of storylines represented as relatively short yet accurate pathways through a narrative framework graph, itself a distillation of the most important—in the view of the readers—characters and relationships in the novel. Similarly, complex, dynamic characters are conceptualized as a series of impressions that, despite their simplicity in an individual review, capture in the aggregate some of the complexity of character that lies at the heart of fiction writing and literary analysis.

Importantly, our approach allows us to preserve an awareness of the individual reader who carries with them their own compact representation of a complex work of fiction while also contributing to a collective, and often more complicated, overview of that work. Because our methods capture both how an individual reads and reviews, and how the broader community of readers of the same work read and review, it is possible to glimpse the relationship between a reader and the communities of interpretation that they are writing with, against and across. The numerous pathways through the narrative framework, in that sense, capture the multiple ways that people understand, remember and recount their own individual engagement with the work of fiction.

Although a frequent refrain of teachers of the literature is that amateur or otherwise 'uninformed' engagements with the literature are nothing more than 'plot summary', our exploration of Goodreads countermands this criticism. The reviews we considered, for example, encode far more information than simple plot summary. Inevitably, reviewers include their impressions of one or two characters as well as some small number of events meaningful to them in their understanding of the novel. Readers, of course, draw their impressions of characters in any work of fiction not only from that work itself, but also from all of their experiences of other characters and events, both real and fictional. Consequently, by considering these reviews in the aggregate, one can derive insight into readers' attempts to draw comparisons across novels, both on the basis of genre and story structure, and also on the level of character. As we show, readers' impressions of a character from one novel resonate with similar impressions of a character from another novel—even if those novels are as unalike as To Kill a Mockingbird and Frankenstein—thereby establishing a network of inference and allusion that resonates throughout the collective reservoir of reading. What we discover in these reader reviews, when taken collectively, echoes—in a data-driven manner—some of the fundamental literary critical ideas of the relationship between readers and texts.

In short, our methods allow one to explore the individual and collective reimaginings of a novel—the constitutive aspects of reader response that have been at the foundation of several strands of literary criticism from the early phenomenological reader response theories of Iser and others [20], through the explorations of communities of interpretation advocated by Fish [6], to concepts of intertextuality rooted in the work of Julia Kristeva [72] and its resonance in the work of Roland Barthes [73] among others. So, while individual reviews might not tell the whole story, and may on the individual level fail to capture the complexity of characters, the collective impressions of thousands of readers provide important insight into how people read, remember, retell and review. In so doing, these methods allow us to do many things, including reassemble a portrait of a tortured scientist and his monster.

Data accessibility. The data that support the findings of this study are available in the electronic supplementary material (published alongside the paper) as well as at https://osf.io/ym7aq./. These data were derived from the following resources available in the public domain: Goodreads.com [59].
Competing interests. We declare we have no competing interests.
Funding. No funding has been received for this article.

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
