## [Peer Review File · Royal Society Open Science]

Review History

RSOS-210797.R0 (Original submission)

Review form: Reviewer 1 (Federico Pianzola)

Is the manuscript scientifically sound in its present form?

No

Are the interpretations and conclusions justified by the results?

No

Is the language acceptable?

Yes

Do you have any ethical concerns with this paper?

No

Have you any concerns about statistical analyses in this paper?

No

Recommendation?

Major revision is needed (please make suggestions in comments)

Comments to the Author(s)

I liked reading this research and I think it has a great potential. However, there are some flaws that don't make it publishable in the present form. I've made annotations in the margins of the submitted manuscript (see Appendix A) and here I will summarise my concerns.

The authors ignore a large body of literature on the computational analysis of reader response, inaccurately framing their research as important because it fills a gap in the literature. I think it is important, but for the methodology applied.

The account of the advancements in literary studies is inadequate. The review is both too general and unnecessary for the purpose of the presented research. I suggest to start the "Related work" section drawing from the existing research about Goodreads and other social reading platforms. The ways in which they have been studied and the questions they helped to answer are an helpful starting point for the framing of the authors' work. The authors present an original contribution but it's framed inappropriately.

The authors rightly underline the gains that can be obtained from the large scale study of reader response, but this has been previously suggested in Finn, E. (2011). *The Social Lives of Books: Literary Networks in Contemporary American Fiction* [PhD]. Stanford University; and later in Reborá, S., & Pianzola, F. (2018). A New Research Programme for Reading Research: Analysing Comments in the Margins on Wattpad. *DigitCult - Scientific Journal on Digital Cultures*, 3(2), 19-36. <https://doi.org/10.4399/97888255181532>

In the introduction the rationale of the methods employed is only briefly sketched. This is a very important part to understand why the authors choose to implement the pipeline they describe. Moreover, this research is about real readers and what they think about books but the authors mostly describe their methodology in a technical way. It would be easier - and more meaningful for an audience of literary scholars, who may be very interested in this research - to have an explanation of the motivation for looking at certain features (e.g. character complexity) and for using the various algorithms employed. E.g. SENT2IMP could be presented as a way of finding the most common impressions about characters. This is only mentioned briefly in the introduction but it will help to remind this also when describing the method in detail. I leave to the authors and editors to decide how to improve this while keeping a good balance between the various sections of the paper. My suggestion is to simplify the Methodology section explaining a bit more in detail - but in a narrative way, like they already do in some paragraphs of the following subsections - the rationale of the pipeline, and moving the technical part in appendix.

I wonder whether simpler techniques would lead to different results. If the authors want to make claims in favour of their (computationally expensive) methodology, I think they should show how it performs in comparison to simpler methods. E.g. is a story network more informative than topic modeling? Is SENT2IMP more effective than measuring the semantic similarity of characters' names (and neighbour words) based on a vector space model of aggregated reviews (similar to what done by Jacobs, A. M. (2019). *Sentiment Analysis for Words and Fiction Characters From the Perspective of Computational (Neuro-)Poetics*. *Frontiers in Robotics and AI*, 6. <https://doi.org/10.3389/frobt.2019.00053>)?)?

Review form: Reviewer 2

Is the manuscript scientifically sound in its present form?

Yes

Are the interpretations and conclusions justified by the results?

Yes

Is the language acceptable?

Yes

Do you have any ethical concerns with this paper?

No

Have you any concerns about statistical analyses in this paper?

No

Recommendation?

Accept as is

Comments to the Author(s)

This is a comprehensive and detailed account of technically sound research that fits the criteria for publication in RSOS.

Indeed, guidelines for submission to RSOS state “the judgement as to importance and significance is left to the individual reader...” This very submission concerns the judgement of individual readers – readers of literature. The targets of the study are the reviews of five famous books by thousands of readers on a social reading site (Goodreads). Unlike RSOS reviewers (one hopes), these readers are not necessarily expert in the “intellectual, social and historical context” of the works they review. Although not a criterion for publication, the prevalence of inexpert views of a plethora of global issues adds topical relevance to this work. Although individual inexpert opinions may carry too much weight in issues of global import, a conclusion of this paper is that they are collectively more insightful (in this case of literature) than critics may have expected.

However, the main contribution of this paper is technical – to AI – the usage of NLP to tackle a challenging computational problem. It appears very competently executed.

What was not clear to me, however, is the repeated reference to an “imagined community” of readers. These readers appear not to directly interact with each other – only to engage in a common task (reading the same book). This is described as a “collective enterprise of literary analysis” but I don’t see how the task is “collective” when it is individual. I do not think that the sum of thousands of individuals constitutes a community – the latter is usually understood as an emergent entity – different than the sum of its parts. In mentioning this, I am not putting this study down in any way – it has value as a study of individual opinions not contaminated by interactions between them (an individual reviewer appears not to be strongly influenced by opinions of expert reviewers, for example). I am merely asking for a little more clarity here; do the reviewers interact or do they not? It seems to be explained that they don’t in the third paragraph of Section 1. But the next paragraph a “collective process” is referred to – and later “negotiated representation” where “readers collaboratively build a collective understanding of complex characters”.

Then we see that it is a “narrative network where the nodes are actants and the edges are interactant relationships.” The nodes here are characters in the story plus “the author,

filmatizations, film directors, actors and other extradiegetic features of the work.” I did not manage to understand from the introduction alone how the nodes are connected in the network. I would very much appreciate this to be elucidated – and clarification as to whether reviewers interact directly or not.

Section 2 on related work is useful but not quite complete and the references may need to be checked. Ref.[53] appears to be ascribed to multiple Joseph Campbells – but alas there was only one. I have not checked all references but I would ask the authors to do so – if there is a typo here there may be typos elsewhere.

I find the term “narrative network” confusing. This suggests to me it is a network that is contained in the narrative. The term has been used many times before as a synonym for character network (alliterations appeal) but here it refers to actants pertaining to the narrative – including characters but also writers and readers. For a good review of the reduced network (characters only), I’d like to point the authors to: V. Labatut, X. Bost, Extraction and analysis of fictional character networks: A survey. *ACM Comput. Surv.* 52, 89 (2019) and references therein.

Section 3 on data is good. It makes sense to limit this study to famous stories for which there are multitude of reviews.

Section 4 on methodology a sizeable part of the paper. The question of what constitutes an edge or link is answered in this section. However, this statement puzzles me: “Within this shared timeline, reviewers intersperse their opinions, argue over aspects of theme and plot, and offer other thoughts about style, imagery or comparisons to other works of fiction.” This gives me the impression the reviewers argue against each other- that they are in a dialogue. Again I remain unsure – are there interactions between reviewers or not (as in, do they actually influence each other). This is only a minor issue in presentation of the paper but it is important for (at least) my understanding of it for, if reviewers interact then the reviews are not independent and may be dominated by a small number of powerful opinions.

The methodology itself has to overcome some very challenging oddities of reviewers. But as stated the fact that “the output of the computational steps yielding aggregated sequencing results close to the ground truth validates the methodology. The methodology extends beyond what was published before and us useful for readers involve in NLP. As is usual in this discipline, the methodology is straightforward and the power of it lies in its ability to extract reasonable results from very noisy data.

Results are presented in Section 5 with one of the main conclusions being that inexpert reviews are not as meaningless as some teachers and experts might think. This is certainly interesting and gives hope to anyone who advocates “wisdom of the crowd” even if those crowds don’t talk to each other. But it also illustrates the individuality of readers as well as individuality of characters in the tales. These are presented in a series of plots which, while visualising what an expert reader is likely to know, are interesting to fans. The usage of entropy to capture the complexity of individual characters is also interesting and satisfactory that it captures the fact that “reviewers describe a wider range of impressions for complex characters than for less complex ones.” The untangling of the individual from the collective is nicely presented and appears methodologically very sound. And the hope that good reviewers and departures from the consensus might be automatically identified raises all sorts of ideas. Likewise, the sequence networks have the potential to aid bewildered readers (such as this one, oftentimes) through highly complex narratives.

The limitations of the study outlined in Section 6 are multifold but that is to be expected in any brave attempt, such as this one, to understand the masses’ reaction to literature. Many of these

limitations are set out in Sec.6 – a list that is interesting (as well as amusing, although also despairing) in its own right. It is amazing that anything coherent can come out of such noisy data. One of the main outcomes of this study is the power of statistical methods/NLP to extract anything from such data. The methods developed and demonstrated here may one day be applied in a beneficial way to multitudes of inexpert opinions on global issues that really matter.

On the general issue of the collective versus the individual, with interactions I point to M Krasnytska et al., Ising model with variable spin/agent strengths 2020 J. Phys. Complex. 1 035008 <https://doi.org/10.1088/2632-072X/abb654> which the authors of this submission may find relevant. The present RSOS submission improves on “previous narrative graphs [which] highlighted the interactions between actors, [but] they do not model the reviewers’ impressions of the actors” - something the Krasnytska et al paper does allow for (variable strengths of opinions). A fascinating future study might be how interactions between reviewers alter the phase diagram laid metaphorically out in this (RSOS) paper.

There are multiple other directions this (RSOS) paper can inspire. Reviews (such as this one) often tell us more about the reviewer than they do about the item reviewed. The tools developed herein may one day be tailored to compare individual reviewers to aggregated (independent) reviewers’ opinions. Such a feedback to reviewers themselves (how they digress from the mass opinion) may be helpful both in literature and global issues.

All in all, I have no doubts that this paper fits into the criteria of RSOS. In fact it does so exceptionally well. NLP knowledge, which is in relatively infancy in the community, is advanced here. But in addition to that, as an individual, I view this as a very interesting body of work, and I am happy to have had occasion to read it. I have no demands for improvements – just some light suggestions to enrich the references for the benefit of people who will want to know how this contribution fits in to a wider set of contemporaneous literature.

Decision letter (RSOS-210797.R0)

Dear Mr Holur

The Editors assigned to your paper RSOS-210797 "Modeling Social Readers: Novel Tools for Addressing Reception from Online Book Reviews" have now received comments from reviewers and would like you to revise the paper in accordance with the reviewer comments and any comments from the Editors. Please note this decision does not guarantee eventual acceptance.

We invite you to respond to the comments supplied below and revise your manuscript. Below the referees’ and Editors’ comments (where applicable) we provide additional requirements. Final acceptance of your manuscript is dependent on these requirements being met. We provide guidance below to help you prepare your revision.

Please submit your revised manuscript and required files (see below) no later than 21 days from today's (ie 11-Oct-2021) date. Note: the ScholarOne system will 'lock' if submission of the revision is attempted 21 or more days after the deadline. If you do not think you will be able to meet this deadline please contact the editorial office immediately.

on behalf of Marta Kwiatkowska (Subject Editor)
openscience@royalsociety.org

Associate Editor Comments to Author:

Comments to the Author:

Both reviewers offer a number of comments that will likely improve a paper that already appears to be on the right lines. Please consider carefully the comments made and how you might respond to them. We will look forward to receiving a revision in due course.

Reviewer comments to Author:

Reviewer: 1

Comments to the Author(s)

I liked reading this research and I think it has a great potential. However, there are some flaws that don't make it publishable in the present form. I've made annotations in the margins of the submitted manuscript and here I will summarise my concerns.

The authors ignore a large body of literature on the computational analysis of reader response, inaccurately framing their research as important because it fills a gap in the literature. I think it is important, but for the methodology applied.

The account of the advancements in literary studies is inadequate. The review is both too general and unnecessary for the purpose of the presented research. I suggest to start the "Related work" section drawing from the existing research about Goodreads and other social reading platforms. The ways in which they have been studied and the questions they helped to answer are an helpful starting point for the framing of the authors' work. The authors present an original contribution but it's framed inappropriately.

The authors rightly underline the gains that can be obtained from the large scale study of reader response, but this has been previously suggested in Finn, E. (2011). *The Social Lives of Books: Literary Networks in Contemporary American Fiction* [PhD]. Stanford University; and later in Reborá, S., & Pianzola, F. (2018). A New Research Programme for Reading Research: Analysing Comments in the Margins on Wattpad. *DigitCult - Scientific Journal on Digital Cultures*, 3(2), 19-36. <https://doi.org/10.4399/97888255181532>

In the introduction the rationale of the methods employed is only briefly sketched. This is a very important part to understand why the authors choose to implement the pipeline they describe. Moreover, this research is about real readers and what they think about books but the authors mostly describe their methodology in a technical way. It would be easier – and more meaningful for an audience of literary scholars, who may be very interested in this research – to have an explanation of the motivation for looking at certain features (e.g. character complexity) and for using the various algorithms employed. E.g. SENT2IMP could be presented as a way of finding the most common impressions about characters. This is only mentioned briefly in the introduction but it will help to remind this also when describing the method in detail. I leave to the authors and editors to decide how to improve this while keeping a good balance between the various sections of the paper. My suggestion is to simplify the Methodology section explaining a bit more in detail - but in a narrative way, like they already do in some paragraphs of the following subsections - the rationale of the pipeline, and moving the technical part in appendix.

I wonder whether simpler techniques would lead to different results. If the authors want to make claims in favour of their (computationally expensive) methodology, I think they should show how it performs in comparison to simpler methods. E.g. is a story network more informative than topic modeling? Is SENT2IMP more effective than measuring the semantic similarity of characters' names (and neighbour words) based on a vector space model of aggregated reviews (similar to what done by Jacobs, A. M. (2019). Sentiment Analysis for Words and Fiction Characters From the Perspective of Computational (Neuro-)Poetics. *Frontiers in Robotics and AI*, 6. <https://doi.org/10.3389/frobt.2019.00053>)?)

Reviewer: 2

Comments to the Author(s)

This is a comprehensive and detailed account of technically sound research that fits the criteria for publication in RSOS.

Indeed, guidelines for submission to RSOS state “the judgement as to importance and significance is left to the individual reader...” This very submission concerns the judgement of individual readers – readers of literature. The targets of the study are the reviews of five famous books by thousands of readers on a social reading site (Goodreads). Unlike RSOS reviewers (one hopes), these readers are not necessarily expert in the “intellectual, social and historical context” of the works they review. Although not a criterion for publication, the prevalence of inexperienced views of a plethora of global issues adds topical relevance to this work. Although individual inexperienced opinions may carry too much weight in issues of global import, a conclusion of this paper is that they are collectively more insightful (in this case of literature) than critics may have expected.

However, the main contribution of this paper is technical – to AI – the usage of NLP to tackle a challenging computational problem. It appears very competently executed.

What was not clear to me, however, is the repeated reference to an “imagined community” of readers. These readers appear not to directly interact with each other – only to engage in a common task (reading the same book). This is described as a “collective enterprise of literary analysis” but I don’t see how the task is “collective” when it is individual. I do not think that the sum of thousands of individuals constitutes a community – the latter is usually understood as an emergent entity – different than the sum of its parts. In mentioning this, I am not putting this study down in any way – it has value as a study of individual opinions not contaminated by interactions between them (an individual reviewer appears not to be strongly influenced by opinions of expert reviewers, for example). I am merely asking for a little more clarity here; do

the reviewers interact or do they not? It seems to be explained that they don't in the third paragraph of Section 1. But the next paragraph a "collective process" is referred to - and later "negotiated representation" where "readers collaboratively build a collective understanding of complex characters".

Then we see that it is a "narrative network where the nodes are actants and the edges are interactant relationships." The nodes here are characters in the story plus "the author, filmatizations, film directors, actors and other extradiegetic features of the work." I did not manage to understand from the introduction alone how the nodes are connected in the network. I would very much appreciate this to be elucidated - and clarification as to whether reviewers interact directly or not.

Section 2 on related work is useful but not quite complete and the references may need to be checked. Ref.[53] appears to be ascribed to multiple Joseph Campbells - but alas there was only one. I have not checked all references but I would ask the authors to do so - if there is a typo here there may be typos elsewhere.

I find the term "narrative network" confusing. This suggests to me it is a network that is contained in the narrative. The term has been used many times before as a synonym for character network (alliterations appeal) but here it refers to actants pertaining to the narrative - including characters but also writers and readers. For a good review of the reduced network (characters only), I'd like to point the authors to: V. Labatut, X. Bost, Extraction and analysis of fictional character networks: A survey. *ACM Comput. Surv.* 52, 89 (2019) and references therein.

Section 3 on data is good. It makes sense to limit this study to famous stories for which there are multitude of reviews.

Section 4 on methodology a sizeable part of the paper. The question of what constitutes an edge or link is answered in this section. However, this statement puzzles me: "Within this shared timeline, reviewers intersperse their opinions, argue over aspects of theme and plot, and offer other thoughts about style, imagery or comparisons to other works of fiction." This gives me the impression the reviewers argue against each other- that they are in a dialogue. Again I remain unsure - are there interactions between reviewers or not (as in, do they actually influence each other). This is only a minor issue in presentation of the paper but it is important for (at least) my understanding of it for, if reviewers interact then the reviews are not independent and may be dominated by a small number of powerful opinions.

The methodology itself has to overcome some very challenging oddities of reviewers. But as stated the fact that "the output of the computational steps yielding aggregated sequencing results close to the ground truth validates the methodology. The methodology extends beyond what was published before and us useful for readers involve in NLP. As is usual in this discipline, the methodology is straightforward and the power of it lies in its ability to extract reasonable results from very noisy data.

Results are presented in Section 5 with one of the main conclusions being that inexpert reviews are not as meaningless as some teachers and experts might think. This is certainly interesting and gives hope to anyone who advocates "wisdom of the crowd" even if those crowds don't talk to each other. But it also illustrates the individuality of readers as well as individuality of characters in the tales. These are presented in a series of plots which, while visualising what an expert reader is likely to know, are interesting to fans. The usage of entropy to capture the complexity of individual characters is also interesting and satisfactory that it captures the fact that "reviewers describe a wider range of impressions for complex characters than for less complex ones." The untangling of the individual from the collective is nicely presented and appears methodologically

very sound. And the hope that good reviewers and departures from the consensus might be automatically identified raises all sorts of ideas.

Likewise, the sequence networks have the potential to aid bewildered readers (such as this one, oftentimes) through highly complex narratives.

The limitations of the study outlined in Section 6 are multifold but that is to be expected in any brave attempt, such as this one, to understand the masses' reaction to literature. Many of these limitations are set out in Sec.6 – a list that is interesting (as well as amusing, although also despairing) in its own right. It is amazing that anything coherent can come out of such noisy data. One of the main outcomes of this study is the power of statistical methods/NLP to extract anything from such data. The methods developed and demonstrated here may one day be applied in a beneficial way to multitudes of inexpert opinions on global issues that really matter.

On the general issue of the collective versus the individual, with interactions I point to M Krasnytska et al., Ising model with variable spin/agent strengths 2020 J. Phys. Complex. 1 035008 <https://doi.org/10.1088/2632-072X/abb654> which the authors of this submission may find relevant. The present RSOS submission improves on “previous narrative graphs [which] highlighted the interactions between actors, [but] they do not model the reviewers' impressions of the actors” - something the Krasnytska et al paper does allow for (variable strengths of opinions). A fascinating future study might be how interactions between reviewers alter the phase diagram laid metaphorically out in this (RSOS) paper.

There are multiple other directions this (RSOS) paper can inspire. Reviews (such as this one) often tell us more about the reviewer than they do about the item reviewed. The tools developed herein may one day be tailored to compare individual reviewers to aggregated (independent) reviewers' opinions. Such a feedback to reviewers themselves (how they digress from the mass opinion) may be helpful both in literature and global issues.

All in all, I have no doubts that this paper fits into the criteria of RSOS. In fact it does so exceptionally well. NLP knowledge, which is in relatively infancy in the community, is advanced here. But in addition to that, as an individual, I view this as a very interesting body of work, and I am happy to have had occasion to read it. I have no demands for improvements – just some light suggestions to enrich the references for the benefit of people who will want to know how this contribution fits in to a wider set of contemporaneous literature.

===PREPARING YOUR MANUSCRIPT===

===PREPARING YOUR REVISION IN SCHOLARONE===

<https://royalsociety.org/journals/authors/author-guidelines/#data>. You should ensure that

you cite the dataset in your reference list. If you have deposited data etc in the Dryad repository, please include both the 'For publication' link and 'For review' link at this stage.

Author's Response to Decision Letter for (RSOS-210797.R0)

See Appendix B.

RSOS-210797.R1 (Revision)

Review form: Reviewer 1

Is the manuscript scientifically sound in its present form?

Yes

Are the interpretations and conclusions justified by the results?

Yes

Is the language acceptable?

Yes

Do you have any ethical concerns with this paper?

No

Have you any concerns about statistical analyses in this paper?

No

Recommendation?

Accept with minor revision (please list in comments)

Comments to the Author(s)

The revision improved the readability of the article and addressed the gaps in the literature review. I still have doubt about the communicative efficacy of the Methodology section but I leave this decision to the editors.

Review form: Reviewer 2

Is the manuscript scientifically sound in its present form?

Yes

Are the interpretations and conclusions justified by the results?

Yes

Is the language acceptable?

Yes

Do you have any ethical concerns with this paper?

No

Have you any concerns about statistical analyses in this paper?

No

Recommendation?

Accept as is

Comments to the Author(s)

Thanks for your nice paper.

Decision letter (RSOS-210797.R1)

Dear Mr Holur,

It is a pleasure to accept your manuscript entitled "Modeling Social Readers: Novel Tools for Addressing Reception from Online Book Reviews" in its current form for publication in Royal Society Open Science. The comments of the reviewer(s) who reviewed your manuscript are included at the foot of this letter.

The proof of your paper will be available for review using the Royal Society online proofing system and you will receive details of how to access this in the near future from our production office (openscience_proofs@royalsociety.org). We aim to maintain rapid times to publication after acceptance of your manuscript and we would ask you to please contact both the production office and editorial office if you are likely to be away from e-mail contact to minimise delays to

publication. If you are going to be away, please nominate a co-author (if available) to manage the proofing process, and ensure they are copied into your email to the journal.

on behalf of Prof Marta Kwiatkowska (Subject Editor)
openscience@royalsociety.org

Associate Editor Comments to Author:

While one of the reviewers continues to question aspects of the methodology, one of the operating principles of RSOS is to allow scientifically sound papers to be made available to the reading public, and let posterity determine the value of an approach, rather than withhold subjective judgements at the reviewer/editor level. Given this, we would recommend acceptance - and if readers do wish to offer commentary on the paper, a number of methods are available for this to take place.

Reviewer comments to Author:

Reviewer: 2
Comments to the Author(s)
Thanks for your nice paper.

Reviewer: 1

Comments to the Author(s)
The revision improved the readability of the article and addressed the gaps in the literature review. I still have doubt about the communicative efficacy of the Methodology section but I leave this decision to the editors.

Appendix A

ROYAL SOCIETY OPEN SCIENCE

rsos.royalsocietypublishing.org

Article submitted to journal

Subject Areas:

mathematical modelling,
computational social science,
behaviour, pattern recognition,
statistics, theory of computing

Keywords:

natural language processing,
narrative theory

Author for correspondence:

Pavan Holur

e-mail: pholur@ucla.edu

THE ROYAL SOCIETY
PUBLISHING

Modeling Social Readers: Novel Tools for Addressing Reception from Online Book Reviews

Pavan Holur^{1,*}, Shadi Shahsavari^{1,*},

Ehsan Ebrahimzadeh¹, Timothy R.

Tangherlini^{2,*} and Vwani Roychowdhury^{1,*}

¹in order pholur,shadihpp,eebrahim,vwani@ucla.edu

²tango@berkeley.edu

* Equal contribution

Readers' responses to literature have received **scant attention** in computational literary studies. The rise of social media offers an opportunity to capture a segment of these responses while data-driven analysis of these responses can provide new critical insight into how people "read". Posts discussing an individual book on *Goodreads*, a social media platform that hosts user discussions of popular literature, are referred to as "reviews", and consist of plot summaries, opinions, quotes, or some mixture of these. Since these reviews are written by readers, computationally modeling them allows one to discover the overall non-professional discussion space about a work, including an aggregated summary of the work's plot, an implicit ranking of the importance of events, and the readers' impressions of main characters. We develop a pipeline of interlocking computational tools to extract a representation of this reader-generated shared narrative model. Using a corpus of reviews of five popular novels, we discover the readers' distillation of the main storylines in a novel, their understanding of the relative importance of characters, as well as the readers' varying impressions of these characters. In so doing, we make three important contributions to the study of infinite-vocabulary networks: (i) an automatically derived narrative network that includes meta-actants; (ii) a new sequencing algorithm, REV2SEQ, that generates a consensus sequence of events based on partial trajectories aggregated from the reviews; and (iii) a new "impressions" algorithm, SENT2IMP, that provides finer, non-trivial and multi-modal insight into readers' opinions of characters.

there's 10 years of
research on the
topic

© 2014 The Authors. Published by the Royal Society under the terms of the Creative Commons Attribution License <http://creativecommons.org/licenses/by/4.0/>, which permits unrestricted use, provided the original author and source are credited.

1. Introduction

Online reader comments about works of literary fiction offer an intriguing window onto how people read. Although largely ignored in the realm of computational literary studies, these comments provide useful insight into how readers imagine the main story lines of a novel, how they understand the fictional struggles of characters, and how they develop varying impressions of the work. Taken together, the reviews of a single novel provide a view onto the collective imagining of what is important in the novel, including aspects of plot, the interactions between various characters, and even the metadiscursive space of authors, critics, film adaptations, and movie stars. These reviews thus provide impetus for a data-driven analysis of readers' responses to a work of literary fiction. They also help us understand how readers create an "imagined community" of readers engaged in the collective enterprise of literary analysis [1,2].

"Reader response theory" experienced a brief and productive heyday in literary theory during the 1960s. Despite the considerable attention this theoretical premise received, the focus of much of this work centered on the hypothetical and highly theorized "individual reader". This theoretical orientation was expanded to include groups of readers, and led in part to Stanley Fish's important contributions concerning "communities of interpretation"—groups of readers who, through their shared experiences, converged on similar readings of texts [2]. The consideration of broad-scale responses of readers to works of fiction, however, remains understudied, not because of a lack of interest on the part of literary historians and theorists, but because of a lack of access to those readers' responses. While there is considerable investigation into how groups of individuals are likely to read (or to have read), there is little work on how large groups of people respond to the same work of fiction.

The advent of social reading sites on the internet has enabled a revisiting of fundamental questions of how people respond to literary fiction. Perhaps best known among these sites in the United States is *Goodreads* that, along with sites like it, represent an online attempt to reproduce the face-to-face space of book clubs and library groups, where there is no "right" answer to reading the work (as there might be, at least implicitly, in a classroom), nor any hierarchy of critical insight (as there might be in a forum where professional reviewers or literary critics might dominate the conversation). Because the reader responses are archived, these sites offer an opportunity to explore computationally how people respond to individual works of fiction. Since these reviews are unguided explorations of fiction, and since many of the readers read and review purely for entertainment, it is unlikely that the readings encode the types of literary-theoretical engagement found in academic work. Instead, these reviews encode a popular engagement with literature, focusing on aspects of plot, character, and the struggles of these characters in their fictional worlds. The goal of our work is to model the collective expressions of thousands of readers as they review the same work of fiction. Can we discover what they see as important? Can we discover divergences in their readings? Do their reviews provide us with information about reading, remembering, and retelling? And do they tell us anything about the process of writing a review itself?

In our work, reader reviews, in the aggregate, constitute a collective process that converges on an underlying narrative framework. This framework, based on a relaxation of the narrative theories of Algirdas Greimas [3–5], is represented as a narrative network where the nodes are actants and the edges are interactant relationships. The actants in the network are expanded beyond the canonical census of a novel's characters to include the metadiscursive space, populated by actants such as the author, filmatizations, film directors, actors and other extra-diegetic features of the work. Importantly, book reviews also encode the varying impressions that readers form of the novel's characters. By reading reviews at "internet scale", one can consider the extent to which partial reviews contribute to a negotiated representation of the character-interaction network of the target novel, how readers' partial comments on the sequence of events in a novel can be aggregated to produce a series of sequential directed acyclic paths through that novel (i.e. different interpretations of the story-line or story-lines present in the work) and

how readers collaboratively build a collective understanding of complex characters, even if their individual views of the characters may not capture that same complexity. These problems can be seen as part of a formal, computational assessment of readers' response to even quite long and complex novels, such as *The Hobbit* or *To Kill a Mockingbird*, to name but two of our target works.

To address these challenges, we develop a pipeline of interlocking computational tools to extract and aggregate a meaningful representation of the reader-generated shared narrative framework modeled as a network. This framework is based on a structured open-world infinite-vocabulary network of interconnected actants and their relationships. In this work, we add two new algorithms to the pipeline, which was originally developed to determine the underlying generative narrative framework for social media conversations [6–8]. The first algorithm, REV2SEQ, determines the readers' collective reconstruction of the plot of the target work by sequencing the interactant relationships and representing these as pathways through the overall narrative framework graph. The second algorithm, SENT2IMP, presents a representation of the collective, at times differing, opinions of characters in a novel. In addition, we expand the narrative framework graph to include the important metadiscursive and extra-diegetic nodes noted above.

about characters

2. Related Work

Work on reader response theory has, in general, focused on the responses of individual readers to the act of reading [9,10]. Theoretical work in this area supported the development of the idea that the process of reading is constitutive of the text itself [11]. In this understanding of the relationship between reader and text, it is the reader and the individual reading of the text, more so than the author, that takes a position of primacy. The study of reader response has had a considerable impact on a broad range of fields outside of literary studies, including cognitive psychology [12], and phenomenology [13,14]. The phenomenology of reading extends to explorations of how different readers can exhibit a wide range of responses to the same piece of fiction [15]. One of the foremost developments in reader response theory is the concept of interpretive communities, anchored in the work of Fish [2]. He suggests, in part, that the interpretation of a text is culturally anchored, non-deterministic, and mutable, yet not infinite in its scope. Rather, each reader reads a work from this position as a member of a community of readers, proffering limits on how a work can be read. Consequently, *Goodreads*, and sites like it, may capture the collective readings of members of various interpretive communities and the boundaries of those possible readings.

Taken as a whole, these lines of inquiry allow one to explore how nonspecialist readers experience a piece of literature. Because of the difficulty of capturing records of nonspecialists' experiences of a particular work of fiction, a great deal of previous work has been done in controlled classroom or laboratory environments, with limited numbers of readers and relatively short texts [16]. By way of contrast, exploring the responses of thousands of readers to the same work of fiction may come close to a natural experiment, where the readers themselves choose what they share about their response to the work. In so doing, the readers may also be providing insight into the formation of interpretive communities, while simultaneously offering clues as to what they value in works of fiction, how they understand the complexities of characters, and how they remember and retell the events and the sequence of those events that they feel are most important to include in their review. To capture this broad-scale reader response, it is necessary to work computationally.

The computational analysis of narrative and literary fiction has a long history at the intersection of literary studies and artificial intelligence [17]. Starting in the early 1900s, the relationship between the underlying narrative and its realization in expression, be it oral or written, fueled the development of formalist and structuralist theories. The important distinction between *fabula*, the chronological order of events for a narrative, and *syuzhet*, the ordering of those events in the work of fiction, became fundamental concepts in the study of narrative [18,19]. In ensuing years, there was considerable interest in the relationship between narrative structures and memory [20], with emphasis on the social nature of storytelling [21]. The computational

Reader response theory has developed far beyond Fish and Iser, expanding into reception studies and audience studies. Moreover, for the topic addressed here, research about online discursive practices and social media behavior is relevant. E.g. the concept of "interpretive community" has been widely criticised and complemented by concepts such as "affinity space", "contact zone", "community of practice", etc.

possibilities of morphological approaches to narrative such as Propp's study of the Russian fairy tale [22] and other structural theorists were widely recognized as opportunities for devising computational approaches to the study of narrative [23,24]. The intersection of reader response theory and artificial intelligence has also been explored in the context of understanding character complexity and narrative unpredictability [25], echoing ideas in early artificial intelligence research [26].

Social reading platforms, such as *Goodreads* offer an opportunity to explore various aspects of reader response outside of the laboratory or classroom setting [27–29]. Work on these platforms has focused on an ethnography of online reading [30], considerations of reader review sentiment [31], how gender and intimacy are intertwined in reviews [32] and broader linguistic characteristics of the reviews [33]. Other studies have focused on *Goodreads* in the context of socially networked reading [34], the moderation of reader responses [35], recommender systems and participants' motivations for reviewing [36], and prediction of the success of a book given certain features in the reviews [37]. There has also been recognition of the disconnect between easy to gather metrics such as positive ratings and an understanding of how people actually read or remember a book [38]. We are not aware of any other work that holistically presents the impressions of members of various communities of interpretation concerning a novel's *fabula* or their varying opinions of the main characters of those works.

There are several computational approaches to modeling stories and event sequences [39,40]. These approaches use labeled original story data via numerical embeddings to train a supervised event sequence generator. The generated embeddings are then used as input to Recurrent Neural Networks (RNN) for modeling *sequences* of events and predicting the probability that one event succeeds another. Our research differs in several key ways: (i) While these models are largely supervised, our pipeline estimates sequences in an unsupervised way: reviews are not inherently sequenced relative to the events in a story; (ii) While these works use stories directly for training, we use *reader reviews* of a literary work rather than the work itself. This secondary data lends our analysis to reception theory in addition to standard knowledge extraction; and (iii) While we do use pretrained embeddings – in this case BERT – we present an *explainable* algorithm to extract story sequences comparable to the ground truth, in addition to summarizing other facets of a story's reception.

Additional work on evaluating the context of reviews focuses largely on high-level sentiment analysis and/or fixed-vocabulary opinion mining and classification tasks [41,42]. In these works, the classes of sentiments or opinions are fixed, and the object of classification is the readers' responses to the work of fiction as a whole. Our approach, by way of contrast, delves deeper into sentiment and genre at the phrase-level and maintains a robust transparency into the raw data throughout the pipeline. In addition, our analysis of reader response goes beyond the analysis of the work as a whole, and focuses on readers' discussions of actants associated with the novel. The aggregation of these deeper sentiments, or *impressions*, is an open-ended task: since we do not know the range of impressions a literary work might create for a reader, our impression-mining is fully unsupervised.

This open-ended task of summarizing reader reviews has been recently discussed in work [43] where authors introduce novel approaches to model “genre tags” from the text of book reviews. These methods use *LibraryThing*, an online cataloguing service that hosts review corpora: one for each book similar to the Goodreads.com corpora that we use for our work. What differentiates this approach from previously noted classical work is that the user-suggested genre tags in *LibraryThing* are sampled from an unconstrained vocabulary or “lexical space”. In this vein, we seek to extend this idea of estimating the genre of a story to the broader context of the novel we call impressions.

Last, there is burgeoning interest in the computational text analytic field of developing infinite vocabulary Knowledge Graph (KG) generation pipelines [44]. In our KG context, *infinite vocabulary* means that the node and edge labels are not defined according to a schema or rule *prior* to data aggregation. Instead the goal is to estimate the schema from the data. A majority

this is just one way of modeling stories and event sequences, various approaches have been tried. E.g. Min, S., & Park, J. (2019). Modeling narrative structure and dynamics with networks, sentiment analysis, and topic modeling. PLOS ONE, 14(12), e0226025. <https://doi.org/10.1371/journal.pone.0226025>

of these approaches aggregate relationships via OpenIE [45] (from the Stanford NLP Toolkit) or a comparable relationship extractor. Once the infinite vocabulary network is estimated, a post-processing block attempts to compress the KG via existing ontologies such as WikiMedia [46] and YAGO [47]. Our pipeline approaches KG creation as a *narrative modeling* problem which is, by nature, infinite-vocabulary: these narratives feature rich characters, complicated story lines and numerous events and situations. The model we develop boosts the off-the-shelf OpenIE relationship extractor with interconnected blocks of coreference resolution and other filtering techniques that increase the recall of the relationship extraction pipeline and the resulting KG [7].

3. Data

We created a corpus of reader reviews for five novels from *Goodreads*: *Frankenstein* [48], *Of Mice and Men* [49], *The Hobbit* [50], *Animal Farm* [51], and *To Kill a Mockingbird* [52]. We chose these books from the list of most commonly rated fictional works on the site, focusing on works with ratings > 500,000. We then considered the number of reviews for these highly rated novels, recognizing that many highly rated novels have considerably fewer reviews than ratings. At the time of data collection, for example, *The Hobbit* had more than 2.5 million ratings, but only 44,831 reviews.

Once we had a candidate list of works to consider, we reduced that list to our five target novels, with the goal of selecting works that had significantly different types of characters (humans, supernatural creatures, animals), different narrative structures, and that employed a variety of narrative framing devices (frame narratives, flashback, etc). *The Hobbit*, for instance, is a complex multi-episodic narrative that unfolds in a largely linear fashion, following in broad strokes the well-known “hero’s journey” plot [53]. It uses a mixture of character types, including wizards, hobbits, dragons and dwarves, while the setting is a fictional fantasy world modeled loosely on the Nordic mythological one. By way of contrast, *Of Mice and Men* is a realistic novella set in depression-era California with a small cast of human characters, and instantiates Vonnegut’s “From Bad to Worse” plot [54]. *To Kill a Mockingbird* is a complex intertwined narrative told from the first person point-of-view of a child, and engages with nuanced ideas about justice and race in America. *Frankenstein*, written as an epistolary novel and thus told largely in flashback, has multiple, complex characters, both human and non-human. Its plot is largely linear with a clearly recognizable hero and villain. *Animal Farm*, in contrast, has a very large cast of characters, many of them anthropomorphized animals, with a plot that cycles through episodes that bear striking resemblance to earlier ones (such as the rebuilding of the windmill).

The reviews of a target novel were extracted using a custom pipeline since existing corpora either did not contain the correct books, enough book reviews, or enough book reviews of appropriate length. For example, the UCSD *Goodreads* review dataset [28,29], despite having an enormous number of book reviews, did not have enough book reviews of appropriate length per book that our methods require.

After choosing our target novels, we designed a scraper to operate in accordance to the specifications of the *Goodreads* site. After downloading the reviews for each book, we filtered and cleaned them to avoid problems associated with spam, incorrectly posted reviews (i.e. for a different novel), or non-English reviews. We also filtered for garbled content such as junk characters and for posts that were so short as to be unusable. Following the process of filtering, we had the following number of eligible reviews per book: *Frankenstein* (2947), *The Hobbit* (2897), *Of Mice and Men* (2956), *Animal Farm* (2482), and *To Kill a Mockingbird* (2893). These reviews can be found in our data repository [55].

We then passed the scraped reviews to the first data-preparation modules of our pipeline. During this largely NLP-based process, we found two main types of phrases: (i) Plot phrases containing both the actants and their relationships. These phrases describe what a reviewer believes happened to a subset of the actants, and how they interacted with each other and, consequently, these phrases are of primary interest to us. (ii) Opinion phrases that reveal a reader’s opinions about the book, the author, or the characters and events in the book. The

good choice of corpus, but I think it is also important to acknowledge that these are literary classics and their prestige can influence readers’ behaviour. A comparison with popular fiction might show completely different patterns (e.g. Castano, E., Martingano, A. J., & Perconti, P. (2020). The effect of exposure to fiction on attributional complexity, egocentric bias and accuracy in social perception. *PLOS ONE*, 15(5), e0233378. <https://doi.org/10.1371/journal.pone.0233378>)

relationships we extract from these phrases are the predominant ones when aggregated across all readers' posts.

4. Methodology

A network of characters (and other actants) interconnected by their relationships can serve as a useful representation of an aggregated model of readers' responses to a literary work. To address this challenge, previous research on a dataset of *Goodreads* reviews introduced an effective pipeline comprising two important tasks, Entity Mention Grouping (EMG) and Inter-Actant Relationship Clustering (IARC), to address this challenge [56].

The EMG task is a labeling process that aggregates multiple entity mentions from the extracted relationship phrases (subject \hat{s} or object \hat{o}) into a single character. This aggregation is accomplished through an evaluation of the similarity between a pair of entity mentions by observing their interactions with other entity mentions. For example, in *The Hobbit*, the two entity mentions, "Bilbo" and "Baggins", frequently interact with the entity mention "Gandalf", and with semantically equivalent relationships; as a result, these two mentions, "Bilbo" and "Baggins", likely refer to the same character, here the hobbit, "Bilbo Baggins".

To formulate this task, let the set of entity mentions empirically observed in reviews be \hat{E} . A smaller character set E refers to a finite vocabulary of distinct characters in a literary work. The EMG step is then defined as a surjective function $f: \hat{E} \rightarrow E$ that maps entity mentions to characters. The resultant mapping of entity mentions to characters in the EMG task provides a semantically-informed aggregation tool for the original corpus of relationship tuples. Tuples sharing entity mentions mapped to the same character can now be aggregated to form larger relationship sets between a pair of characters (as opposed to a pair of entity mentions). For example, in *Of Mice and Men*, the entity mentions "George" and "Milton" are successfully mapped to the character "George" with the EMG task.

The IARC task, on the other hand, is designed to aggregate relationship phrases generated as output from a relationship extraction module. A relationship tuple consists of a subject mention, relationship phrase and object mention $(\hat{s}, \hat{r}, \hat{o})$ that is directly extracted from a review and thus contains partial information about the structure of the underlying narrative model [56]. For every ordered pair of characters $\{e_i, e_j\}$, we obtain an aggregated set of relationship phrases from the reviews, \hat{R}_{ij} , that connects e_i to e_j .

The EMG task implicitly aids in the aggregation of larger sets of relationship phrases since it aggregates entity mentions for each character. \hat{R}_{ij} is the union of all the relationship phrases between entity mentions that compose e_i and e_j . We seek to cluster these relationship phrases in \hat{R}_{ij} and assign each resulting cluster to a label in a set R_{ij} . This process of clustering, Inter-actant Relationship Clustering (IARC), can accordingly be defined by another surjective function $g_{ij}: \hat{R}_{ij} \rightarrow R_{ij}$. Specific details about assembling the set of labels R_{ij} for the IARC task are provided in the relevant subsections below. For example, a few of the relationship phrases between "Atticus" Finch and "Tom" Robinson in reviews include: defends, is defending, protects, represents, supports. These phrases are semantically similar and appear in the same cluster; this cluster is subsequently labeled "defends".

In earlier work, relationship extraction along with the two-step process (EMG, IARC) to condense the relationship tuple space was applied to the corpus of reader reviews for four works of literary fiction: *Of Mice and Men*, *To Kill a Mockingbird*, *The Hobbit*, and *Frankenstein* [56]. The resulting aggregated networks described consensus (across reviews) story network graphs, with each node in the network representing a character and each directed edge representing a relationship between a pair of characters.

This preliminary work lacked important KG tools helpful for addressing problems in reader response and narrative modeling. For example, in addition to estimating the narrative framework of a novel, which may reflect the readers' understanding of important characters and inter-character relationships, it is useful to estimate a larger graph that also includes key, often metadiscursive or extra-diegetic, actants related to the novel's *reception* such as the «author», a

excellent evaluation of the limits of previous research! The modelling of both extra-textual and temporal information is extremely important.

It may not be obvious to everybody what this is

this part fits better in the introduction, not in the methodology section

«reviewer», or even references to cross-media adaptations such as the «movie director». Similarly, while earlier work generates story network graphs that are static and that summarize the entire story, estimating the temporal dynamics of the story underlying the reviews can greatly expand the usefulness of these graphs. This dynamic view of the work can provide insight into how reviewers remember the unfolding of the narrative. Last, while the previous narrative graphs highlighted the interactions *between* actors, they do not model the reviewers' *impressions* of the actors, important information in the context of an analysis of reader response.

In this vein, the following sections describe the rest of our methodology that: (a) Extends the network generation pipeline to create expanded narrative framework network graphs; (b) Introduces a new application of this pipeline for story *sequencing* and; (c) Profiles frequently-mentioned characters in the selected novels based on context-specific reviewers' impressions.

(a) Expanded Story Network Graph

While the nodes in a *regular* narrative framework graph for a novel are derived from an associated external resource such as a canonical character list from *SparkNotes* or the work itself, the discovered relationships from our pipeline (edges) are extracted from readers' reviews. These extractions reveal both readers' thoughts and their impressions of characters in the works under discussion. Not surprisingly, then, the reader-derived networks expand upon the regular story network graphs, with additional nodes for film directors, authors, and other interlocutors, and additional edges for these additional actants' activities with the core story.

To facilitate this expansion, we augment the regular story network for a novel with nodes for frequent entities *not* among the character mentions. First, we rank mentions based on their frequency and then we pick the top ranked entity mentions to add them to the story graph. For example, in *The Hobbit*, some of the extra candidate mentions are: ['tolkien', 'book', 'story', 'adventure', 'movie', 'jackson'].

Next, we find the edges that exist between these candidate nodes and the regular story network nodes. If a candidate node has significant edges with the existing characters in the story network, we augment the graph with this node. For example, the candidate node "Tolkien" is connected to the character "Bilbo" with the verb phrases ['to masterfully develop', 'provides', 'knocked', 'introduced']. There are other interesting but distant mentions of candidate nodes that do not have *direct* connections to nodes in the original story graph: for example, "Jackson" (a reference to the director of *The Hobbit* filmatization, Peter Jackson) only appears in the triplet (Jackson 's changes, distort, Tolkien 's original story). While this node represents the reviewers' acknowledgement of the novel's movie adaptation, we do not represent "Jackson" as a node in our expanded graph due to its sparse connection with the rest of the story network.

As a result, our algorithm discovers all the available relationships between main novel characters and each metadiscursive candidate node mention. It draws an edge only if there are more than 5 relationships between the candidate and a story graph character. If that candidate node has a degree of 3 or greater, it is added to the expanded story network graph.

(b) REV2SEQ: A New Event Sequencing Algorithm

Each *Goodreads* review provides partial information for creating a collective summary of the reviewed work. Reviewers share their interpretation of the story's sequence of events, motivated in part by a universal understanding of time and in part to convey information about the story line(s) to the reader of the review [57]. Within this shared timeline, reviewers intersperse their opinions, argue over aspects of theme and plot, and offer other thoughts about style, imagery or comparisons to other works of fiction. Individual reviews sometimes highlight pivotal or critical moments in the story and their interdependence, and at other times elaborate on minutiae that have, at least for the reviewer, great importance.

who are they? professional literary critics or amateur readers?

Within an individual review, one can use multiple syntactic tools – including, tense, punctuation, retrospective language and conjunctions – to encode temporal precedence relationships among events. Unfortunately, the NLP tools required to decode a linear timing sequence from such complex syntax are not very well developed [58]. Given that the reader reviews are short and meant for social media sharing, we hypothesize that most reviews are written in a linear manner, and that temporal precedence is usually accurately encoded by the order in which events are mentioned in the text. Any errors introduced by this assumption in the inferred temporal precedence order – for example, due to some readers using tense to indicate temporal dependence when the event mention order is actually reversed – would create cyclical temporal dependencies among events when aggregated over all the reviews. Our accompanying hypothesis is that, because of the large number of reviews in which the majority follows a linear time encoding scheme, such cyclical dependencies can be computationally resolved by incorporating a majority voting scheme in the associated algorithms. Our results, evaluated by **story-aware testers**, validate our hypotheses, with the output of the computational steps yielding aggregated sequencing results close to the ground truth.

In order to verify this hypothesis, we first define an event as an ordered tuple of characters connected by a relationship cluster. In other words, an event $p \in P$ is an ordered tuple of the subject character (s), relationship label (r), object character (o), where $s \in E$, $o \in E$ and $r \in R_{s,o}$. Recall that E is the set of characters as returned by the EMG task and $R_{s,o}$ is the set of labels that tag the relationship clusters in the IARC task. Each review, p_k , then consists of a sequence of events: $[(s_1, r_1, o_1), \dots, (s_n, r_n, o_n)]$. For example, a reviewer of *The Hobbit* might mention «Bilbo Baggins»'s upbringing in «Hobbiton» before describing «Bard»'s slaying of «Smaug». Similarly, in reviews for *Of Mice and Men*, reviewers more frequently mention that «George» finds the «ranch» before noting that «Lennie» kills «Curley's wife». If one considers all of the reviews in the aggregate, they should present a summary of the novel's story lines from the perspective of the reviewers, including a series of alternate or abbreviated paths from the beginning of the novel to its end. Consequently, the challenge is to jointly estimate *the sequence of events* that describes the story of a particular work of fiction based solely on the reviews.

(i) Implementation Details

To construct an event, we extract entity groups and relationships by running the Entity Mention Grouping (EMG) and the Inter-Actant Relationship Clustering (IARC) functions. A new implementation of the IARC task encodes relationship phrases with entailment-based pretrained BERT embeddings [59] that are subsequently clustered with HDBSCAN, a density-based algorithm suitable for clustering BERT embeddings [60,61]. The resulting clusters are labeled with the most frequent sub-phrase in each cluster along with contextual words that best connect the subject character to the object character based on the review text from which the relationship tuples were extracted. Noisy (diffuse) clusters are filtered by HDBSCAN and further manually filtered so that we only consider high-quality relationship clusters. For example, clusters that involve relationships that are not sequenceable – such as the relationship cluster in *Of Mice and Men*: «Lennie» ["wish" {for}, "hope" {for}] «ranch» – are avoided via a keyword search for hypothetical words "wish" and "hope" and also by manually parsing the resultant HDBSCAN clusters.

Occasionally, some of the automatically generated clusters are semantically similar. For example, in *The Hobbit*, we find two semantically similar clusters: «Gandalf» ["recruits", "recruit"] «Bilbo» and «Gandalf» ["chose", "chooses"] «Bilbo». In such cases, the clusters are merged before labeling and the final event unambiguously describes the action of «Gandalf» *recruiting* «Bilbo» for the task of leading the Dwarves. These labels constitute the range of the mapping for the IARC task $\mathcal{R}(g_{i,j})$ where i refers here to «Bilbo» and j to «Gandalf».

Events are not aggregated from relationship tuples where: (a) the parent sentences are hypothetical (e.g. ones that contain "would", "could", "should"); (b) subject and object phrases have headwords that are identical; are in a list of stopwords that contain mentions such as

I think subsections i, ii, and iii could be summarized here briefly and the details moved in appendix. This technical part is interrupting the flow of the research narrative. E.g. the following paper is an example of highly technical paper communicated in a way that successfully reached humanities scholars: Reagan, A. J., Mitchell, L., Kiley, D., Danforth, C. M., & Dodds, P. S. (2016). The emotional arcs of stories are dominated by six basic shapes. *EPJ Data Sci.*, 5, 5–31. <https://doi.org/10.1140/epjds/s13688-016-0093-1>

["I", "We", "Author"]; or do not have associated characters produced by the EMG task; or (c) parent relationship phrases start with verbs in their infinitive form. If the relationship part of a relationship tuple features a single word (generally a verb), it is reduced to its lemmatized form. These filters are applied while retaining the relative ordering of the events in each review.

The event sequence described by every review is aggregated into a shared precedence matrix $\hat{M} \in \mathbb{R}^{(|P|+2) \times (|P|+2)}$ where $|P|$ is the number of distinct events. In this matrix, one row and one column is reserved for each event with 2 additional rows and columns for the objective START and objective TERMINATE of a story. $\hat{M}[i, j]$ aggregates the number of reviews in which event p_i precedes p_j .

Matrix \hat{M} is preprocessed as follows: (1) The row representing START is set to a constant to model a reviewer's starting event in their own partial sequence of events as a uniform distribution; (2) Similarly, the column representing TERMINATE is set to a very small constant to represent the likelihood that a reviewer might terminate their review after any event; (3) Finally, the rows of \hat{M} are normalized to model the *likelihood* of a subsequent event mapped to a particular column based on a current event mapped to a particular row. We call this processed matrix M .

Matrix M can now be re-imagined as modeling the empirical likelihood of directed edges in a graph \mathcal{G} whose nodes are the events mapped to the rows and columns of M . In an ideal scenario, \mathcal{G} would be a Directed Acyclic Graph (DAG) with every event p_i unambiguously preceding or succeeding another event p_j . This, however, is not the case in our estimation – our approach uses event precedence as a *proxy* for event sequences and this proxy is noisy. This noise manifests itself in the form of cycles, where $p_i \rightarrow p_j \dots p_k \rightarrow p_i$ is an inconclusive and cyclical ordering of events. Therefore, we seek to find a way to resolve these cycles.

Estimating a DAG from \mathcal{G} by removing the minimum number of edges is a candidate solution but it is also an NP-complete problem. We can, however, perform two trivial steps toward accomplishing this goal: 1. Remove isolates (single event disjoint paths from START to END) and self-loops; and 2. Remove 2-node cycles by comparing $M[i, j]$ and $M[j, i]$ for all (i, j) pairs and keeping only the larger value – a majority rule. To remove longer cycles and to create the final event sequence network, we introduce the Sequential Breadth First Search (SBFS) algorithm which employs a greedy heuristic to remove edges that form loops.

(ii) Sequential Breadth First Search (SBFS) Algorithm

The SBFS algorithm (see Algorithm 1) is motivated by Topological Sorting, a classical algorithm used to query whether a directed graph is indeed acyclic. Starting from a node with 0 in-degree, the naive algorithm parses the directed edges of the graph to generate a flattened and linear representation. If the algorithm is unable to create this representation, the graph is not a DAG. Our algorithm is reminiscent of the Topological Sorting algorithm in that it allows an event (node) discovered in an earlier iteration of the sorting to be pushed further along the shared linear representation if another event discovered in a later iteration in fact precedes it. This linear representation is interpreted as the estimated objective timeline of the story.

On the other hand, our implementation differs from a simple topological sorting in that, at each step, multiple vertices can co-occur *in parallel* to one another. This feature is critical in our use case because stories often have parallel plot lines which a simple and linear topological sorting algorithm would not explore. Furthermore, a greedy heuristic removes cycles of length > 2 (note that self loops and frequent 2-node cycles have already been filtered). If an encountered edge creates a cycle during the construction of a DAG, that edge is broken. This sequencing algorithm satisfies precedence constraints greedily and models closely the iterative steps that a human would attempt in order to resolve the event sequence for a novel: having observed an event, a human would explore all potential succeeding events and resolve precedence conflicts based on their existing worldview of the event sequence.

The output of the algorithm is a directed edge cover \mathcal{G}' of \mathcal{G} and associated time steps per event $T(v) \forall v \in V(\mathcal{G}')$. The resulting event sequence network now describes a generalized topological ordering of an estimated DAG, ordered by time steps T . Extraneous edges are automatically

Algorithm 1 Sequential Breadth First Search (SBFS)**Input:** M , the filtered precedence matrix**Output:** \mathcal{G} , the consensus event sequence, T , the resulting time-steps

```

CURRENT TIME = 0
Current Stack,  $s_{curr} \leftarrow [0]$ 
Next Stack,  $s_{next} \leftarrow []$ 
Precedence matrix for the final event sequence,  $M' \leftarrow \text{zeros like } M$ 

while  $s_{curr}$  is not empty do
  for pop element  $i$  from  $s_{curr}$  do
    Succeeding events to  $i$ :  $\text{children} = \text{indices where } (M[i][:] > 0)$ 
    for child in children do
      if there already exists a path in  $M'$  from child to  $i$  then
        Ignore path  $i \rightarrow \text{child}$  causing cycle.
      else
         $M'[i][\text{child}] = 1$ 
         $T[\text{child}] = \text{CURRENT TIME} + 1$ 
        if child not in  $s_{next}$  then
          push child into  $s_{next}$ 
        end if
      end if
    end for
  end for

   $s_{curr} \leftarrow s_{next}$ 
   $s_{next} \leftarrow []$ 
  CURRENT TIME = CURRENT TIME + 1
end while
 $\mathcal{G} \leftarrow \text{graph of } M'$ 

```

removed from \mathcal{G}' such that if there is a path from e_k to e_j through e_i , the direct edge from e_k to e_j is deleted.

During visualization, directed edges always face downwards and, as a trivial verification, the node representing START appears at the top of the event sequence and the node representing TERMINATE, appears at the bottom.

(iii) Evaluating the Sequencing Networks

To quantify the accuracy of these sequencing results, we asked $N_{rev} = 5$ independent reviewers to evaluate every edge in the graphs for each novel by labeling them as 1: *correct*, 0: *incorrect* or X : *unsure*. Those marked X are globally labeled as either all 0 or all 1 to obtain the lower bound and upper bound (along with the error bound) for the algorithm's performance. Reviewers were required to have read the stories and they used *SparkNotes* as a reference to disambiguate otherwise obscure precedence relationships. Each reviewer reviewed all 5 networks to negate any variability due to implicit human biases across the stories. Edges that comprised a source labeled START or a target labeled TERMINATE were ignored in the computation of accuracy: these specific source and target labels are not intrinsic to a story's plot but are instead artefacts from our algorithm – not removing these edges would result in inflated performance measures. Accuracy is measured in 2 ways - where \hat{E} is the subset of edges in each sequence network that does not originate from START and does not end at TERMINATE:

- Weighted Accuracy, $\text{score}_{\text{weighted}} = \frac{1}{N_{\text{rev}} \times |\hat{E}|} \sum_{i=1}^{N_{\text{rev}}} \sum_{j=1}^{|\hat{E}|} \text{label}_{i,j}$,
- Simple Majority, $\text{score}_{\text{simple majority}} = \frac{1}{|\hat{E}|} \sum_{j=1}^{|\hat{E}|} \text{Majority}\{\text{label}_{1,j}, \dots, \text{label}_{N_{\text{rev}},j}\}$.

(c) SENT2IMP: Character Impression Extraction

Our relationship extraction pipeline captures not only the relationships *between* characters, but also a set of relationships in which readers have expressed their impressions of characters. For example, while we obtain the classical relationships such as «Gandalf» “chooses” «Bilbo» for our expanded story networks, we also obtain relationships such as «Bilbo» “is” «unbelievably lucky». These later relationships when aggregated, filtered and clustered bring to light the different contexts, behaviors, and reader sentiments associated with each character. In our setting, an *impression* is formally defined as a cluster of semantically similar phrases from reviews that imply a single aspect of a character in the eyes of the readers. Our pipeline promises to provide an unsupervised approach to model a character as a *mixture* of such impressions.

We start by selecting the relationship tuples labeled as “SVCop” from the full set of extracted relationships. “SVCop” are those with the structure, *Subject Phrase* → *Verb Phrase* → *Copula*. These relationship tuples typically consist of a noun phrase and an associated adjective phrase that provide descriptive information about the noun phrase. We observe that a majority of these “copular” phrase relationships contain information about character impressions. For example, in the reviews of *Animal Farm*, the sentence “Snowball was humble and a good leader” yields the two SVCop relationships: (Snowball, was, humble), (Snowball, was, a good leader), where the phrases “humble” and “a good leader” describe Snowball. In the aggregate, these phrases contribute to *impressions* of Snowball being both humble and a good leader. In addition, we note that these relationships are frequent and comprise a significant portion of the extracted relationships per literary work: reviews contain a wide range of reader impressions when compared to the original work. Filtering and clustering these frequent relationships supports the creation of a *robust* pipeline for impression extraction.

Once we have extracted the SVCop relationships from the entire corpus, we group the relationships with respect to the character $e_i \in E$ present in the subject phrases of the extractions. Next, these phrases are transformed into vectors using a fine-tuned BERT [59] embedding. We embed these phrases in the BERT space to acquire a measure of semantic similarity. These vectors are then passed to the HDBSCAN clustering algorithm, which then determines the optimal set of clusters, C'_{e_i} , per character e_i . This approach results in a qualitatively optimal clustering of phrases.

Due to the noisiness of the reader review data, as well as the inherent non-Gaussian distribution of BERT embeddings, some of the resulting clusters are not homogeneous. To mitigate the noisy clusters, we employ a modified TF-IDF [62] scoring function to weight the words in each cluster. The score of a word in a cluster is equal to its frequency in the same cluster times the TF-IDF score in the review corpus for that word, where each review is a document and each word is a term. This score for a distinct word w_m with frequency f_{m, \hat{C}_k} in a cluster of phrases $\hat{C}_k \in C'_{e_i}$, $|\hat{C}_k| = N$ is given as:

$$x[m, \hat{C}_k] = \text{TF-IDF}(w_m) \times f_{m, \hat{C}_k}.$$

After removing the stop words, we observe that the meaningful clusters have a skewed-tailed distribution over these scores. In some extreme cases, where a cluster only contains a few words or phrases, the distribution is centered on a high score with low variance. For example, for the character “Gandalf”, we find numerous “Gandalf” clusters, with one that contains only the word “wizard”. We select a cluster to be meaningful based on the variety of its highly scored words. An ideal cluster consists of a handful of high scoring words. The fewer low scoring words a cluster contains, the higher its quality and, consequently, we expect an ideal cluster to have less noise.

something is odd here: the copula is the verb phrase connecting a subject to a complement

I guess there will be also many statements about characters in the form “I didn't like (how) character_x (did/behaved) ...”. The SVCop technique will miss them.

To quantify this cluster quality measure, we calculate the skewness g_1 [63]:

$$g_1 = \frac{m_3}{m_2^{3/2}},$$

where,

$$m_i = \frac{1}{N} \sum_{n=1}^N (x[n, \hat{C}_k] - \bar{x})^i,$$

is the biased sample's i^{th} central moment, and \bar{x} is the sample mean [64].

For a cluster \hat{C}_k , the skewness of the distribution of $x[m, \hat{C}_k]$, determines the quality of the cluster. High positive skewness shows that the cluster consists mostly of high-scored words with a low number of infrequent words. Occasionally, when a cluster has a small number of low-scored words, such as the "wizard" cluster for "Gandalf", it loses its skewness. Instead, we observe a high word score ($x(\cdot, \cdot)$) average. We consider a cluster to be valid if it has skewness above a certain threshold or if its words' scores have a high average with low variance. Let this filtered set of impression clusters for a character i be C_{e_i} .

As a result of these selection, clustering and filtering tasks, we obtain a filtered mixture of clusters of reader impressions per character C_{e_i} . Each cluster presents a unique description of a character within a novel. For example, "George" in *Of Mice and Men* has various clusters associated with him including one recognizing that he is Lennie's protector: ['basically Lennies protector', 'the guardian', 'in charge of Lennie ', 'Lennie guardian']. In another cluster, he is considered to be a somewhat tragic character with a series of descriptive phrases creating this impression: ['clumsy', 'unhappy', 'sad', 'nervous', 'very rude', 'selfish', 'painfully lonely'].

In order to quantify the geometry of the obtained mixture, we define a distance metric between every pair of clusters of numerical embeddings (see Algorithm 2). The resultant measure is in the range [-2,2] and is close to 2 if the pair is semantically similar and close to -2 if the pair of clusters are semantic opposites. We once again use BERT embeddings where semantic similarity is measured as the cosine distance between a pair of phrase embeddings. Before computing the cosine similarity, we reduce the dimension of the BERT embeddings to 4-principal components using Principal Component Analysis (PCA), having found that the resultant scores generalize well with this choice of principal components. It follows that for a *mixture* of clusters, we can represent the obtained inter-cluster distance measure between \hat{C}_i and \hat{C}_j (within a mixture C_{e_i}) on a heatmap that is symmetric – because our distance measure is symmetric – (see Algorithm 3) for a particular character. To ensure that the heatmaps we generate are rich, we limit our study to those characters with at least 4 clusters of descriptive phrases ($|C_{e_i}| \geq 4$).

(i) Quantifying and Visualizing the Complexity of a Character

The heatmap for a single character shows an empirical measure of *character complexity*. In this task, we run Algorithm 3 such that both characters in the pairwise comparison are identical. A large and high-variance heatmap implies that readers consider the character to be *complex*. This complexity may (i) reflect disagreement in the reader discussions about that character, implying that impressions of the character are *controversial*, or (ii) capture contradictory and multi-modal character traits that authors develop – or that readers constitute through their reading – to portray remarkable characters in their novels. One finds instances of both in the impression clusters of Bilbo Baggins (See Table 2): The first two clusters that portray Bilbo as "unpleasant/boring" and "loveable/charismatic" are most likely instances of readers' contradictory perceptions; Clusters 2 and 3, however, portray Bilbo simultaneously as a hero and a thief, and are most likely diverse dimensions inherent in the novel and picked up on by the readers. In other cases, when the heatmap is smaller and of low-variance, readers' perceptions of the profiled character are not as diverse, perhaps because of the character's secondary role in the novel and/or the character's narrow/singular purpose in the story. Such a character would be less *complex*.

This qualitative understanding of *complexity* can be quantitatively described by *entropy*. One can assume that the heatmap entries of any character are samples from an underlying random

move this paragraph to the beginning of this section (c), so it will be easier to understand the purpose and follow the various steps of the procedure

why PCA now if you previously said that HDBSCAN is good to cluster BERT embeddings?

variable, and the complexity of a character is its entropy. The more spread out the underlying distribution is, the higher its entropy. Since we have only a few samples – coming only from the lower half of any given symmetric heatmap matrix – it is best to model the random variable as a discrete probability distribution in the range of scores in our heatmap, $[-2, +2]$.

First, we define the number of bins for the distribution, b . Then, from the impressions heatmap of a single character, we slot all the values below the major diagonal into these bins. Because most of our heatmaps do not have enough values to populate each bin, we adopt a standard numerical technique [65] of using a smoothing kernel – in our case uniform – (of width w bins) across the bins. In general, w and b are hyperparameters that change the sensitivity of calculating the *entropy*.

More formally, for a heatmap $S \in \mathbb{R}^{N \times N}$ where N is the number of impression clusters for the associated character, the entropy (complexity) \mathcal{H} is defined:

$$\mathcal{H} = - \sum_{\{i \in [b]\}} P_i \log P_i,$$

where,

$$S_{ij} \xrightarrow[\forall i > j]{\text{aggregate}} \text{histogram bins}_{[-2,2]}^b \xrightarrow{\text{smoothing kernel (w), norm}} \text{Prob. Distribution } P_i.$$

(ii) Distinct Character Comparison

The observation that some characters are, in the minds of the readers as reflected in their reviews, more *complex* than others motivates our use of the distance measure employed in Algorithm 2 for impression clusters derived from a pair of *distinct* characters. In this case, we project onto the heatmap the distance measures between clusters from two separate mixtures (the heatmap will not be symmetric and may not be square). Such a representation highlights the smaller number of *contexts* in which two characters are similar even across novels. For example, in *To Kill a Mockingbird*, one of Atticus’s clusters consists of the phrases [‘the father of kids’, ‘the father of protagonist’, ‘the father of Jem’, ‘lenient father’, ‘the father of character’, ‘a father figure’] and can be compared to the first example of “George”’s attribute of being a “guardian”. Although these two sets of phrases are not exactly the same, one can still recognize that “George” and “Atticus” are perceived similarly in their shared roles within their respective works.

Algorithm 2 Computing the Similarity Score between a Pair of Clusters of Numerical Embeddings

Input: \hat{E}_m, \hat{E}_l : Two Clusters of Numerical Embeddings

Output: $S_{l,m}$

***** From \hat{E}_l to \hat{E}_m *****

$s_1 = 0$

for every embedding u in \hat{E}_l **do**

$s_1 = s_1 + \max_{v \in \hat{E}_m} \cos(v, u)$

end for

$S_1 = \frac{s_1}{|\hat{E}_l|}$

***** From \hat{E}_m to \hat{E}_l *****

$s_2 = 0$

for every embedding v in \hat{E}_m **do**

$s_2 = s_2 + \max_{u \in \hat{E}_l} \cos(v, u)$

end for

$S_2 = \frac{s_2}{|\hat{E}_m|}$

$S_{l,m} = S_1 + S_2$

Algorithm 3 Computing the Heatmap between a Pair of Mixtures of Phrase Clusters

```

Input:  $C_{e_i}, C_{e_j}$  two mixtures of phrase clusters such that,
 $C_{e_i} = [\hat{C}_{1,e_i}, \hat{C}_{2,e_i}, \dots, \hat{C}_{M,e_i}]$ 
 $C_{e_j} = [\hat{C}_{1,e_j}, \hat{C}_{2,e_j}, \dots, \hat{C}_{N,e_j}]$ 
Output:  $S \in \mathbb{R}^{M \times N}$ : A heatmap  $S$  with  $S_{lm}$  equal to the similarity between the  $l^{th}$  cluster in one mixture and the  $m^{th}$  cluster in another mixture.

for iter in  $[1, \dots, M]$  do
     $E_{iter,e_i} = \text{BERT} [\hat{C}_{iter,e_i}]$ 
     $\{ \triangleright E_{iter,e_i} \in \mathbb{R}^{768 \times |\hat{C}_{iter,e_i}|}$ ; our BERT embeddings are 768-dimensional vectors.  $\}$ 
end for
for iter in  $[1, \dots, N]$  do
     $E_{iter,e_j} = \text{BERT} [\hat{C}_{iter,e_j}]$ 
end for

 $\{ \{ \hat{E}_{1,e_i}, \hat{E}_{2,e_i}, \dots, \hat{E}_{M,e_i} \}, \{ \hat{E}_{1,e_j}, \hat{E}_{2,e_j}, \dots, \hat{E}_{N,e_j} \} \}$ 
 $= \text{PCA} [E_{1,e_i}, E_{2,e_i}, \dots, E_{M,e_i}, E_{1,e_j}, E_{2,e_j}, \dots, E_{N,e_j}]$ 
 $\{ \triangleright \hat{E}_{iter,e_i} \in \mathbb{R}^{4 \times |\hat{C}_{iter,e_i}|}$ ; 4 principal components.  $\}$ 

for iterM in  $[1, \dots, M]$  do
    for iterN in  $[1, \dots, N]$  do
        Perform Algorithm 2 for the pair of clusters of numerical embeddings,  $\{ \hat{E}_{iterM,e_i}, \hat{E}_{iterN,e_j} \}$ 
    end for
end for

```

5. Results and Discussion

(a) Story Network Creation and Expansion

The resulting story network graphs for the five works are presented in Figure 1, with a clear visual distinction between the diegetic nodes (i.e. the novel’s characters) and the metadiscursive or extra-diegetic nodes (i.e. actants not in the novel *per se*).

These expanded graphs reveal interesting features not only about readers’ perceptions of the stories, but also of how readers conceptualize authorship as well as other external features relevant to an understanding of the novel. For example, the authors of *Of Mice and Men* and *The Hobbit* are directly linked to main characters in those novels, whereas for the other novels the authors are only connected to the main story graph through intermediary nodes. The close connection of both **Tolkien and Steinbeck** to the main story graphs possibly highlights the readers’ perceptions of the author as equally important to any discussion of the novel for these two works. For *Frankenstein*, by way of contrast, the expanded graph captures the complex discussions of “authoriality” that pervade both the narrative and meta-narrative space. For the other two novels in our corpus, the author appears to play a slightly more divorced role, at least in the reader discussions. While our data collection occurred prior to the release of Harper Lee’s *Go Set a Watchman* [66], which may well have triggered greater awareness of Lee as an author, the reviews for *To Kill a Mockingbird* and *Animal Farm* may be capturing a reduced awareness of the authorships of Lee and Orwell, as opposed to the broadly recognized authorships of Tolkien and Steinbeck.

The extended networks include a proliferation of generic terms such as “people” and “story” possibly capturing readers’ awareness of other readers – the generic “people” – and the narrativity of the work itself. Other terms such as “place” and “home” may reflect readers’ efforts to tie the novel into their own understanding of the world and, possibly, considerations of locality and the

this claims about authorship need a bit more unpacking. The high out-degree of author-nodes can just be a reflection of the general fact the an author crafts the story and, thus, they are more likely to be the subject of a proposition about a story or its elements

Orwell, too (-> Napoleon, humans, animals).

domestic. The “ways” node appearing in most of the expanded story graphs accounts for the strategies that characters pursue in the target novel. For example, in *To Kill a Mockingbird*, “Scout” is connected to the “ways” node with relationships such as “looked” and “thought”, reflecting readers’ awareness of Scout’s efforts to both comprehend and solve the fundamental challenges she encounters in the novel.

These metadiscursive nodes share an additional interesting feature: the edges connecting them to the character nodes in the main story graph are mostly in a single direction (either toward the story graph or away from it) as in, for example, *Of Mice and Men* 1a. By way of contrast, main story characters interact with each other generating a mixture of in- and out-edges. The *Animal Farm* graph presents an intriguing example of this directionality, with the “Orwell” node having only outwardly directed edges, and the remaining additional metadiscursive nodes only having inwardly directed edges. There are two exceptions in this graph to this general rule: the nodes, “revolution” and “rebellion” share an outwardly directed edge connecting them to the diegetic node of the “farmhouse”. Readers here collectively recognize not only the strategy of “revolution” that animates the novel, but also the focus of the uprising on the locus of institutional authority, here the “farmhouse”.

The expanded novel graph for *Frankenstein* reveals a large number of metadiscursive nodes that are not directly connected to the main story component, creating a secondary network of high-degree metadiscursive and extra-diegetic nodes, thereby capturing a broad reader conversation not centered on the novel itself. To highlight this aspect of the reader conversations, we extended the additional nodes by finding the inter-connections between a pair of candidate mentions. This secondary network reveals a lively conversation not only about the story plot(s) but also about meta-narrative considerations such as the composition of the novel, its epistolary frame narrative, Mary Shelley’s authorship, and philosophical speculation about “God”.

Given the implications of the expanded narrative graphs, we would be remiss to dismiss *Goodreads* reviews as amateurish plot-focused summaries, since they capture more sophisticated speculations of a broad readership, reflecting a latent diversity of opinion that may well be an echo of Fish’s communities of interpretation. Indeed, the methods presented here capture the voices of emerging literary critics, and their engagement with the works of fictions and the other readers as they negotiate the boundaries of their interpretive communities. Consequently, we can see these graphs as capturing an emerging consensus of the complexity of a work of fiction, the relationship between authors and their works, the constitutive role that the acts of reading and reviewing play on the works in question, and the interpretive range of reader engagements that extends well beyond straight forward plot summary.

we can't know this. Maybe these verbs are very frequent in the story and are more likely to appear in reviews.

why are these nodes extra-diegetic? They are terms explicitly mentioned in the story

interesting case! A broader corpus including other epistolary novels (or widely famous and adapted in other media) may confirm/disprove this

quite a strong claim, I don't think we can make this inference (about a social and cultural role) only based on the available evidence. Moreover, there's no evidence that reviewers feel part of a community

why are they constitutive? in what constitutive process do they participate?

maybe this is just a reflection of the importance of subjectivity in reader response, rather than the complexity of the work itself

(c) *To Kill a Mockingbird*: Important and intangible actants such as “racism”, “lawyer”, “personality” compose the extended story network nodes in this graph. The “personality” node reflects the novel’s dedication to character development, be it of “scout”, “atticus” or even “arthur”.

(d) *The Hobbit*: The readers’ classification the novel’s genre is immediately apparent in the nodes “adventures”, “quest”, “home” and “journeys”. Inanimate actants such as “home”, “journey”, “quest” and “ways” typically have a very low out-degree (in this case 0) whereas “tolkien” has a very low in-degree. The node “ways” signals strategy: “Dwarves” have “ways” or “Bilbo Baggins” took “ways”.

(e) *Animal Farm*: The nodes “rebellion” and “revolution”, in conjunction with the nodes “power” and “control” highlight the sustained themes of power struggle, social dynamics and politics that lay at the ideological root of the novel. The author “orwell” once again has a high out-degree and the node “ways” once again signals strategy: “hens” think of “ways” and “pigs” wanted “ways”.

Figure 1: Expanded Story Network Graphs: The expanded story networks for the 5 literary works – nodes that represent characters in the story are in green while the actants extending the original character story network are in orange.

result of mis-remembering, forgetting or misreading, there are certain common features to these generated networks that, in the aggregate, reveal readers’ complex understanding of the target novel’s storylines. These features include: (a) The number of nodes is proportional to the number of distinct characters obtained from the EMG task considered jointly with the distinct relationship clusters between a pair of characters from the IARC task. Because of this correlation, *Animal Farm* and *The Hobbit* have more nodes than *Frankenstein*, *Of Mice and Men* or *To Kill a Mockingbird*, thereby revealing that reader reviews preserve this aspect of the narrative complexity of the target novel; (b) A node that aggregates a key event in a narrative is found to have a higher degree. For example, in *The Hobbit*, «Bilbo» finds «ring» is a central event that has a high in-degree and out-degree; (c) Contextual events such as «Lennie» dreams of «ranch» in *Of Mice and Men* that are true throughout the work but are inherently less subject to sequencing generally appear at the beginning of the sequence graphs, and help set the stage for the story. This effect is highlighted in *The Hobbit*, which is known for its detailed descriptions and intricate universe. Consequently, the event sequence estimated for *The Hobbit* has a large out-degree from the START node; (d) Key finale events such as «George» killing «Lennie» (*Of Mice and Men*) or «Bard» killing «Smaug» (*The Hobbit*) are suitably positioned towards the end of the sequence within the estimated objective timeline from START to TERMINATE; (e) Stories with multiple parallel subplots (such as *The Hobbit*) are shown to have a larger branching factor than linear stories (such as *Frankenstein* and *To Kill a Mockingbird*); (f) Disjoint paths from START to TERMINATE that do not interfere with the core network often describe supporting characters, the fringe events in which they place a role, and secondary story lines.

Each of these features provide us with specific information about how readers review (if not directly read) a novel, while also providing information on what individual readers see as

important (or remember as important), and also the multiple perspectives on the novel generated by large numbers of readers, thus echoing the reader response literature on communities of interpretation. Indeed, the multiple paths through the graph echo the idea that there are many ways to read – and review – a novel.

For example, *The Hobbit* is characterized by a series of richly drawn characters and an intricate world in which the action unfolds. The plot of the novel is complex and multi-stranded, with multiple pivotal high-degree event nodes.

Figure 2: *The Hobbit*: The complete event sequence network – nodes with degree > 2 are shaded turquoise and edges verifiable by at least 2 review samples are in red.

The emergent event sequence network for the novel has a high average branching factor, multiple high-degree nodes and parallel branches (see Figure 2). There are several relevant and verifiable event sub-sequences in this network including: 1. «gandalf» chooses «bilbo» → «bilbo» encounters «troll» → «bilbo» finds «ring» → «bard» kills «smaug»; and 2. «bilbo» leaves «hobbiton» → «bilbo» accompanies «gandalf» → «bilbo» leads «dwarf» → «bilbo» confronts «smaug» → «bilbo» deceives «smaug» → «bard» kills «smaug».

We empirically find that longer event sequences in this event sequence network (those that include many high-degree nodes) most often contain many critical details of the literary work’s plot. For instance, the path of event nodes with the maximum total node degree from START to TERMINATE for *The Hobbit* is shown in Figure 3. As such, there seems to be a broad reader consensus on which story points are “important”, while there are still many alternate pathways through the narrative thicket.

Events not explicitly connected in this chain (and in the network in general), may have precedence relationships between them. However, since our interest is to capture the most descriptive event sequence network, these shortcut edges are removed. A trivial example of this is the absence of edges from START to every single event. The resulting event sequence network can be verified against the precedence constraints determined by a majority of reviews. Some samples include:

Figure 3: *The Hobbit*: Highest total node degree event sequence in the network; edges such as the shaded edge shown are removed during sequencing

- «gandalf» chooses «bilbo» → «bilbo» encounters «troll»: "[...] Gandalf chose this particular hobbit to round out the unlucky number of the dwarves [...] trolls encountered by Bilbo and co. [...]"
- «bilbo» deceives «smaug» → «bard» kills «smaug»: "[...] [Bilbo] deceives the dragon, Smaug. [...] the dragon was slain by a man of the lake [...]"

These examples support our initial hypothesis that reviewers implicitly encode the relative structure of time, the *fabula*, in their reviews; our algorithm extracts precedence information to yield insightful and true event sequences. The examples also highlight the complexity of using the reviews’ tense and other linguistic structures to sequence events: the first review is entirely written in the past tense (*chose, encountered*), while the second review mixes the past and present tenses (*deceives, was slain*).

The relationship extractor is intrinsically not programmed to extract relationships in a sequential manner. *The Hobbit* features 4 two-node cycles and 0 cycles of > 2 nodes. As mentioned in the methodology section, two-node cycles are resolved intuitively by considering the dominant (most frequent) direction of an edge between the two nodes. A closer look into the pair of event nodes comprising a few of these averted cycles provides additional insight into the complexity of the sequencing task and into the difficulty of ordering highly correlated and concurrent events: [«bilbo» *finds* «ring» AND «bilbo» *meets* «gollum»], [«dwarf» *battles* «troll» AND «gandalf» *fights* «troll»].

It might be useful to revisit the pipeline that is driving the aggregation of these event tuples. The *man of the lake* is a mention of «bard» as aggregated by the EMG task; similarly the *hobbit* is labeled «Bilbo». The HDBSCAN-driven IARC process clusters *slain* with «kills», which is the label of that particular directed relationship cluster between «bard» and «smaug». The relative positions of these events are also noteworthy: «bard» kills «smaug» appears toward the end of the story, «bilbo» accompanies «gandalf» is placed toward the beginning. A two-node detached event sequence in this network involves «Thorin», a supporting character, whose activities are largely ignored by reviewers in the aggregated reviews. As a result, the character’s activities are not well-incorporated into the core event sequence structure.

The sequence networks resulting from the reviews of the other 4 novels – *Of Mice and Men*, *Frankenstein*, *To Kill a Mockingbird* and *Animal Farm* – are provided in Figure 4 and **the interactive graphs are included in the data repository** [55].

I couldn't find the interactive graphs in the data repository

(a) *Of Mice and Men*: Critical events such as «Lennie» killing «puppy», «Lennie» killing «Curley’s wife» and «George» killing «Lennie» occur in succession in the story and this sequencing is reflected in the reviewers’ posts.

(b) *Frankenstein*: The network correctly captures the order of important events including: «Frankenstein» creates «monster» → «monster» kills «Elizabeth» → «Frankenstein» chases «monster». We also capture Frankenstein’s regret upon his creating the monster. «Frankenstein» abandons «monster» occurs before his creating the monster, which is a false positive.

(c) *To Kill a Mockingbird*: The event, «Atticus» nurtures «Jem» and «Scout», appears in the early part of the event sequence. The core sequencing structure is also verifiable: «Tom» is accused of raping «Mayella» → «Atticus» defends «Tom» → «Bob» attacks «Jem» → «Boo» saves «Jem» → «Boo» carries Jem to «Atticus».

(d) *Animal Farm*: The story has several characters and routinely introduces events that have past precedent: the windmill has to be constructed twice; there are several uprisings on the farm; the commandments are changed multiple times. The character set promises to enlarge the event sequence network, while the similarities drawn between entities and their relationships in recurrent situations makes event sequence network generation a challenge. Nevertheless, we extract many useful sub-sequences including: «snowball» is exiled from «farmhouse» → «napoleon» attacks snowball with «pigs» → «napoleon» chases «snowball» → «napoleon» chases snowball off «farmhouse» → «pigs» change «commandments».

Figure 4: The complete event sequence network of the remaining 4 novels: nodes with degree > 2 are shaded turquoise and edges verifiable by at least 2 review samples are in red. The graphs can be further explored on our data repository [55]

The accuracy of our sequencing networks with respect to human perception across the five novels is presented in Table 1. The generally high accuracy (see Methodology) and tight error margins suggest not only that our networks consist of useful event sequences but also that our

precedence edges are consistent across reviewers: the similarity of the two scores suggest that the event tuples are themselves interpretable resulting in less confusion (there are few precedence relationships marked *X*). The higher error rate of *The Hobbit* and *Animal Farm* may be due to the style of each story; in *The Hobbit*, action scenes generally have many characters working together, in parallel or in the same setting, while in *Animal Farm*, there are many intentionally repetitive events that are difficult for the algorithm to resolve.

Story	score _{weighted} (%)	score _{simple majority} (%)
Of Mice and Men	92.35 ± 5.29	94.12 ± 5.88
The Hobbit	75.75 ± 5.45	77.27 ± 1.51
Frankenstein	88.00 ± 4.00	90.00 ± 3.34
To Kill a Mockingbird	95.71 ± 4.28	100.00 ± 0.00
Animal Farm	78.57 ± 4.50	80.77 ± 2.74

Table 1: Performance of the Sequencing Algorithm REV2SEQ on the 5 stories: the error margins were computed by estimating bounds for each score by replacing the labels marked *X* or *unsure* with all 0s or all 1s.

(c) SENT2IMP: Character Impression

(i) Single Character Impression

Our unsupervised method of character impression discovery provides insightful clusters about each character. A subset of these clusters for “Bilbo” in *The Hobbit* is described in Table 2. The first cluster provides a convincing argument that this character is *unpleasant*. The second one, in contrast, describes him as a hobbit of *impeccable personality*. These contradictory representations may capture a dichotomy of readers’ impressions of Bilbo. The third and fourth clusters, however, are comparatively different, revealing disparate information about the character not related to sentiment at all, characterizing *Bilbo* as both a burglar and a hobbit. Such findings justify our assumption that SVcop relationships are worth consideration, as they not only capture the readers’ broad range of perspectives on a character but also because these phrases form rich clusters of semantically aligned meanings that are not captured by standard supervised sentiment detection methods.

In addition to extracting rich clusters of actant-conditioned impressions, the HDBSCAN algorithm (with the default and constant distance threshold) clusters all *noisy* impressions into a separate cluster labeled “-1”. In this algorithm, we used a relatively high *eps* = 2 parameter to decrease the extreme sensitivity to noise. For example, for “Bilbo”, phrases such as [‘the uncle of Frodo’, ‘unbelievably lucky’, ‘nostalgic’] are classified in the noise cluster. More examples can be found in the last row of Table 2. Our results also show that there is a correlation between the perceived popularity of a character and the complexity of the impressions **he or she** elicits. These clusters of impressions can be informatively visualized with a dendrogram-heatmap that sorts similar clusters with respect to correlation scores (see Methodology for how this score is computed). In order to find a label for a cluster, we pick the most frequent word in the cluster’s phrase list excluding stop words.

A sample heatmap for “Victor Frankenstein” is shown in Figure 5. In this figure, there are three groups of impression clusters: 1) With labels such as, “brilliant scientist”, “scientist of story”, “responsible” and “instinctively good”; 2) A group that includes “young student”, “name of creator”, “name of man”; and finally, 3) another group comprising, “horrible person”, “selfish brat”, “real monster” and “mad scientist”. On closer inspection, a cluster labeled “Victor Frankenstein” stands out as a separate group with almost no correlation to other groups.

these infos should go in the methodology section or in appendix

I suggest using the more inclusive singular pronoun “they”

This representation also provides insight into the performance and limitations of BERT embeddings. Clusters that are similar should have a high similarity score (red) and clusters that are dissimilar should have a low similarity score (blue). For example, the clusters labeled “selfish,pitiful” and “horrible,person” have a high similarity score and the clusters labeled “scientist,mad” and “responsible” have a low similarity score. All the clusters are most similar to themselves so the major diagonal is deep red. There is a high similarity score between the clusters labeled “sympathetic,character” and “selfish,pitiful”, due to the similarity between representative phrase {“a sympathetic character”} in the first cluster and the phrases {“miserable”,“sad”,“pitiful”} in the second cluster: after all, Frankenstein was a “horrible person” for having created the monster but was also a person deserving of “sympathy” and “pity” for all the loss and grief that creation caused him.

Character	Descriptors
Bilbo	The Hobbit
Cluster 1	['not the interesting character', 'timid not', 'not enthusiastic', 'reluctant', 'not the type of hero', 'less cute', 'not as cool', 'unsure of situation', 'a small unadventurous creature', 'Perhaps just not the kind of character', 'not as important', 'less cute']
Cluster 2	['a true personality', 'an exemplary character', 'such a great character', 'resourceful', 'likable', 'still loveable', 'quite content', 'such a strong character', 'an amazing character', 'respectable', 'a great protagonist too', 'clever', 'such an amazing character', 'a peaceful', 'such an endearing character', 'a great choice', 'a fantastic lead character', 'quite engaging', 'cute', 'much charismatic character', 'such a fantastic Character', 'truly beautiful', 'enjoyable', 'just so charming', 'personable', 'able', 'the best character', 'quite skilled gets', 'awesome', 'smart']
Cluster 3	['of course the burglar', 'a thief', 'a thief go', 'to a burglar', 'to a thief', 'to a thief', 'the burglar', 'their designated burglar', 'could a burglar', 'of course the burglar', 'a Burglar', 'a Burglar']
Cluster 4	['a respectable hobbit', 'a respectable Hobbit', 'a sensible Hobbit', 'a clean well mannered hobbit', 'a respectable Hobbit', 'a sensible Hobbit', 'a proper hobbit']
Cluster 5	['small', 'small', 'little', 'small', 'little']
Cluster -1	['rich', 'the right man', 'a feisty character', 'the uncle of Frodo', 'unbelievably lucky', 'the perfect example of success', 'nostalgic', 'middle aged']

Table 2: Example impression clusters for “Bilbo” in *The Hobbit*: Clusters 1 and 2 describe impressions of “Bilbo”’s character while clusters 3 and 4 describe his profession and community. Cluster marked –1 is noise. Labels for each cluster are aggregated based on the most frequent monograms per cluster.

Figure 5: The (symmetric) heatmap for the character “Victor Frankenstein”: The similarity scores between clusters of impressions labelled by the row/column headers are computed by Algorithm 3. The sub-matrices that are deep red or blue imply a hierarchical structure to the mutual similarity or dissimilarity between groups of impression clusters. The diagonal entries are +2 as a cluster of impressions is most similar to itself.

(ii) Pairwise Character Impression Comparison

Generally, in a literary work, each character plays various roles across a wide range of events in the *syuzhet* or story line(s) of the novel. Characters most often also exhibit a diverse range of character traits. As such, each character is an individual, even if they share certain characteristics, or play similar roles, to other characters. For example, in *Animal Farm*, nearly all the characters are anthropomorphized animals, and live on a fictitious farm. Although there are multiple pigs, for example, in the story, each pig is distinctive from every other pig. In *To Kill a Mockingbird*, the characters are grounded in reality, sharing many recognizable characteristics (at least for American audiences) of small town America, and the central crisis of the novel and the myriad reactions of the characters creates an empathetic potential for the reader. Yet each reader brings to their experience of the novel a set of external experiences and conditions. These experiences allow each reader – and each reviewer – an opportunity to augment the construction of story lines and

characters in the novel. To avoid falling prey to the “intentional fallacy” [67], where a critic tries to untangle the intentions of an author, the methods we devise here turn instead to an exploration of the constitutive nature of the reader reviews. Because each reviewer brings with them their own unique approach to reading, and given the wide range of characters and events in a novel, one might expect that these characters, especially mined from reviews, cannot be compared.

We find, however, that, while writing reviews, reviewers collate their character impressions into clusters of descriptors that are more semantically consistent *across characters* than the raw reviews would initially suggest. **One possible reason for this finding could be a result of reviewers mapping their impressions into a shared consensus model of a character in an effort to write more convincing reviews and thereby receive more positive response from the broader community of reviewers.** Because of this semantic similarity, the impression clusters *enable inter-character comparison*.

The results of these inter-character comparisons may capture readers’ broader understanding of fictional characters, and the process by which communities of interpretation emerge. The alignments of character impressions across multiple fictional works may in turn reflect the consistency of approaches to reading, so that the text is constituted in a complex manner across many readings.

To illustrate these intriguing areas of character overlap across different works of fiction, we produce heatmaps for pair-wise comparison of distinct characters, as in Figure 6. Here we compare the impression clusters of “Victor Frankenstein” from *Frankenstein* to those of “Atticus Finch” from *To Kill a Mocking Bird*. The seemingly unlikely pair exhibit a surprising series of overlaps based on the readers’ impressions of these characters. For example, the two have a high similarity score for clusters describing aspects of gender, responsibility and overall strength of character (as evidenced by the row/column labels in the figure).

One particularly interesting similarity is found in the clusters labeled “father,kids” for “Atticus” and “creator,name” for “Frankenstein”. This similarity reflects a twofold process: first, the recognition of the readers of the similarity in these roles and second, the worldview encoded into BERT embeddings. BERT embeddings, as seen in single character heatmaps, carry additional artifacts into the realm of cross-character evaluation. For example, the cluster labeled “responsible” from “Frankenstein” and the cluster labeled “lawyer” from “Atticus” Finch have a highly negative similarity score. This suggests either that the readers have a negative bias against lawyers, or that pretrained BERT embeddings are biased, or both: regardless of the source of this bias, the combined model integrating reviewer comments and the cosine-distance measure when applied to BERT embeddings, seem to suggest that lawyers are *not* responsible.

this is an example of intentional fallacy applied to reader response rather than authors. I think it's more plausible that writing a review invites an attitude prone to the moral judgment of characters and events. Or that the SVCop method only catches a limited range of expressions regarding characters.

good point!

Figure 6: The (asymmetric) heatmap comparing the character “Victor Frankenstein” from *Frankenstein* and “Atticus Finch” from *To Kill a Mockingbird*: The similarity scores between clusters of impressions labelled by the row/column headers are computed by Algorithm 3. The color coding of impression clusters suggests valuable information stored in these representations about pairwise character similarity across novels, capturing the readers’ process of aligning impressions from one novel to impressions created while reading another novel.

Finally, we plot the entropy of the single-character heatmaps in an effort to quantify the character’s perceived *complexity*. The resulting bar plot is presented in Figure 7. Not surprisingly, the relative number of impression clusters empirically correlates to the entropy: reviewers describe a wider range of impressions for complex characters than for less complex ones. However, this feature alone cannot explain all the trends observed in the plot. For “Jones”, “Napoleon” and “Boxer” in *Animal Farm*, each character is associated with a roughly equal fraction of impression clusters, yet “Napoleon” emerges as a more complex character in the readers’ conceptualizations of *Animal Farm* characters; this is not surprising, as “Napoleon” is the **most enduring villain in the plot**. It is also noteworthy that the three actors are ascribed by readers similar roles in the plot and this similarity extends partially to their complexity measure.

this suggest that even the duration of the “presence on stage” of a character may be related to the variety of impressions

In *To Kill a Mockingbird*, “Atticus” is a central focus of the novel and it is his character that takes the spotlight as he defends “Tom”. Indeed, “Boo” and “Scout” appear in the novel in many scenes to support “Atticus”.

Of Mice and Men focuses on the dynamic between “George” and “Lennie”, a pair of characters with notably different personalities, and the inherent complexity in their relationship. The resulting duality in character impressions, the limited number of additional characters in the novel, and a linear timeline results in a similar complexity profile for this pair of actants. The readers’ impressions of characters that extend beyond the one-dimensional dismissal of inherently bad characters such as “Curley” may motivate them to focus intently on these two, plumbing the depths of their personalities and trying to understand their decisions in the context of a cruel environment.

The Hobbit rather rigorously follows the genre conventions of fantasy action-adventure, with a fairly clear delineation of “good guys” and “bad guys”. As a result, “Bilbo”, the main protagonist, attracts the most attention in discussion forums, which, in turn, contributes to a greater perceived complexity. Indeed, a feature of these complexity measures is the amount of attention an actant attracts from the reviewers. This aspect is not a failing, however, and captures instead a part of the mental model – here character impressions – that readers remember from the novel and feel are important enough to share with other readers.

Last, in *Frankenstein*, while the “Monster” wreaks havoc, it is in fact “Frankenstein” the scientist whose work raises ethical, moral and social concerns. Reader discussions about the character “Frankenstein” and his complex positioning in the novel ultimately foster debate about the purpose of science and frequently consider whether the scientist “Frankenstein” was perhaps the real monster. This effect is projected on both the relative number of impression clusters and the resulting complexity measure for the character in the reader reviews.

something's odd here: readers remember a mental model?

Figure 7: A measure of perceived complexity per character across novels: The color blue corresponds to the relative number of empirical samples per character-specific heatmap used to compute entropy (prior to smoothing). Each translucent color corresponds to a specific novel and plotted are the respective entropies of characters that have at least 4 impression clusters. We found $b = 50$, and $w = 3$ to be optimal hyperparameter choices to explore the differences in the complexity measure between characters.

The consensus impression and narrative models created by our framework enable one to turn the spotlight back on individual readers. While the individual reviews collectively reflect and encode the whole, the whole in turn constructs a rubric to better understand the parts, i.e. how individuals both align with and differ from the collective. For example, one might ask, what makes a review informative and useful? We may not be able to identify the exact features that constitute a good review but, according to our model, expanded interpretations of the overall story space, rich event sequences that emphasize main characters, and unique impressions of these characters constitute important proxy targets. If a review makes use of at least some of

these features, it will consist of enough information about the novel to be self-sustaining in the overall consensus discussion on the work without requiring access to that original work. Equally important is the presence of markers that indicate departures from the consensus models. Such variants not only set apart the individual from the rest, but might be the seeds for the future emergence of new collective impressions in an evolving dialog over novels and characters.

Consider, for example, an impression cluster for “Atticus Finch” extracted by the SENT2IMP algorithm and labeled “Man,Great” (see Figure 6). This cluster of impressions collected from all the reviews of the novel consists of the phrases: {‘a good father’, ‘the loving father’, ‘the best dad’, ‘a man of integrity’ ...}. Similarly, there is another impression cluster labeled “Father,Kids,” comprising the phrases { ‘the father of protagonist’, ‘the father of Jem’, ...}, and emphasizing his role as a father, while naming his children. Scout, the daughter and protagonist in the novel, has an impression cluster, labeled “Narrator,Smart” comprising the phrases: {‘really smart’, ‘very thoughtful’, ‘a smart girl’, ...}, and bringing out a key attribute that has given the novel a lasting legacy. Tom Robinson, yet another pivotal character, has an impression cluster “Innocent,Man” with the phrases: {‘a mere poor victim of circumstances’, ‘innocent’, ‘a good black man’, ...}, and correctly portraying him as a victim of racial bias and violence.

In light of the preceding collective impression clusters, let us consider the following review for *To Kill a Mockingbird*:

Review 1: “I think that *To Kill A Mockingbird* has such a prominent place in American culture because it is a naive, idealistic piece of writing in which naivete and idealism are ultimately rewarded. [...] Atticus is a good father, wise and patient; Tom Robinson is the innocent wronged; Boo is the kind eccentric; Jem is the little boy who grows up; Scout is the precocious, knowledgable child.”

The reviewer clearly aligns with and contributes to the majority views on the characters, Atticus Finch and Scout. The review also hints at an important event in the overall story line, namely that “Tom Robinson” is innocent (implicitly of a crime) but has been wronged, a situation captured clearly in our event sequencing graph (see Figure 4c). Reviews such as this one **contribute significantly more information** to the review ecosystem than a more cryptic review such as the following about *The Hobbit*:

Review 2: “Maybe one day soon I’ll write a proper review of *The Hobbit*. In the meantime, I want to say this: If you are a child, you need to read this for Gollum’s riddles. [...]”

This review is not only brief, but also skips references to a majority of the story lines, event sequences or character impressions. It does, however, emphasize the role of Gollum, a hugely popular character in the movie adaptation of *The Hobbit*, and the reader’s evaluation of the character’s riddles suitability for children. Our model thus provides an evidential measure, and one can objectively conclude –as admitted by the reviewer – that the review plays only a peripheral role in the overall review space, yet retains potential to seed further discussions.

6. Limitations

Reviews in *Goodreads* are often short (within a post), noisy (with non-ascii characters, emojis, incomplete sentences, slang expressions) and casual, marked by poor grammar and punctuation. In addition, not all of the reviews are reflective of a substantive engagement with the work in question. For example, **at one point it became popular among high school teachers to ask students to post reviews to *Goodreads* as homework** and, because of the forced nature of this assignment, some of the reviews are less thoughtful than might otherwise be expected. Possibly because of this – and frequent teacher admonitions not to write “plot summaries” – reviews that contain information about sequencing (which requires at least 2 eligible events occurring in succession to one another) are few. Furthermore, the poor punctuation in reviews, possibly reflective of the

but consider that this information is redundant, while Review 2 can generate new knowledge

anecdotal claim, evidence of the extent of this phenomenon is needed

ways in which readers write their reviews, often results in poor NLP performance during different stages of the relationship extraction pipeline.

Other limitations are typical of most unsupervised pipelines. The story network and expansion method, for example, uses the EMG task to cluster entity mentions into character labels. This mapping may not be intuitive at times: For example, some reviewers use generic terms for characters such as “the pig” or “a guy”. These general terms may create confusion in some novels where they could describe multiple characters (e.g. in *Animal Farm* there are many characters that are pigs).

The story sequencing pipeline is sensitive to rare events, since these often do not have connections with more commonly aggregated events. Rare events, consequently, are susceptible to isolation from reliable trajectories in the sequence network. Our greedy algorithm places these events at the earliest possible timestamp and, without much interaction with the core network, these events quickly reach the TERMINATE state. For example, in *The Hobbit*, «Gandalf» brings «Thorin» is an event that could be integrated into the mainline event trajectories but, since reviewers are not inspired to talk about this event, the event largely disappears. At other times, reviewers retrospectively provide their opinions and highlight later events early in their reviews. As a result, these reviews contribute to the phenomenon of certain narrative-altering events appearing earlier than they should when compared to the ground truth. Cycles that are broken by the SBFS algorithm may also include edges that are important in sequencing. Breaking multi-event (≥ 3) cycles greedily does not impact our sequences significantly because this scenario is rare, as illustrated by the fraction of edges (weighted from matrix M) neglected due to multi-event cycle creations relative to the total weighted edges in the processed precedence matrix, M_{ij} : *Of Mice and Men*: 0.0%, *Frankenstein*: 0.0%, *The Hobbit*: 0.0%, *To Kill a Mockingbird*: 17.6%, *Animal Farm*: 6.80%.

There are also certain novel-specific limitations to this work. For example, the event sequence network for *Animal Farm* is subject as a whole to more wrongly-ordered sub-event sequences than the other novels. *To Kill a Mockingbird* has a high fraction of edges that create cycles partially due to the very few eligible event sequence samples in reviews – the produced sequence network is sparse (see Figure 4). In addition, relationship extractors are poor at evaluating improperly punctuated, retrospective, qualified and/or otherwise syntactically sparse reviews resulting in noise in the aggregated relationships and derived trajectories. They are also poor at evaluating the order of relationships in sentences such as, “Frankenstein creates a monster, abandons it.” In this example, the relationship extractor cannot, with a general rule, evaluate whether the abandonment occurs before or after the creation. Much, but not all, of this noise is removed through aggregation across reviews.

Finally, in the impression extraction algorithm, several characters do not have enough clusters representing the readers’ different perceptions of that character. While this is, in part, a property of the reviews themselves – rare characters do not garner much attention – it limits our analysis of the *complexity* of characters to only popular ones: entropy can only be computed on those heatmaps that have representative samples across the range of similarity scores. Even with these scores, we employ a smoothing kernel the characteristics of which, including width w and bins b , change the absolute value of the entropy. Furthermore, the clustering of phrases into the “noise” cluster is highly sensitive to the *core distance* parameter in density clustering. While the finetuned BERT embeddings employed in this work are trained to optimize similarity based on the cosine distance, we have seen that there are inherent biases – the cosine distance between “lawyer” and “responsible” is highly negative. The embeddings are also occasionally random – proper names such as “Atticus” and “Bilbo” do not have tuned representations in the BERT space – and this affects the quality of the impressions heatmaps.

7. Conclusion

Reader reviews, such as those from *Goodreads*, are not often considered in the context of literary analysis. We believe, however, that they provide an intriguing window into the broad cultural

memories of “what a book is about.” Sophisticated analyses of theme, or the deep anchoring of a literary work in a detailed intellectual, social and historical context, may at times elude the thousands of reviewers contributing individual reviews to these social reading sites. Yet, despite these failings, the reviews still capture the meaningful thoughts of thousands of readers, each with their own diverse motivations for reading and reviewing, and are thus reflective of these readers’ literary engagement. Although they are usually unknown to each other, the readers of a particular work of fiction implicitly create an imagined community that shares, at least for some time, an interest in that work [1]. The evidence suggests that both individually and in the aggregate, these imagined communities of readers whittle down the novel to certain essential features: **a stripped down series of story lines** represented as relatively short yet accurate pathways through a narrative framework graph, itself a distillation of the most important—in the view of the readers—characters and relationships in the novel. Similarly, complex, dynamic characters **are conceptualized as a series of impressions** that, despite their simplicity in an individual review, capture in the aggregate some of the complexity of character that lies at the heart of fiction writing and literary analysis.

Importantly, our approach allows us to preserve an awareness of the individual reader who carries with them their own compact representation of a complex work of fiction while also contributing to a collective, and often more complicated, overview of that work. Because our methods capture both how an individual reads and reviews, and how the broader community of readers of the same work read and review, **it is possible to glimpse the relationship between a reader and the communities of interpretation that they are writing with, against and across**. The numerous pathways through the narrative framework, in that sense, capture the multiple ways that people understand, remember, and recount their own individual engagement with the work of fiction.

Although **a frequent refrain of teachers of literature is that amateur or otherwise “uninformed” engagements with literature are nothing more than “plot summary”**, our exploration of *Goodreads* countermands this criticism. The reviews we considered, for example, encode far more information than simple plot summary. Inevitably, reviewers include their impressions of one or two characters as well as some small number of events meaningful to them in their understanding of the novel. Readers, of course, draw their impressions of characters in any work of fiction not only from that work itself, but also from all of their experiences of other characters and events, both real and fictional. Consequently, by considering these reviews in the aggregate, one can derive insight into readers’ attempts to draw comparisons across novels, both on the basis of genre and story structure, and also on the level of character. As we show, readers’ impressions of a character from one novel resonate with similar impressions of a character from another novel – even if those novels are as unlike as *To Kill a Mockingbird* and *Frankenstein* – thereby **establishing a network of inference and allusion** that resonates throughout the collective reservoir of reading. What we discover in these reader reviews, when taken collectively, echoes – in a data-driven manner – some of the fundamental literary critical ideas of the relationship between readers and texts.

In short, our methods allow one to explore the individual and collective reimaginings of a novel – the constitutive aspects of reader response that have been at the foundation of several strands of literary criticism from the early phenomenological reader response theories of Iser and others, through the explorations of communities of interpretation advocated by Fish, to concepts of intertextuality rooted in the work of Julia Kristeva [68] and its resonance in the work of Roland Barthes [69] among others. So, while individual reviews might not tell the whole story, and may on the individual level fail to capture the complexity of characters, the collective impressions of thousands of readers provide important insight into how people read, remember, retell and review. In so doing, these methods allow us to do many things, including reassemble a portrait of a tortured scientist and his monster.

I think the methodology used (with clustering and filtering) specifically detected these story lines, but it's not reported in what percentage of the total reviews these story lines are present.

Again, the used methodology specifically looked for such “impressions”, there are likely other expressive forms with which readers share their thought about characters.

Great! Then you can easily show what proportion of readers talked about the main story lines and impressions.

I suggest to refrain from reporting anecdotes

I don't think your analyses can support this claim. You found a recurring pattern but it doesn't mean that the reader response pattern of a story is linked to those of other stories.

References

1. Benedict Anderson.
Imagined communities: Reflections on the origin and spread of nationalism.
Verso books, 2006.
2. Stanley Eugene Fish.
Is there a text in this class?: The authority of interpretive communities.
Harvard University Press, 1980.
3. Algirdas Julien Greimas.
Éléments pour une théorie de l'interprétation du récit mythique.
Communications, 8(1):28–59, 1966.
4. AJ Greimas.
Sémantique structurale, paris 1966.
ss. *On trouvera des applications de la méthode d'analyse du récit de Greimas p. ex. chez H. Quéré et al., Analyse narrative d'un conte littéraire: Le Signe de Maupassant,* " Doc. de travail, 9:200–222, 1968.
5. Algirdas Julien Greimas.
Les actants, les acteurs et les figures.
Sémiotique narrative et textuelle, pages 161–176, 1973.
6. Timothy R Tangherlini, Vwani Roychowdhury, Beth Glenn, Catherine M Crespi, Roja Bandari, Akshay Wadia, Misagh Falahi, Ehsan Ebrahimzadeh, and Roshan Bastani.
"mommy blogs" and the vaccination exemption narrative: results from a machine-learning approach for story aggregation on parenting social media sites.
JMIR public health and surveillance, 2(2):e166, 2016.
7. Timothy R. Tangherlini, Shadi Shahsavari, Behnam Shahbazi, Ehsan Ebrahimzadeh, and Vwani Roychowdhury.
An automated pipeline for the discovery of conspiracy and conspiracy theory narrative frameworks: Bridgegate, pizzagate and storytelling on the web.
PLOS ONE, 15(6):1–39, 06 2020.
8. Shadi Shahsavari, Pavan Holur, Tianyi Wang, Timothy R Tangherlini, and Vwani Roychowdhury.
Conspiracy in the time of corona: automatic detection of emerging covid-19 conspiracy theories in social media and the news.
Journal of computational social science, 3(2):279–317, 2020.
9. J. Boyarin.
The Ethnography of Reading.
University of California Press, 1993.
10. Elizabeth Long.
Textual interpretation as collective action.
Discourse, 14(3):104–130, 1992.
11. Steven Mailloux.
Interpretive conventions: The reader in the study of American fiction.
Cornell University Press, 2018.
12. Keith Rayner, Alexander Pollatsek, Jane Ashby, and Charles Clifton Jr.
Psychology of reading.
Psychology Press, 2012.
13. Wolfgang Iser.
The act of reading: A theory of aesthetic response.
JHU Press, 1979.
14. Wolfgang Iser.
Texts and readers.
Discourse Processes, 3(4):327–343, 1980.
15. Jane P Tompkins.
Reader-response criticism: From formalism to post-structuralism.
JHU Press, 1980.
16. G. Brenner.
Performative Criticism: Experiments in Reader Response.
State University of New York Press, 2004.
17. Inderjeet Mani.

- Computational narratology.
Handbook of narratology, pages 84–92, 2014.
18. BV Tomashevsky.
Theory of literature (poetics).
M.-L.: Gosizdat, 1925.
 19. Viktor Shklovsky.
Theory of prose [1925], trans.
Benjamin Sher (Elmwood Park, Ill.: Dal key Archive Press, 1990), page 102, 1991.
 20. Teun A Van Dijk.
Story comprehension: An introduction.
Poetics, 9(1-3):1–21, 1980.
 21. Walter Anderson.
Kaiser und Abt: Die Geschichte eines Schwanks, volume 42 of *Folklore Fellows Communications*.
Academia Scientiarum Fennica, 1923.
 22. Vladimir Propp.
Morphology of the Folktale, volume 9.
University of Texas Press, 2010.
 23. Roger C Schank and Kenneth Mark Colby.
Computer models of thought and language.
WH Freeman San Francisco, 1973.
 24. Mark Alan Finlayson.
Inferring propp's functions from semantically annotated text.
The Journal of American Folklore, 129(511):55–77, 2016.
 25. Patricia Galloway.
Narrative theories as computational models: Reader-oriented theory and artificial intelligence.
Computers and the Humanities, pages 169–174, 1983.
 26. Ira Goldstein and Seymour Papert.
Artificial intelligence, language, and the study of knowledge.
Cognitive science, 1(1):84–123, 1977.
 27. Mike Thelwall and Kayvan Kousha.
Goodreads: A social network site for book readers.
Journal of the Association for Information Science and Technology, 68(4):972–983, 2017.
 28. Mengting Wan and Julian J. McAuley.
Item recommendation on monotonic behavior chains.
In Sole Pera, Michael D. Ekstrand, Xavier Amatriain, and John O'Donovan, editors, *Proceedings of the 12th ACM Conference on Recommender Systems, RecSys 2018, Vancouver, BC, Canada, October 2-7, 2018*, pages 86–94. ACM, 2018.
 29. Mengting Wan, Rishabh Misra, Ndapa Nakashole, and Julian J. McAuley.
Fine-grained spoiler detection from large-scale review corpora.
In Anna Korhonen, David R. Traum, and Lluís Màrquez, editors, *Proceedings of the 57th Conference of the Association for Computational Linguistics, ACL 2019, Florence, Italy, July 28-August 2, 2019, Volume 1: Long Papers*, pages 2605–2610. Association for Computational Linguistics, 2019.
 30. Daniel Allington and Bethan Benwell.
Reading the reading experience: an ethnomethodological approach to 'booktalk'.
University of Massachusetts Press, 2012.
 31. Amal Htait, Sébastien Fournier, Patrice Bellot, Leif Azzopardi, and Gabriella Pasi.
Using sentiment analysis for pseudo-relevance feedback in social book search.
In *Proceedings of the 2020 ACM SIGIR on International Conference on Theory of Information Retrieval*, pages 29–32, 2020.
 32. Beth Driscoll and DeNel Rehberg Sedo.
Faraway, so close: Seeing the intimacy in goodreads reviews.
Qualitative Inquiry, 25(3):248–259, 2019.
 33. Lala Hajibayova.
Investigation of goodreads' reviews: Kakutanied, deceived or simply honest?
Journal of Documentation, 2019.
 34. Lisa Nakamura.

- Words with friends": Socially networked reading on " goodreads.
Pmla, 128(1):238–243, 2013.
35. Bronwen Thomas and Julia Round.
Moderating readers and reading online.
Language and Literature, 25(3):239–253, 2016.
 36. Stefan Dimitrov, Faiyaz Zamal, Andrew Piper, and Derek Ruths.
Goodreads versus amazon: the effect of decoupling book reviewing and book selling.
Proceedings of the International AAAI Conference on Web and Social Media, 9(1), 2015.
 37. Suman Kalyan Maity, Abhishek Panigrahi, and Animesh Mukherjee.
Book reading behavior on goodreads can predict the amazon best sellers.
In *Proceedings of the 2017 IEEE/ACM International Conference on Advances in Social Networks Analysis and Mining 2017*, pages 451–454, 2017.
 38. Simon Peter Rowberry.
The limits of big data for analyzing reading.
reading, 2019.
 39. Vincent Fortuin, Romann Weber, Sasha Schriber, Diana Wotruba, and Markus Gross.
Inspireme: Learning sequence models for stories.
Proceedings of the AAAI Conference on Artificial Intelligence, 32(1), Apr. 2018.
 40. Parag Jain, Priyanka Agrawal, Abhijit Mishra, Mohak Sukhwani, Anirban Laha, and Karthik Sankaranarayanan.
Story generation from sequence of independent short descriptions.
CoRR, abs/1707.05501, 2017.
 41. Xing Fang and Justin Zhan.
Sentiment analysis using product review data.
J Big Data, 2, 12 2015.
 42. Khairullah Khan, Baharum Baharudin, Aurnagzeb Khan, and Ashraf Ullah.
Mining opinion components from unstructured reviews: A review.
Journal of King Saud University - Computer and Information Sciences, 26(3):258–275, 2014.
 43. Maria Antoniak, Melanie Walsh, and David Mimno.
Tags, borders, and catalogs: Social re-working of genre on librarything.
Proc. ACM Hum.-Comput. Interact., 5(CSCW1), April 2021.
 44. Gerhard Weikum, Luna Dong, Simon Razniewski, and Fabian Suchanek.
Machine knowledge: Creation and curation of comprehensive knowledge bases, 2021.
 45. Gabor Angeli, Melvin Jose Johnson Premkumar, and Christopher D. Manning.
Leveraging linguistic structure for open domain information extraction.
In *Proceedings of the 53rd Annual Meeting of the Association for Computational Linguistics and the 7th International Joint Conference on Natural Language Processing (Volume 1: Long Papers)*, pages 344–354, Beijing, China, July 2015. Association for Computational Linguistics.
 46. Denny Vrandečić and Markus Krötzsch.
Wikidata: A free collaborative knowledgebase.
Commun. ACM, 57(10):78–85, September 2014.
 47. Fabian M. Suchanek, Gjergji Kasneci, and Gerhard Weikum.
Yago: A Core of Semantic Knowledge.
In *16th International Conference on the World Wide Web*, pages 697–706, 2007.
 48. Mary Shelley.
Frankenstein. london: Lackington, hughes, harding, mavor, and jones, 1818. ed. stuart curran.
Romantic Circles Electronic Editions, 16, 2015.
 49. J Steinbeck.
Of mice and men. new york: Covici & friede, 1937.
 50. John Ronald Reuel Tolkien.
The Hobbit.
Houghton Mifflin Harcourt, 2012.
 51. John Griffin and George Orwell.
Animal farm.
Harlow: Longman, 1989.
 52. Harper Lee.
To kill a mockingbird. philadelphia & new york, 1960.
 53. Joseph Campbells.

The hero with a thousand faces.

Princeton, New Jersey: Princeton University Press, 1973.

54. Kurt Vonnegut.
At the blackboard: Kurt vonnegut diagrams the shapes of stories, 2005.
55. Pavan Holur, Shadi Shahsavari, Ehsan Ebrahimzadeh, Timothy R Tangherlini, and Vwani Roychowdhury.
Modeling social readers: Novel tools for addressing reception from online book reviews, May 2021.
56. Shadi Shahsavari, Ehsan Ebrahimzadeh, Behnam Shahbazi, Misagh Falahi, Pavan Holur, Roja Bandari, Timothy R. Tangherlini, and Vwani Roychowdhury.
An automated pipeline for character and relationship extraction from readers literary book reviews on goodreads.com.
In *12th ACM Conference on Web Science*, WebSci '20, page 277–286, New York, NY, USA, 2020. Association for Computing Machinery.
57. Seymour Benjamin Chatman.
Story and discourse: Narrative structure in fiction and film.
Cornell University Press, 1980.
58. Branimir Boguraev and Rie Kubota Ando.
Timebank-driven timeml analysis.
In *Dagstuhl Seminar Proceedings*. Schloss Dagstuhl-Leibniz-Zentrum für Informatik, 2005.
59. Nils Reimers and Iryna Gurevych.
Sentence-bert: Sentence embeddings using siamese bert-networks.
In *Proceedings of the 2019 Conference on Empirical Methods in Natural Language Processing*. Association for Computational Linguistics, 11 2019.
60. Martin Ester, Hans-Peter Kriegel, Jörg Sander, and Xiaowei Xu.
A density-based algorithm for discovering clusters in large spatial databases with noise.
In *Proceedings of the Second International Conference on Knowledge Discovery and Data Mining*, KDD'96, page 226–231. AAAI Press, 1996.
61. Leland McInnes, John Healy, and Steve Astels.
hdbscan: Hierarchical density based clustering.
The Journal of Open Source Software, 2(11), mar 2017.
62. Claude Sammut and Geoffrey I. Webb, editors.
TF-IDF, pages 986–987.
Springer US, Boston, MA, 2010.
63. D. N. Joanes and C. A. Gill.
Comparing measures of sample skewness and kurtosis.
Journal of the Royal Statistical Society. Series D (The Statistician), 47(1):183–189, 1998.
64. S. Kokoska and D. Zwillinger.
Crc standard probability and statistics tables and formulae, student edition.
1999.
65. Simon DeDeo, Robert X. D. Hawkins, Sara Klingenstein, and Tim Hitchcock.
Bootstrap methods for the empirical study of decision-making and information flows in social systems.
Entropy, 15(6):2246–2276, 2013.
66. Harper Lee.
Go Set a Watchman.
Harper Collins, 2015.
67. William K. Wimsatt and Monroe C. Beardsley.
The Verbal Icon.
University of Kentuck Press, 1954.
68. Julia Kristeva.
Desire in language: A semiotic approach to literature and art.
Columbia University Press, 1980.
69. Roland Barthes.
The death of the author.
Contributions in Philosophy, 83:3–8, 2001.

incomplete reference

Appendix B

A Response to Editor(s) and Reviewers

Thank you for your recent evaluation of our article, “Modeling Social Readers”. We appreciate the time and care that you and the referees have obviously taken in considering our work, and believe that many of the suggestions, which we have incorporated into our revised paper, contribute to making the paper stronger. We found the suggestions, individually and collectively, to be quite helpful in guiding the substantive revisions that we have made. These changes can be tracked in the “diff” version that we include here.

To ensure that our responses are properly understood, we present short responses to the excellent suggestions made by the two referees in comments below. We consider their comments in the order in which they appear in the two separate reviews. For each comment, we respond in a short section. These changes can also be tracked in the accompanying “diff” version.

Again, we want to express our sincere thanks for the deep and thoughtful comments provided by each of the referees.

In the following response to specific issues raised in the reviewer comments, we have printed (in grey), the relevant (and necessarily abbreviated) parts of the reviewer comments, and in the subsequent paragraph (in black), responded to the reviewer’s concerns.

Reviewer: 1

A. Comments about the Related Work:

I liked reading this research and I think it has a great potential. However, there are some flaws that don't make it publishable in the present form.

We thank the referee for this broad positive assessment of our work. We have worked to address these flaws in our updated version of the paper. In our answers below, we respond to specific comments and suggestions.

Comment 1 (abbreviated)

The authors ignore a large body of literature on the computational analysis of reader response... The account of the advancements in literary studies is inadequate... I suggest to start the “Related work” section drawing from the existing research about Goodreads and other social reading platforms... The authors present an original contribution but it’s framed inappropriately.

Answer 1: Thank you for this suggestion. In the current version of the article, we have made the overview discussion of reader response more compact, and framed our article as contributing to the growing body of work on the computational analysis of literature review sites. We now include a fuller discussion of this related work in the

context of studying social reading and the review space on these platforms as a means for framing our work.

We have significantly rewritten and improved the Related Works section to a) incorporate the suggested references including Finn (2011), Rebora et al (2018 and 2019), and Pianzola et al (2020). We have also included exciting new work, such as that by Mimno et al (2021) among others in the context of LibraryThing; b) we frame the work in this new context of considerable work on studies of social reading sites; c) we cover a broader swath of previous work in the use of computational methods to advance literary studies in general, and narrative analysis and social reading sites in particular; and d) add additional references to work done specifically on the Goodreads corpus. We believe these additions properly situate our work as providing complementary methodological advances to the various advances described in these new references. Again, we thank the referee for directing us to this work.

B. Comments about the Methodology:

In the introduction the rationale of the methods employed is only briefly sketched. ...It would be easier – and more meaningful for an audience of literary scholars...to have an explanation of the motivation for looking at certain features... My suggestion is to simplify the Methodology section explaining a bit more in detail...and moving the technical part in appendix.

Answer 2: Thank you for your helpful comments concerning how to address the important technical aspects of the paper for a potentially non-technical audience. We recognize that the Methodology section for each constituent tool (Expanded Story Graphs, REV2SEQ, SENT2IMP) could be more clearly expressed and better motivated for this type of interested and engaged, yet largely non-technical, audience. Therefore, we have added/refined the introductory paragraphs to each of these methodological sections, and summarize them below:

Expanded Story Networks (see Page 7 Sec (a) Paragraph 1): Each review on its own consists of a fragment of the overall narrative framework and represents in the aggregate the underlying knowledge shared by most of the reviewers of a novel. Our goal is to extract this narrative framework. Reviews often contain partial story summaries that retell some portion of the relationships between characters. For example, in *Frankenstein*, several reviewers write about the complex and diverse relationships between Frankenstein and his monster. Yet other reviews mention the relationship between the monster and Frankenstein's wife. Capturing these relationships from not just one post, but jointly and scalably across many posts, provides a view of the reader review space representing a static summary of the different characters and their inter-character relationships. A natural way to model these relationships is through a network where the nodes are entities {Frankenstein}, {Frankenstein's monster} and {Frankenstein's wife}, for example, and the edges between the nodes represent inter-actant relationships.

The natural semantic structure of reviews brings with it the presence of meta-characters, those who do not directly feature in the novel (such as the author or the director of a movie adaptation). This affords us an interesting opportunity to explore this meta-discursive space in the context of discussions of the novel, an important aspect of the readers' contextual understanding of a novel. Consequently, in the process of aggregating relationships between characters, we also aggregate relationships that include these meta-characters.

REV2SEQ (see Page 8 Sec (b) Paragraph 1,2): With the expanded story network alone, we have not yet included the event sequence and timing information embedded within the reviews. Some reviewers of *The Hobbit*, for instance, not only write (a) that “Bilbo meets Gandalf” and (b) that “Bilbo faces Smaug”, but also that Bilbo faces Smaug *after* meeting Gandalf. We would like to capture this additional information as a means for understanding how readers imagine the sequence of events in a novel. To complement the aggregated representation of the novel in the Expanded Story Graphs, we present REV2SEQ, a dynamic event sequencing algorithm that attempts to order the events in a story, as reflected in the reviews, by finding the optimal (consensus) route(s) through all the identified events. In this tool, events are described in a very simple manner: each event consists of the involved character nodes and the relationship edge between them at a particular time.

SENT2IMP (see Page 11 Sec (c) Paragraph 1,2): Considering the descriptive phrases about a novel's characters should reveal the different dimensions of character traits not directly expressed by the author but added by the readers. This is different from typical NLP tasks such as Sentiment Analysis (SA) which uses a coarse-grained Good/Bad/Neutral classifier to segment phrases. For example, while a review <<Bilbo is small>> directly associates the property of being “small” to Bilbo, RoBERTa-large-based SA classifies this phrase coarsely as “Negative”.

By way of contrast, we do not want to impose a predefined set of categories on reader impressions of characters. Instead, we hypothesize that, given the various descriptive phrases pertaining to a character, their numerical representations in a vector space should cluster into separate and interpretable groups of descriptors that conserve the rich and varied opinions of users. Such a per-character embedding in a vector space model (VSM) now facilitates character comparison as well: do a pair of clusters pertaining to different characters overlap? Along which dimensions of reader impressions are two characters said to be similar? What about characters across novels? SENT2IMP attempts to construct a simple unsupervised framework that answers these questions.

As far as the reviewer's suggestion of moving the technical part to the Appendix, we hope that the added non-technical descriptions and motivation sections in the Methodology, along with the algorithmically-simple models used in the different tools, make the technical part considerably more approachable for a non-technical audience. By keeping the technical aspects in the main section, now introduced with easily understood motivating paragraphs, we hope to show that the approaches are straight-forward and interpretable, thus avoiding the lamentable situation where non-technical audiences see machine learning approaches as impenetrable black boxes. We find the referee's suggestion to include these introductory paragraphs to be an excellent one, and believe it allows us to keep the technical discussions in place, which should be of considerable interest to the increasing group of computationally savvy literary scholars and scholars of data/culture analytics as well as the broader NLP and ML communities.

Comparisons to Other Approaches:

I wonder whether simpler techniques would lead to different results. If the authors want to make claims in favour of their (computationally expensive) methodology, I think they should show how it performs in comparison to simpler methods. E.g. is a story network more informative than topic modeling? Is SENT2IMP more effective than measuring the semantic similarity of characters' names (and neighbour words) based on a vector space model of aggregated reviews...?

Answer 3: We thank the reviewer for this discussion of “simpler techniques,” and for pointing us to the work by Jacobs (2019), which we have incorporated into our revision. Below is a brief comparison of that work with ours - in particular our SENT2IMP model. We also provide a brief comment about topic modeling.

Jacobs uses *SentiArt* to predict the “emotion potential” of a passage of text, and also performs expanded sentiment analysis (SA) on famous characters in a novel through 2 tasks: Emotional Figure Profiling and Figure Personality Profiling.

The methodology can be succinctly described as follows: (i) Customize a vector embedding obtained from a large corpus such as FastText or GLoVE to the novel/book so that **each word** in the novel (and of course every other word in the language) has a vector representation in a shared embedding space. (ii) Select a set of predefined words, **Set-A**, (for example from *Eckman 99*, (<https://doi.org/10.1037/0033-295X.99.3.550>)), where each word has been manually labeled to **have either positive or negative emotions/valence**, e.g. happy vs sad. (iii) Select a set of predefined words, **Set-B**, (again, for example, *Eckman 99*), where each word has been manually labeled to have either **positive or negative arousal**, e.g. excited vs calm. (iv) Finally, represent each word (including character names) as a point in a 2D space: where the X-axis is the net valence (the signed sum of the cosine similarity between the vector representations of the word and the words in Set-A); and the Y-axis is net arousal, computed similarly using the words in Set-B. This 2-D representation can also be mapped to a single scalar number by taking the product of the two scores, which is referred to as the Emotion Potential.

In the context of our work, this would imply that a character is viewed at most as a 2-dimensional object in the valence-arousal space, based on expert labels. While both this 2-D and the associated 1-dimensional representations on the Emotional Potential line can provide useful insights, it is clearly a limited, hard-coded view of characters and readers' perceptions of those characters, which are by design multidimensional and are often not adequately captured by predefined labels.

Our work is specifically intended to explore whether such multidimensional and nuanced complexity of characters can be captured effectively. To this end, we let data drive the process, and follow the thesis that each dimension of a character will comprise a well-defined cluster of vectors in the embedding space. In particular, we use an NLP dependency parsing tree tool to aggregate all the descriptive phrases for each character and then use pre-trained BERT as an embedding space. We then apply clustering tools to discover the different impression clusters. **Each such group of semantically similar phrases -- capturing a single aspect of a character in the eyes of the readers -- is defined as an impression.** This type of unsupervised, multidimensional representation of characters not only has the potential to capture nuanced and often contradictory impressions of a single character, but also holds the potential to create representations that enable one to capture similarity/dissimilarity in character impressions across many different novels.

In the following brief notes, we provide additional, more technical details on how our work differs from Jacobs', and some of the limitations of working at the word level and performing sentiment analysis using labeled data.

1. Unsupervised Analysis: Our analysis steers clear of comparing the *impressions* about characters, a broader form of sentiments, to hard-coded groundtruth vectors. In Jacobs', this hard-coded set was drawn from (among others) a larger set of emotion words such as *Eckman 99* (see <https://doi.org/10.1037/0033-295X.99.3.550> for a reference). In the set(s) used, each emotion word has a measure of valence (the extent of happiness or sadness, for example: content vs depressed) and arousal (the intensity of the emotion, for example: excited vs calm).

While we appreciate the benefits of projecting character representations into a minimal vector space, we instead attempt to estimate the features directly from the data (expanded sentiment clusters) that are important in defining a character. An unsupervised approach such as this is an important step toward infinite context-conditioned character pair comparison, a task that cannot scale when vectors are projected into predefined and limited context spaces.

2. Benefits of Phrase-Level Representations and Syntactic Parsing: While continuous bag-of-words, skip-gram Word2Vec, and co-occurrence-based GloVe models have been shown to perform exceptionally well at the word-level, recent developments in the field of Natural Language Processing have had significant success in surpassing these models:

- a. **Richness of the Representation:** We believe that reader impressions are not found at the word-level and must at the very least exist at the phrase-level. For instance, consider the phrase

Bilbo Baggins {is not useless in the plot}.

In this circumstance, a word-level model might attempt to (i) Find the relevant word to represent Bilbo; i.e {useless}, (ii) Find the group of words that represent Bilbo and sum them $V\{\text{not}\} + V\{\text{useless}\}$; or (iii) Represent Bilbo as an averaged representation across contexts via the Skip-Gram Model. This is the method implemented by Jacobs.

Method (i) above would yield the incorrect attribute to Bilbo; method (ii) would average vectors that, up to some context length, remains interpretable before quickly becoming noise; and method (iii) provides an aggregated representation of Bilbo, as opposed to a *distribution of impressions* that SENT2IMP is designed to discover.

A lack of semantic understanding can be found even in very recent word-level sentiment analysis models. GLoVE+LSTM sentiment classification, for example, on the above-mentioned sample phrase returns **a somewhat negative sentiment**. This output most likely stems from the GLoVE word-level representation for {useless} dominating the representation for {not} in classifying a phrase that contains both.

By way of contrast, our BERT model tries to represent the entire phrase {is not useless in the plot} with one representative vector. Other phrases with similar semantic intent, for example, {is active in the plot}, can then be clustered along with this one. At the level of phrases, the vector representations of {is not useless in the plot} and {is active in the plot} are close to one another.

b. **Syntactic Parsing and the Computational Limits of Window-Length**

Limited Vector Representation: An issue with the skip-gram based approach is that the locality of the words surrounding a character play an important role in defining the attributes of the character. While most models, including even the most recent GPT3 model, uses such context, this approach favors Sequential/Masked Language Modeling more so than Language Understanding. Consider the following example:

{{Bilbo Baggins, who left Hobbiton to conquer his quest}}, was an amiable character.

Extracting descriptive attributes of characters from sentences with a finite window length (maked here by {{ }}) is not guaranteed to capture the description of Bilbo Baggins as amiable and funny.

In SENT2IMP we circumvent this problem by extracting descriptive relationships using Dependency Parse Trees that evade the window-length

limit. Instead, we extract clauses at the sentence-level invariant of the interword distance.

For example, the comment above, when parsed with our dependency parse tree pattern specifically tuned to extract adjective phrases, returns a relationship tuple that describes Bilbo:

Bilbo Baggins → was → an amiable character.

- c. Entity Mention Grouping:** Our method uses an entity mention grouping layer that uses the relationships between characters to aggregate descriptive phrases for each entity (for example, Bilbo Baggins can be referred to in reviews as {Bilbo, Baggins, Bilbo Baggins, The Hobbit, the hero, the thief}). This is a challenging task in the word-based windowed context model.

The benefit of such a semantically-informed aggregation is that character descriptions can be aggregated more effectively. A descriptive phrase describing entity A via a mention A1 and another phrase via mention A2 can be compared to one another with the prior knowledge that $A1 \sim A2$.

3. Computational Complexity - The reviewer has raised concerns about the computational complexity of our methods. Our model is no more computationally complex than prior work, especially the impactful work of Jacobs. Both works deal with pretrained models for generating vector embeddings of words and phrases; these models are readily available and using them to generate vector representations is a straightforward computational task. In both works, semantic similarity of the underlying words and phrases are approximated by cosine similarity, a very fast and computationally inexpensive method. Admittedly, the sequencing work in our paper requires basic graph theoretic knowledge, but we do not see how it can be done in any other way.

How does a story graph compare to topic modeling? First, we recall a succinct definition of Topic Modeling: “a type of statistical modeling for discovering the abstract ‘topics’ that occur in a collection of documents”. When translated to practice, it involves (i) a corpus of N distinct documents, (ii) a set of k distributions over the vocabulary of words, each such distribution (listed as a bag-of-words ordered by their probability) representing a topic, (iii) each document is mapped as a distribution over these k topics. As such, it is an effective way of embedding a document in a k-dimensional space, so that similarity among documents can be estimated. Topic modeling is useful to identify the different themes (as can be captured by different bags of words) in an ensemble of documents and therefore group documents by their similarities.

Performing topic modeling on our corpus of reader reviews (each review being considered as a document) about the same novel would be uninformative beyond finding the groups of reviews specific to characters, locations, etc. It would not enable one to perform the tasks that this paper is designed for. For example, we present

here the topics using BERTopic (number of topics = 10) from the reviews of *Of Mice and Men*. They do not appear to be informative and ill-suited for our tasks. The topic descriptions focus too heavily on the characters whose mentions are in the different reviews.

Top Descriptive Words per Topic
steinbeck, book, character
have, read, book
lennie, george
lennie, george, men
lennie, george, book
him, book, ending
lenny, george, him
american
book, mice, men
hard, friendship, book

Reviewer: 2

As with our response to Reviewer 1, we respond below to specific concerns Reviewer 2 has raised with our work. We hope that these responses, as well as the substantive revisions we have made, including the clarification of the “community of interpretation(s)” approach that the reviewer rightfully notes, adequately address these concerns.

This is a comprehensive and detailed account of technically sound research that fits the criteria for publication in RSOS.

We thank the reviewer for this broad and positive review of our work.

What was not clear to me, however, is the repeated reference to an “imagined community” of readers... These readers appear not to directly interact with each other – only to engage in a common task (reading the same book). This is described as a “collective enterprise of literary analysis” but I don’t see how the task is “collective” when it is individual... I am merely asking for a little more clarity here; do the reviewers interact or do they not?

Answer 4: We thank the reviewer for bringing to light this confusion. We hope the revision to the introduction and Answer 9 clarify this point.

I did not manage to understand from the introduction alone how the nodes are connected in the network.

Answer 5: The reviewer asks for clarification about the construction of the narrative network in the Expanded Story Graphs section. In the introduction, we now have

included an expanded description of the nodes and how they are connected (i.e. what constitutes an edge) in the network. In brief, the constructed aggregated representations of the 5 novels are represented as character interaction networks, consisting of actants as nodes and inter-actant relationships as edges. We expand this beyond the simple character interaction graph by relaxing the concept of actant, and including actants from the extra-discursive space of the novel's broader context (thus including the author, filmic versions, the film directors, etc). This is then our narrative network.

Nodes: A node exists if the character referred to by the node's label, or its constituent mentions, appears in a post. For example, the character {Victor Frankenstein} might be referred to, in posts, as Victor, Frankenstein, Victor Frankenstein, the creator, the scientist.

Edges (connecting the nodes): Edges between a pair of nodes are aggregated by isolating the verb phrases that connect the two characters in the review text. Isolating the verb phrase requires extracting *relationships* from the unstructured text, a methodology rooted in a previous work by Tangherlini et al (<https://doi.org/10.1371/journal.pone.0233879>.)

I would very much appreciate this to be elucidated – and clarification as to whether reviewers interact directly or not.

Answer 6: Please see answer 9 below.

Section 2 on related work is useful but not quite complete and the references may need to be checked.

Answer 7: We thank the reviewer for bringing this to our attention. We have checked the references, corrected any infelicities, and augmented the Related Work in accordance with both reviewers' suggestions.

I find the term “narrative network” confusing.

Answer 8: We hope that Answer 5, in addition to the updates to the Methods section provide the necessary clarification. Again, thank you for bringing this to our attention.

[T] his statement puzzles me: “Within this shared timeline, reviewers intersperse their opinions, argue over aspects of theme and plot, and offer other thoughts about style, imagery or comparisons to other works of fiction.” This gives me the impression the reviewers argue against each other- that they are in a dialogue. Again I remain unsure – are there interactions between reviewers or not (as in, do they actually influence each other).

Answer 9: As is common in most social media platforms, some reviewers engage more than others in back-and-forth conversations in the comment threads in order to clarify and justify a plot point. While reviewer interactions occur and might bias a particular post one way or another, we view the dataset collectively as a result of a

natural experiment on a collaborative online forum for free dialogue. *In the context of Fish's concept of "interpretive communities," there is a general sense that readers, whether or not they had spoken with each other or otherwise interacted, were part of a community whose interpretations of a literary work were similar.* On social media platforms, this community increases considerably in size, in some ways capturing Benedict Anderson's notion of the "imagined community." Here, we recognize that, by virtue of reviewing a book, and in engaging with the reviewing platform, the reviewers have joined an imagined community of book readers whose interactions, whether in the comment threads, or in the asynchronous nature of reading others' reviews, created a community (or communities) of interpretation. When we read, we necessarily read in the context of what we have read before; and when reviewers write a review, it is in this spirit of contributing to an ongoing and growing interpretive body of knowledge about the work in question. In the work presented here, we are not focused on explicitly mining the source network of the interconnected users on the platform, but instead restrict the analysis to the source network of the narrative they recall and modify. Any deliberations, debate and uncertainty among users flow through our tools to the output. These tools remain invariant to this collective bargaining as well as to the extent of it. An explicit example can be observed in our representation of character complexity wherein dissonance among users about a character's purpose, whether they express it against one another or to the Goodreads platform blind to others' opinions, is captured in our complexity histogram (see Figure 7).

On the general issue of the collective versus the individual, with interactions I point to M Krasnytska et al., Ising model with variable spin/agent strengths 2020 J. Phys. Complex. 1 035008 <https://doi.org/10.1088/2632-072X/abb654> which the authors of this submission may find relevant.

Answer 10: We thank the reviewer for referring this paper to us. Krasnytska et al. generalize the Ising model such that the spins of each constituent particle are not confined to binary $\{+/-1\}$ and instead consider a continuous exponentially distributed spin. The subsequent modeling of social networks is appealing because with such a model we can account for finer relationships between actants rather than binary EXIST / DOES NOT EXIST relationships. We will consider this model in future work; this approach may well allow us to model social networks that generate these narratives via the Ising Model (IM).

All in all, I have no doubts that this paper fits into the criteria of RSOS. In fact it does so exceptionally well. NLP knowledge, which is in relatively infancy in the community, is advanced here. But in addition to that, as an individual, I view this as a very interesting body of work, and I am happy to have had occasion to read it. I have no demands for improvements – just some light suggestions to enrich the references for the benefit of people who will want to know how this contribution fits in to a wider set of contemporaneous literature.

Again, we thank the reviewer for this endorsement of our work, and the very interesting points brought up in the review. These comments engendered a great deal of interesting conversation in our group, and we hope our revisions capture that.